# PoLAR: Polar-Decomposed Low-Rank Adapter Representation

**Kai Lion**    **Liang Zhang**    **Bingcong Li**[*]    **Niao He**[*]
Department of Computer Science
ETH Zurich
Zurich, Switzerland
{kai.lion, liang.zhang, bingcong.li, niao.he}@inf.ethz.ch

## Abstract

We show that low-rank adaptation of large-scale models suffers from a low stable rank that is well below the linear algebraic rank of the subspace, degrading fine-tuning performance. To mitigate the underutilization of the allocated subspace, we propose PoLAR, a parameterization inspired by the polar decomposition that factorizes the low-rank update into two direction matrices constrained to Stiefel manifolds and an unconstrained scale matrix. Our theory shows that PoLAR yields an exponentially faster convergence rate on a canonical low-rank adaptation problem. Pairing the parameterization with Riemannian optimization leads to consistent gains on three different benchmarks testing general language understanding, commonsense reasoning, and mathematical problem solving with base model sizes ranging from 350M to 27B.

## 1 Introduction

Large language models (LLMs) transformed the field of machine learning through their proficiency in understanding and generating text, as well as handling complex multimodal tasks. Typically, those models require a number of parameters that is on the scale of billions, rendering fine-tuning of large-scale models infeasible in hardware-constrained setups. Consequently, parameter-efficient fine-tuning methods have been developed to enable adaptation of LLMs with limited resources (Houlsby et al., 2019; Li and Liang, 2021; Lester et al., 2021; Hu et al., 2022). One notable work is that of Hu et al. (2022), which performs fine-tuning by learning an additive low-rank update parameterized as the product of two low-rank matrices. Such Low-Rank Adapters (LoRA) have been the subject of study in a large body of work. Various improvements, ranging from initialization strategies (Meng et al., 2024; Li et al., 2025a) and custom learning rate settings for stable feature learning (Hayou et al., 2024b) to alternative parameterizations (Liu et al., 2024a) have been proposed ever since.

Recent work attempts to overcome the low-rank constraint imposed by LoRA while preserving its parameter-efficiency (Xia et al., 2024; Lialin et al., 2024; Zhao et al., 2024; Huang et al., 2025; Jiang et al., 2024). The underlying premise is that the low-rank nature of the adapter limits its expressiveness. However, this premise is at odds with recent theoretical results that LoRA can approximate any target transformer model reasonably well under mild assumptions on the rank that depend on the relationship between the pre-trained and the target model (Zeng and Lee, 2024). Additionally, Kalajdzievski (2023) shows that raising the nominal rank does little to improve performance. Taken together, these findings suggest that the low-rank space offers sufficient expressiveness, but the classical low-rank adapter formulation struggles to fully utilize this potential.

---

[*]Equal supervision.

39th Conference on Neural Information Processing Systems (NeurIPS 2025).

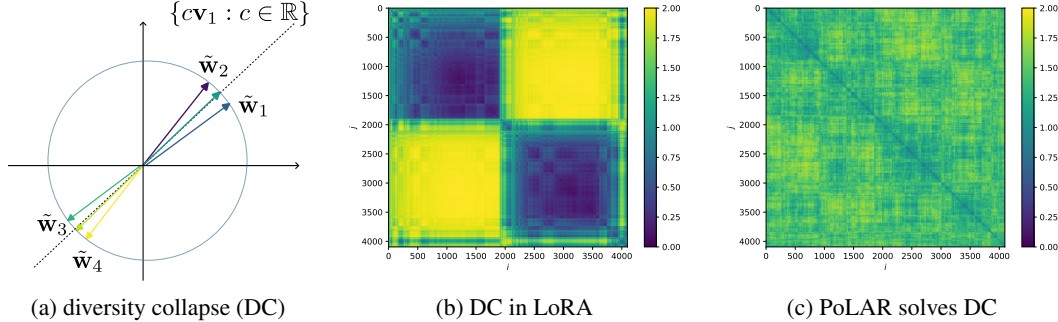

| (a) diversity collapse (DC) | (b) DC in LoRA | (c) PoLAR solves DC |

Figure 1: *(a)* Illustration of directional diversity collapse (DC) of $\tilde{\mathbf{w}}_i = \mathbf{w}_i/\|\mathbf{w}_i\|_2$ where $\mathbf{w}_i$ denotes the $i$-th row of low-rank update $\Delta\mathbf{W}$. *(b) and (c)* Diversity of update directions of LoRA and PoLAR for a Llama-2-7B down-projection layer on dataset Social-IQA, respectively. Each pixel shows $\|\tilde{\mathbf{w}}_i - \tilde{\mathbf{w}}_j\|_2$; rows and columns are rearranged to reveal cluster patterns in both plots. Emergence of a cluster pattern is evidence for DC. The algebraic rank is 32 for both methods, yet the stable rank is 1.06 and 5.49 for LoRA and PoLAR, respectively. See also Section 2.1.

Indeed, we observe comprehensive empirical evidence for this underutilization: when fine-tuning Llama-2-7B with LoRA, we find that the stable rank, a robust analogue of the matrix rank and measure of expressiveness, of the resulting update remains well below the linear algebraic rank. For some learned LoRA updates $\Delta\mathbf{W}$, the stable rank, defined as $\mathrm{sr}(\Delta\mathbf{W}) := \|\Delta\mathbf{W}\|_\mathsf{F}^2/\|\Delta\mathbf{W}\|_2^2$ (Rudelson and Vershynin (2006), see Appendix B), is as low as 1.06. This reveals that approximately a rank 1 subspace is utilized even if the LoRA rank is chosen as 32. Similar behaviors of low stable rank are consistently observed across layers and datasets; see Figs. 2 and 6. Such deficiency results in a *diversity collapse* in the update directions among different neurons, where in extreme cases the updates for all neurons strongly align along a single direction (up to a sign flip); see Fig. 1a for an illustration and Fig. 1b for the directional diversity collapse when fine-tuning Llama-2-7B.

To address this pathology, we put forth PoLAR, a co-design of architecture and optimizer that mitigates the directional diversity collapse, as shown in Fig. 1c. On the *architecture side*, PoLAR facilitates effective exploitation of the linear algebraic rank by expressing the low-rank update as the product of two column-orthogonal direction matrices and a $r \times r$ scale matrix. This parameterization can also be interpreted from the perspective of decoupling direction and magnitude, which arises naturally when the low-rank factors are lifted via the polar decomposition. On the *optimizer side*, we apply methods from Riemannian optimization (Boumal, 2023). Theoretically, we demonstrate that our co-design enables exponentially faster convergence than vanilla LoRA on a canonical problem. To further enable training at scale, we draw on recent advances in infeasible methods for optimization over Stiefel manifolds (Ablin and Peyré, 2022). The resulting method is tailored to GPU hardware, catalyzing a speedup of at least $3\times$ over feasible methods. In a nutshell, our contribution is four-fold:

❖ Our empirical analysis demonstrates that the update matrices learned by LoRA have a stable rank far below their full linear algebraic rank, leading to a collapse in directional diversity and, in turn, preventing the adapters from fully realizing their expressiveness.

❖ We introduce PoLAR, an architecture-optimizer co-design that ensures diverse update directions by factoring the low-rank updates into column-orthogonal direction matrices and an arbitrary scale matrix. Riemannian optimization is then adopted for our PoLAR parameterization.

❖ On a matrix factorization prototype problem, we prove that our PoLAR parameterization achieves an *exponentially* faster convergence rate than vanilla LoRA.

❖ We evaluate PoLAR on three benchmarks (language understanding, commonsense reasoning, and mathematical problem solving) using several backbones: DeBERTa-v3, Llama-2-7B, Gemma-3-12B, and Gemma-3-27B. Consistent performance gains are observed across all settings.

**Notation.** Bold capital (lowercase) letters denote matrices (vectors); $(\cdot)^\top$ and $\|\cdot\|_\mathsf{F}$ refer to transpose and Frobenius norm of a matrix; $\|\cdot\|$ is the $\ell_2$ (spectral) norm of a vector (matrix); $\sigma_i(\cdot)$ and $\lambda_i(\cdot)$ denote the $i$-th largest singular value and eigenvalue, respectively. For a matrix $\mathbf{X} \in \mathbb{R}^{r\times r}$, let $\mathsf{Skew}(\mathbf{X}) = \frac{1}{2}(\mathbf{X} - \mathbf{X}^\top)$ be its skew-symmetric part. The set of matrices with orthonormal columns, i.e., the Stiefel manifold, is denoted as $\mathsf{St}(m, r) := \{\mathbf{X} \in \mathbb{R}^{m\times r} : \mathbf{X}^\top\mathbf{X} = \mathbf{I}_r\}$. The set of $r \times r$ positive semi-definite (PSD) matrices is denoted as $\mathbb{S}_{\succeq 0}^r := \{\mathbf{X} \in \mathbb{R}^{r\times r} : \mathbf{X} \succeq 0\}$.

## 1.1 Related Work

**Memory-Efficient Fine-Tuning.** Low-Rank Adapters (Hu et al., 2022) are now a standard approach for fine-tuning large language models. There are various efforts to further enhance LoRA, e.g., via adaptivity (Zhang et al., 2023), chaining (Lialin et al., 2024; Xia et al., 2024), low-bit training (Dettmers et al., 2023; Li et al., 2024), modifications for long sequences (Chen et al., 2024), weight decomposition (Liu et al., 2024a), or sparsity (Nikdan et al., 2024). For this work, LoRA-variants that change the parameterization of $\Delta\mathbf{W}$ are particularly relevant. DoRA (Liu et al., 2024a) reparameterizes the low-rank update according to weight normalization (Salimans and Kingma, 2016), learning direction and magnitude of the update separately. AdaLoRA (Zhang et al., 2023) allocates the rank of adapters adaptively according to layer importance, learning an SVD-type update with a diagonal matrix and two orthogonal low-rank factors. A work concurrent to ours (Li et al., 2025b) proposes a reparameterization of LoRA that is similar to ours, using feasible methods from the literature of Riemannian optimization for training. VeRA and LoRA-XS propose novel parameterizations to further shrink the number of trainable parameters (Kopiczko et al., 2024; Bałazy et al., 2024). Building on top of the LoRA-XS parameterization, Ponkshe et al. (2024) develop a custom initialization scheme which is informed by the SVD of the gradient at the pre-trained model. Another closely related line of research is dedicated to the development of subspace optimizers which maintain the memory efficiency of low-rank adapters, while enabling full rank parameter updates. Such subspace methods project gradients to a low-rank subspace within which optimizer states are maintained (Zhao et al., 2024; Liang et al., 2024; Hao et al., 2024; Robert et al., 2025; Chen et al., 2025).

**Optimization on Manifolds.** Riemannian optimization generalizes gradient methods to smooth manifolds (Absil et al., 2008; Bonnabel, 2013; Boumal, 2023) and has proven useful in scenarios such as matrix completion (Vandereycken, 2013) and eigenvalue problems (Edelman et al., 1998). Despite the theoretical appeal, such methods are rarely applied in large-scale settings, as expensive retraction operations are required to ensure the feasibility of iterates. Ablin and Peyré (2022) propose an infeasible method for optimization on the orthogonal manifold that does not require retractions. Rather than ensuring the feasibility of all iterates, the specifically designed *landing field* guarantees convergence to the manifold in the limit. Gao et al. (2022) extend their analysis to the Stiefel manifold, and Schechtman et al. (2023) generalize the landing method to more general cases.

## 2 PoLAR: A Co-Design of Architecture and Optimizer

We begin by empirically identifying inefficiencies in LoRA's utilization of its nominal rank, an insight that motivates our PoLAR parameterization based on polar decomposition.

### 2.1 Overcoming Low Stable Rank with PoLAR

Given the pretrained weight (of a linear layer) $\mathbf{W}_0 \in \mathbb{R}^{m \times n}$, LoRA learns an additive low-rank update $\Delta\mathbf{W} \in \mathbb{R}^{m \times n}$ with $\text{rank}(\Delta\mathbf{W}) \leq r$. The adapted weight is thus given by $\mathbf{W}_0 + \Delta\mathbf{W}$. In Hu et al. (2022), the parameterization $\Delta\mathbf{W} = \mathbf{Z}_1\mathbf{Z}_2^\top$ with $\mathbf{Z}_1 \in \mathbb{R}^{m \times r}$ and $\mathbf{Z}_2 \in \mathbb{R}^{n \times r}$ is used.

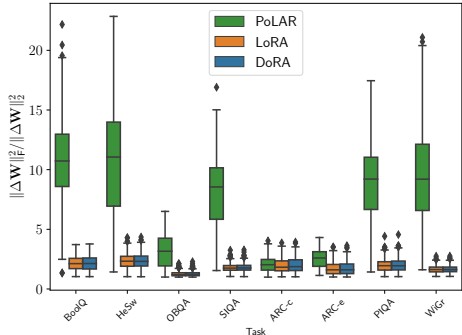

Figure 2: $\text{sr}(\Delta\mathbf{W})$ of Llama-2-7B low-rank updates fine-tuned on commonsense tasks with rank 32.

While LoRA significantly enhances parameter efficiency as $(m + n)r \ll mn$, it turns out that it struggles to fully utilize the expressiveness of its parameterization. In particular, when fine-tuning Llama-2-7B with LoRA, we find that the stable rank $\text{sr}(\Delta\mathbf{W})$ remains small on various datasets, oftentimes approaching 1 for many layers even with a reasonably large choice of $r = 32$; see Fig. 2. Similar observations are also evident in *official* LoRA weights; see more in Fig. 6. Such a low stable rank translates to a loss in directional diversity of the neural updates.

The directional diversity of an update matrix is measured by the average pairwise Euclidean distance of neurons when projected to the unit sphere. Note that this distance is within $[0, 2]$ with the lower and upper bounds attained by a pair of collinear neurons, pointing in the same or opposite directions,

respectively. Consequently, a distribution of pairwise distances with most mass at the ends of the interval can be interpreted as evidence for low directional diversity.

As observed in Fig. 1b, the LoRA update closely aligns along a single direction, with most neural updates being nearly collinear. Using the compact SVD $\Delta\mathbf{W} = \mathbf{U}\boldsymbol{\Sigma}\mathbf{V}^\top$, it is possible to explain this observation and identify a low stable rank as a driver behind the lack of directional diversity. Here, $\mathbf{U} = [\mathbf{u}_1, \ldots, \mathbf{u}_r] \in \mathbb{R}^{m \times r}$ and $\mathbf{V} = [\mathbf{v}_1, \ldots, \mathbf{v}_r] \in \mathbb{R}^{n \times r}$ have orthonormal columns and $\boldsymbol{\Sigma} = \mathrm{diag}(\sigma_1, \ldots, \sigma_r) \in \mathbb{R}^{r \times r}$ contains the singular values on its diagonal. Also, let $\mathbf{w}_i$ denote the $i$-th row of $\Delta\mathbf{W}$. With this notation, we can see that the direction of the LoRA update for the $i$-th neuron can be approximated by:

$$\frac{\mathbf{w}_i}{\|\mathbf{w}_i\|} = \frac{1}{\sqrt{\sum_{j'=1}^r \sigma_{j'}^2 \mathbf{U}_{ij'}^2}} \sum_{j=1}^r \sigma_j \mathbf{U}_{ij} \mathbf{v}_j^\top \overset{(a)}{\approx} \mathrm{sign}(\sigma_1 \mathbf{U}_{i1}) \mathbf{v}_1^\top, \forall i \in \{1, 2, \ldots, m\} \tag{1}$$

where $(a)$ comes from $\mathsf{sr}(\Delta\mathbf{W}) \approx 1$. Equation (1) suggests that the weight update of all neurons tends to strongly align with the direction of the leading right singular vector up to a sign flip, causing the two-cluster pattern in Fig. 1b. Moreover, Fig. 2 demonstrates the wide applicability of this finding, as the LoRA updates across several datasets suffer from a very low stable rank.

We provide more thorough empirical evidence of this collapse in directional diversity in Appendix F.1.

This pathology suggests that LoRA does not fully utilize its rank capacity, and we conjecture that a parameterization addressing this pathology increases the performance of LoRA. These insights lead to the following desiderata: we wish to learn $\mathcal{O}(r)$ roughly orthogonal directions whose contributions to the unit-norm neural update are roughly balanced, avoiding the collapse of directional diversity.

To this end, we advocate to incorporate orthogonality directly into the architecture. This can be done by applying the polar decomposition (Golub and Van Loan, 2013) to each of the low-rank factors.

**Definition 1** (Polar Decomposition). *Let $\mathbf{Z}_1 \in \mathbb{R}^{m \times r}$ with $m \geq r$ and $\mathbf{Z}_2 \in \mathbb{R}^{n \times r}$ with $n \geq r$. It admits factorizations $\mathbf{Z}_1 = \mathbf{X}\boldsymbol{\Theta}_1$ and $\mathbf{Z}_2 = \mathbf{Y}\boldsymbol{\Theta}_2$, where $\mathbf{X} \in \mathsf{St}(m, r)$ [2] and $\mathbf{Y} \in \mathsf{St}(n, r)$ have orthonormal columns and $\boldsymbol{\Theta}_1, \boldsymbol{\Theta}_2 \in \mathbb{R}^{r \times r}$ are positive semi-definite, respectively.*

When $r = 1$, Definition 1 reduces to the familiar magnitude-direction decomposition of vectors.

The polar decomposition thus generalizes magnitude-direction decomposition to matrices, where the semi-orthogonal factor represents the directional component and the positive semi-definite factor captures the magnitude. Using this magnitude-direction decomposition, the desirable orthogonality is naturally enforced through the manifold structure of $\mathbf{X}$ and $\mathbf{Y}$. Moreover, rather than relying on two individual scale matrices, it is more convenient to learn a joint $\boldsymbol{\Theta} \in \mathbb{R}^{r \times r}$ matrix for the overall update, which amounts to merging the product $\boldsymbol{\Theta}_1 \boldsymbol{\Theta}_2^\top$.

These considerations give rise to our **Po**lar-decomposed **L**ow-rank **A**dapter **R**epresentation (PoLAR):

$$\Delta\mathbf{W} = \mathbf{X}\boldsymbol{\Theta}\mathbf{Y}^\top \text{ with } \mathbf{X} \in \mathsf{St}(m, r), \mathbf{Y} \in \mathsf{St}(n, r), \text{ and } \boldsymbol{\Theta} \in \mathbb{R}^{r \times r}. \tag{2}$$

As a byproduct, PoLAR admits a natural interpretation in terms of a direction–magnitude decomposition. However, unlike alternative decompositions, such as the column-wise weight normalization used in DoRA (Salimans and Kingma, 2016; Liu et al., 2024a), PoLAR enforces orthogonality, substantially increasing the stable rank and generating low-rank updates with more competitive downstream task performance (see later in Section 4).

In the following subsection, we show that our PoLAR parameterization in (2) offers provable advantages over LoRA when paired with appropriate optimization algorithms.

## 2.2 Faster Optimization with PoLAR Parameterization

We now compare the convergence of PoLAR and LoRA on the problem of learning a low-rank adapter for a single layer with whitened data. As discussed in Arora et al. (2018); Zhang and Pilanci (2024); Li et al. (2025a) and Appendix C.3, applying LoRA in this case is equivalent to matrix factorization under the asymmetric Burer-Monteiro (BM) parameterization (Burer and Monteiro, 2003)

$$\min_{\mathbf{Z}_1 \in \mathbb{R}^{m \times r}, \mathbf{Z}_2 \in \mathbb{R}^{n \times r}} \frac{1}{2} \|\mathbf{Z}_1 \mathbf{Z}_2^\top - \mathbf{A}\|_\mathsf{F}^2. \tag{3}$$

---

[2] The Stiefel manifold is defined as $\mathsf{St}(m, r) := \{\mathbf{X} \in \mathbb{R}^{m \times r} : \mathbf{X}^\top \mathbf{X} = \mathbf{I}_r\}$.

In problem (3), the matrix to be factorized (or the target matrix of LoRA; see Appendix C.3) is denoted as $\mathbf{A} \in \mathbb{R}^{m \times n}$, and $\mathbf{Z}_1, \mathbf{Z}_2$ represent the LoRA weights. In light of the low-rank setting, $r \ll \min\{m, n\}$ is assumed. Problem (3) has been widely adopted as a testbed for developing advanced optimization schemes for LoRA; see, e.g., Zhang and Pilanci (2024) or Li et al. (2025a).

---

**Algorithm 1** RGD for PoLAR parameterized (4)

---

**Require:** Learning rates $\eta$, $\gamma$; sample $\mathbf{X}_0$ and $\mathbf{Y}_0$ uniformly from $\mathsf{St}(m, r)$ and $\mathsf{St}(n, r)$, respectively.
    **for** $t = 0, \ldots, T - 1$ **do**
        Find $\mathbf{\Theta}_t$ via (5a)
        Obtain Riemannian gradients $\mathbf{E}_t$ and $\mathbf{F}_t$
        Update $\mathbf{X}_{t+1}$ and $\mathbf{Y}_{t+1}$ with (5b) and (5c)
    **end for**

---

Our goal in this subsection is to understand the optimization dynamics of our PoLAR parameterization applied to the same one-layer setting, yielding the problem below

$$\min_{\mathbf{X}, \mathbf{Y}, \mathbf{\Theta}} \frac{1}{2} \|\mathbf{X} \mathbf{\Theta} \mathbf{Y}^\top - \mathbf{A}\|_\mathsf{F}^2, \quad \text{s.t.} \quad \mathbf{X} \in \mathsf{St}(m, r), \ \mathbf{Y} \in \mathsf{St}(n, r), \ \mathbf{\Theta} \in \mathbb{R}^{r \times r}. \tag{4}$$

Considering the sufficient expressiveness of LoRA (Zeng and Lee, 2024), we focus on the overparameterized regime for both (3) and (4), where $r > r_A := \mathsf{rank}(\mathbf{A})$. In this setting, zero loss is attainable. Let the compact SVD of $\mathbf{A}$ be $\mathbf{U}\mathbf{\Sigma}\mathbf{V}^\top$ with $\mathbf{U} \in \mathbb{R}^{m \times r_A}$, $\mathbf{V} \in \mathbb{R}^{n \times r_A}$ and diagonal $\mathbf{\Sigma} \in \mathbb{R}^{r_A \times r_A}$. Without loss of generality, we assume $\sigma_1(\mathbf{\Sigma}) = 1$ and $\sigma_{r_A}(\mathbf{\Sigma}) = 1/\kappa$ where $\kappa$ is the condition number. We also assume $m \geq n$, as one can transpose $\mathbf{A}$ if necessary.

Note that the semi-orthogonal low-rank factors live on Stiefel manifolds, requiring a treatment with Riemannian gradient descent (RGD). For technical simplicity, we consider a procedure that alternates between updating $\mathbf{\Theta}_t$ and $(\mathbf{X}_t, \mathbf{Y}_t)$. At iteration $t$, it starts by finding $\mathbf{\Theta}_t$ with gradient descent (GD) using learning rate $\gamma > 0$, i.e.,

$$\mathbf{\Theta}_t = (1 - \gamma)\mathbf{\Theta}_{t-1} + \gamma \mathbf{X}_t^\top \mathbf{A} \mathbf{Y}_t. \tag{5a}$$

Setting $\gamma = 1$ significantly simplifies our analysis. With this value and the updated matrix $\mathbf{\Theta}_t$, the Riemannian gradient of $\mathbf{X}_t$ can be obtained via $\mathbf{E}_t = -(\mathbf{I}_m - \mathbf{X}_t \mathbf{X}_t^\top)\mathbf{A}\mathbf{Y}_t\mathbf{\Theta}_t^\top$. Further involving polar retraction to remain on the manifold, the RGD update on $\mathbf{X}_t$ is given by

$$\mathbf{X}_{t+1} = (\mathbf{X}_t - \eta\mathbf{E}_t)(\mathbf{I}_r + \eta^2 \mathbf{E}_t^\top \mathbf{E}_t)^{-1/2}. \tag{5b}$$

Likewise, the Riemannian gradient of $\mathbf{Y}_t$ is $\mathbf{F}_t = -(\mathbf{I}_n - \mathbf{Y}_t \mathbf{Y}_t^\top)\mathbf{A}^\top \mathbf{X}_t\mathbf{\Theta}_t$, leading to the update

$$\mathbf{Y}_{t+1} = (\mathbf{Y}_t - \eta\mathbf{F}_t)(\mathbf{I}_r + \eta^2 \mathbf{F}_t^\top \mathbf{F}_t)^{-1/2}. \tag{5c}$$

Alg. 1 summarizes the resulting RGD procedure. Despite the main technical challenge of establishing global convergence in the presence of saddles, RGD for our PoLAR parameterization admits a natural geometric interpretation in terms of principal angles between subspaces. In particular, define the alignment matrices $\mathbf{\Phi}_t := \mathbf{U}^\top \mathbf{X}_t \in \mathbb{R}^{r_A \times r}$ and $\mathbf{\Psi}_t := \mathbf{V}^\top \mathbf{Y}_t \in \mathbb{R}^{r_A \times r}$. The singular values $\{\sigma_i(\mathbf{\Phi}_t)\}_{i=1}^{r_A}$ of $\mathbf{\Phi}_t$ give the arc-cosine of the principal angles between the two subspaces spanned by bases $\mathbf{X}_t$ and $\mathbf{U}$, respectively. In other words, $\{\sigma_i(\mathbf{\Phi}_t)\}_{i=1}^{r_A}$ capture the alignment of the subspaces, yielding $\mathsf{Tr}(\mathbf{\Phi}_t\mathbf{\Phi}_t^\top) = \sum_{i=1}^{r_A} \sigma_i^2(\mathbf{\Phi}_t)$ as a suitable metric to measure total alignment; see more in Appendix C.1. Analogous statements apply to $\{\sigma_i(\mathbf{\Psi}_t)\}_{i=1}^{r_A}$. This alignment perspective further complements the discussion of direction-magnitude decomposition in the context of equation (2). Our first theoretical result demonstrates that the alignment between $\mathbf{X}_t$ and $\mathbf{U}$ is non-decreasing over iterations. This geometric observation bears resemblance to the descent lemma in standard GD.

**Lemma 1** (Increasing Alignment). *Let $\beta_t := \sigma_1(\mathbf{I}_{r_A} - \mathbf{\Phi}_t\mathbf{\Phi}_t^\top)$ and $\delta_t := \sigma_1(\mathbf{I}_{r_A} - \mathbf{\Psi}_t\mathbf{\Psi}_t^\top)$, and suppose that the learning rates are chosen as $\eta < 1$ and $\gamma = 1$. If the following conditions are met,*

$$\frac{2(1 - \eta^2\beta_t)\sigma_{r_A}^2(\mathbf{\Psi}_t)}{\kappa^2}\mathsf{Tr}\big((\mathbf{I}_{r_A} - \mathbf{\Phi}_t\mathbf{\Phi}_t^\top)\mathbf{\Phi}_t\mathbf{\Phi}_t^\top\big) \geq \eta\beta_t\mathsf{Tr}\big(\mathbf{\Phi}_t\mathbf{\Phi}_t^\top\big)$$

$$\frac{2(1 - \eta^2\delta_t)\sigma_{r_A}^2(\mathbf{\Phi}_t)}{\kappa^2}\mathsf{Tr}\big((\mathbf{I}_{r_A} - \mathbf{\Psi}_t\mathbf{\Psi}_t^\top)\mathbf{\Psi}_t\mathbf{\Psi}_t^\top\big) \geq \eta\delta_t\mathsf{Tr}\big(\mathbf{\Psi}_t\mathbf{\Psi}_t^\top\big)$$

*Alg. 1 guarantees that $\mathsf{Tr}(\mathbf{\Phi}_{t+1}\mathbf{\Phi}_{t+1}^\top) \geq \mathsf{Tr}(\mathbf{\Phi}_t\mathbf{\Phi}_t^\top)$ and $\mathsf{Tr}(\mathbf{\Psi}_{t+1}\mathbf{\Psi}_{t+1}^\top) \geq \mathsf{Tr}(\mathbf{\Psi}_t\mathbf{\Psi}_t^\top)$.*

Next, we show that, with a proper $\eta$, Lemma 1 leads to global convergence of Alg. 1.

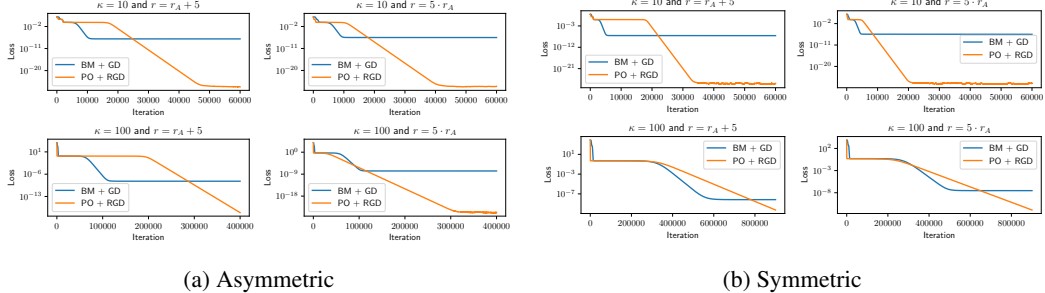

(a) Asymmetric             (b) Symmetric

Figure 3: Overparameterized matrix factorization for different condition numbers and degrees of overparameterization using the PoLAR parameterization with Riemannian Gradient Descent (PO + RGD) and the Burer-Monteiro parameterization with Gradient Descent (BM + GD). *(a)* Asymmetric scenario and *(b)* symmetric case. The experimental details can be found in Appendix G.4.

**Theorem 1** (Global Convergence). *Suppose that $r_A \leq \frac{n}{2}$. Let $\rho := \min\{\frac{1}{m}, \frac{(r-r_A)^2}{rm}\}$. W.h.p. over the random initialization of $\mathbf{X}_0$ and $\mathbf{Y}_0$, choosing the learning rates $\eta = \mathcal{O}(\frac{(r-r_A)^2\rho}{r^2\kappa^2 m})$ and $\gamma = 1$, Alg. 1 ensures $\frac{1}{2}\|\mathbf{X}_T\mathbf{\Theta}_T\mathbf{Y}_T^\top - \mathbf{A}\|_{\mathsf{F}}^2 \leq \epsilon$ for all $T \geq \mathcal{O}\big(\frac{m^2 r^3 r_A \kappa^4}{\rho^2(r-r_A)^4} + \frac{m^2 r^3 \kappa^4}{\rho(r-r_A)^4}\log\frac{1}{\epsilon}\big)$.*

Theorem 1 builds on a three-phase analysis to prove global convergence of Alg. 1. It demonstrates that RGD with PoLAR parameterization avoids getting trapped by saddles of (4) without the need for artificial noise. Our rate compares favorably to previous results of GD in the overparameterized regime.[3] In particular, Xiong et al. (2024) show that overparameterization slows down GD, leading to an undesirable $\kappa$ dependence with $\mathcal{O}\big(\max\{\kappa^{15}, \kappa^{\kappa}\}\log(1/\epsilon)\big)$. Although this bound is originally obtained for matrix sensing, the numerical example in Fig. 3 confirms that GD barely progresses beyond certain iterations. Our rates in Theorem 1 exponentially improve the $\kappa$ dependence to a quartic one. Compared with other works of overparameterized GD on BM such as Soltanolkotabi et al. (2023), our bound not only has a more favorable dependence on $\kappa$, but also ensures last-iteration convergence, removing the need for early stopping.

It is also worth noting that, unlike GD with BM, PoLAR *improves* with overparameterization. To see this, consider the slightly overparameterized regime with $r \approx r_A$, e.g., $r = r_A + 5$. Our bound in Theorem 1 reads $\mathcal{O}\big(m^4 r^6 \kappa^4 + m^3 r^4 \kappa^4 \log(1/\epsilon)\big)$. However, once at the highly overparameterized regime $r = Cr_A$ for some constant $C$, e.g., $C = 5$, we have a bound with a drastically improved dependence on the dimension: $\mathcal{O}\big(m^4\kappa^4 + \frac{m^3\kappa^4}{r_A}\log(1/\epsilon)\big)$. A graphical illustration can be found in Fig. 3a, where the convergence of $r = 5r_A$ is much faster than that of $r = r_A + 5$.

Theorem 1 also suggests practical merits of PoLAR, notably its *simplified initialization* (via, for instance, Lemma 6). In contrast, initialization of BM can impose multiple requirements both theoretically and empirically (Ye and Du, 2021; Xiong et al., 2024; Hayou et al., 2024b). In addition, our choice of an *unconstrained* $\mathbf{\Theta} \in \mathbb{R}^{r\times r}$ plays a crucial role for convergence. Empirical evidence in (Mishra et al., 2013, Fig. 5) shows that substituting $\mathbf{\Theta}$ by a diagonal matrix $\mathbf{\Theta}^{\mathsf{d}}$, as done in AdaLoRA (Zhang et al., 2023), can adversely affect convergence, potentially because of the presence of non-strict saddles in the loss landscape (Levin et al., 2024). These spurious stationary points can be removed by a parameterization with positive-definite $\mathbf{\Theta}^{\mathsf{s}}$ (Levin et al., 2024, Prop. 2.10). Our PoLAR parameterization poses no constraints on $\mathbf{\Theta}$. It thus avoids computational overheads associated with enforcing positive-definiteness (e.g., matrix exponentials), yet still ensures global convergence.

Lastly, we extend our analysis to the symmetric matrix factorization problem in Appendix E, where we prove last-iterate convergence with complexity $\mathcal{O}\big(\kappa^4 \log(1/\epsilon)\big)$. This represents an *exponential* improvement in the $\epsilon$-dependence over GD on BM-type problems, which is subject to the lower bound of $\Omega(\kappa^2/\sqrt{\epsilon})$ established in Xiong et al. (2024); see also Fig. 3b for an illustration.

---

[3]The works of (Ward and Kolda, 2023; Xu et al., 2024b) provide improved $\kappa$ dependence of Alternating GD and GD when the column/row space of $\mathbf{A}$ is known at initialization. They are excluded from comparison because knowing the column space of $\mathbf{A}$ amounts to setting $\mathsf{Tr}(\mathbf{\Phi}_0\mathbf{\Phi}_0^\top) = r_A$, which is already optimal at initialization.

# 3  Practical PoLAR for Scalable Fine-Tuning

Although PoLAR increases expressiveness and accelerates convergence, optimizing under the manifold constraint of (2) with standard feasible methods does not scale to tasks such as LLM fine-tuning. Every retraction back onto the Stiefel manifold requires a matrix inversion or SVD (see (5b)), operations whose sequential nature limits GPU parallelism and becomes a runtime bottleneck (see Table 4 and Sun et al. (2024)). Retractions are also numerically brittle in the low-precision arithmetic, now standard in training. To sidestep these issues, we take inspiration from the recently proposed *landing algorithm* which completely eschews retractions, producing iterates that are not necessarily on the manifold, but provably *land* on it as training proceeds (Ablin and Peyré, 2022; Gao et al., 2022).

To fine-tune the PoLAR parameterization of (2), we initialize $\mathbf{X}_0$ and $\mathbf{Y}_0$ randomly on the Stiefel manifold and set $\mathbf{\Theta}_0 = \mathbf{0}$. At iteration $t$, we update $\mathbf{X}_t$ and $\mathbf{Y}_t$ with the landing field. Taking the update on $\mathbf{X}_t$ as an example, we use

$$\mathbf{\Gamma}(\mathbf{X}_t) := \underbrace{\psi(\mathbf{X}_t)\mathbf{X}_t}_{\text{Riemannian grad.}} + \underbrace{\lambda\nabla\mathcal{N}(\mathbf{X}_t)}_{\text{Infeasibility penalty}} \quad (6)$$

as a drop-in replacement for the Riemannian update described in Section 2.2. The first component in (6) is the standard Riemannian gradient for matrices on the Stiefel manifold with $\psi(\mathbf{X}) := \mathsf{Skew}(\nabla_{\mathbf{X}}\mathcal{L}(\mathbf{X}, \mathbf{\Theta}, \mathbf{Y})\mathbf{X}^\top)$. The second component in (6) is given by the gradient of the infeasibility penalty, $\mathcal{N}(\mathbf{X}) := \|\mathbf{X}^\top\mathbf{X} - \mathbf{I}_r\|_{\mathsf{F}}^2$, which attracts the iterate towards the Stiefel manifold, making retraction obsolete. The landing field (6) has several appealing properties. First, the two components of the field are orthogonal to each other, decoupling loss minimization and convergence to the Stiefel manifold. Second, its computation involves only matrix multiplications, eliminating the need for sequential retraction routines that do not map well to GPU parallelism; see Section 4.4 for numerical comparisons.

---

**Algorithm 2** PoLAR Fine-tuning

**Require:** Parameterize via (2); Initialize $\mathbf{X}_0, \mathbf{Y}_0$ uniformly random from Stiefel manifolds, set $\mathbf{\Theta}_0 = \mathbf{0}$, and denote $\lambda$ regularization strength, $\rho_t$ stateful gradient transformation (e.g., Adam)
 **for** $t = 0, \ldots, T-1$ **do**
  $\mathbf{\Gamma}(\mathbf{X}_t) \leftarrow \psi(\mathbf{X}_t)\mathbf{X}_t + \lambda\nabla\mathcal{N}(\mathbf{X}_t)$
  $\mathbf{\Gamma}(\mathbf{Y}_t) \leftarrow \psi(\mathbf{Y}_t)\mathbf{Y}_t + \lambda\nabla\mathcal{N}(\mathbf{Y}_t)$
  $\mathbf{X}_{t+1} \leftarrow \mathbf{X}_t - \eta_t\rho_t(\mathbf{\Gamma}(\mathbf{X}_t))$
  $\mathbf{Y}_{t+1} \leftarrow \mathbf{Y}_t - \eta_t\rho_t(\mathbf{\Gamma}(\mathbf{Y}_t))$
  $\mathbf{\Theta}_{t+1} \leftarrow \mathbf{\Theta}_t - \eta_t\rho_t(\nabla_{\mathbf{\Theta}_t}\mathcal{L}(\mathbf{X}_t, \mathbf{\Theta}_t, \mathbf{Y}_t))$
 **end for**

---

The complete PoLAR fine-tuning procedure is summarized in Alg. 2. We treat $\lambda > 0$ in (6) as a tunable hyperparameter chosen such that iterates converge to the Stiefel manifold in the limit (see Fig. 8). Alg. 2 also omits the alternating update for $\mathbf{\Theta}$ used in Alg. 1 to simplify training by avoiding a second backward pass. Additional details relating to deployment appear in Appendix A.

# 4  Experiments

To demonstrate PoLAR's effectiveness at scale, we fine-tune backbones spanning 350M–27B parameters on three benchmarks of varying difficulty. Our study includes both masked and autoregressive language models and evaluates their abilities in terms of general language understanding, commonsense reasoning, and multi-step mathematical problem solving. The code for our experiments is available at `https://github.com/kcc-lion/polar/`.

## 4.1  Commonsense Reasoning

Commonsense reasoning tasks aim to assess how well LLMs can mimic human-like understanding. A standard suite of eight different datasets is chosen, and LLMs are finetuned separately on each of the datasets. For evaluation, we employ the widely-adopted *lm-evaluation-harness* framework from Eleuther-AI (Biderman et al., 2024). We report the accuracy based on multiple-choice log-likelihood evaluation to facilitate reproducibility. More details can be found in Appendix G.1.

We first isolate the contribution of our $\mathbf{\Theta}$ parameterization and the optimizer based on the landing field. To this end, we compare PoLAR with a modified procedure which (i) sets $\mathbf{\Theta} \in \mathrm{Diag}(r)$ and (ii) replaces the Riemannian gradient with the Euclidean gradient. These changes bring our procedure closer to the SVD-type parameterization discussed in AdaLoRA (Zhang et al., 2023). Gemma-2-2B is chosen as the base model (Gemma Team, 2024), and adapters with diverse ranks are tested. We report the results in Table 1 and remark that PoLAR outperforms the diagonal parameterization across all ranks, validating the effectiveness of our co-design.

Table 1: Accuracy of Gemma-2-2B with PoLAR on commonsense reasoning tasks for different gradient types and parameterizations. Rie. (Eucl.) refers to Riemannian (Euclidean) gradient.

| Rank | Param. $\Theta$ | Grad. | BoolQ | PIQA | SIQA | HeSw | WiGr | ARC-e | ARC-c | OBQA | **Avg.** |
|------|------|------|------|------|------|------|------|------|------|------|------|
| 4 | $\mathrm{Diag}(r)$ | Eucl. | 86.24 | 81.50 | 58.70 | 79.88 | 78.69 | 78.75 | 52.39 | 56.80 | 71.62 |
|  | $\mathbb{R}^{r \times r}$ | Rie. | 86.48 | 81.66 | 58.90 | 79.69 | 80.03 | 81.78 | 54.69 | 56.40 | **72.45** |
| 8 | $\mathrm{Diag}(r)$ | Eucl. | 86.06 | 82.05 | 59.52 | 80.38 | 77.19 | 79.50 | 55.03 | 57.20 | 72.12 |
|  | $\mathbb{R}^{r \times r}$ | Rie. | 86.94 | 81.77 | 58.80 | 80.49 | 79.64 | 81.94 | 56.14 | 55.80 | **72.69** |
| 16 | $\mathrm{Diag}(r)$ | Eucl. | 86.82 | 81.83 | 59.52 | 81.08 | 78.37 | 78.87 | 54.35 | 54.80 | 71.96 |
|  | $\mathbb{R}^{r \times r}$ | Rie. | 86.91 | 81.18 | 59.93 | 81.01 | 79.16 | 82.45 | 53.92 | 57.40 | **72.75** |
| 32 | $\mathrm{Diag}(r)$ | Eucl. | 87.03 | 81.39 | 60.08 | 81.73 | 77.51 | 79.21 | 55.29 | 56.60 | 72.35 |
|  | $\mathbb{R}^{r \times r}$ | Rie. | 87.28 | 81.61 | 59.72 | 81.40 | 77.74 | 81.78 | 54.86 | 57.80 | **72.77** |

Table 2: Performance on commonsense reasoning tasks with Llama-2-7B using PoLAR for different ranks in a single-task setup. HeSw refers to HellaSwag and WiGr to WinoGrande.

| Rank | Adapter | BoolQ | PIQA | SIQA | HeSw | WiGr | ARC-e | ARC-c | OBQA | **Avg.** |
|------|------|------|------|------|------|------|------|------|------|------|------|
| 4 | LoRA | 87.16 | 81.01 | 58.85 | 82.36 | 74.35 | 81.90 | 57.68 | 56.80 | 72.51 |
|  | DoRA | 87.22 | 80.30 | 58.96 | 82.39 | 75.22 | 81.69 | 57.85 | 56.80 | 72.55 |
|  | **PoLAR** | 87.49 | 82.59 | 59.31 | 81.23 | 81.77 | 81.61 | 56.31 | 55.80 | **73.26** |
| 8 | LoRA | 87.34 | 81.07 | 58.80 | 82.68 | 74.66 | 82.24 | 58.11 | 55.80 | 72.59 |
|  | DoRA | 87.49 | 80.58 | 58.39 | 82.32 | 75.22 | 82.24 | 58.19 | 55.60 | 72.50 |
|  | **PoLAR** | 87.74 | 82.70 | 59.62 | 82.14 | 82.40 | 81.36 | 56.91 | 55.20 | **73.51** |
| 16 | LoRA | 86.94 | 81.18 | 58.96 | 82.57 | 76.01 | 82.58 | 57.85 | 56.00 | 72.76 |
|  | DoRA | 86.85 | 81.34 | 58.29 | 82.39 | 74.59 | 82.62 | 57.68 | 55.40 | 72.39 |
|  | **PoLAR** | 88.01 | 82.92 | 59.93 | 82.68 | 81.77 | 81.82 | 57.51 | 57.40 | **74.00** |
| 32 | LoRA | 87.89 | 81.56 | 59.06 | 82.51 | 72.61 | 82.37 | 56.83 | 54.60 | 72.18 |
|  | DoRA | 87.61 | 81.45 | 58.70 | 82.50 | 74.43 | 82.28 | 57.17 | 55.60 | 72.47 |
|  | **PoLAR** | 88.13 | 82.64 | 60.03 | 83.12 | 82.00 | 81.99 | 56.14 | 55.60 | **73.71** |

Scaling up to Llama-2-7B (Touvron et al., 2023), PoLAR again delivers the highest mean accuracy across tasks, outperforming both LoRA and DoRA (Table 2). Whereas the gains of DoRA and LoRA appear to be mostly flat as the adapter rank grows, PoLAR's accuracy increases with larger rank. This is consistent with our conjecture that PoLAR counteracts directional-diversity collapse by exploiting the allocated rank more effectively. We next examine the stable rank of the learned update matrices (Fig. 2) at nominal rank 32. For all datasets, except for ARC-e and ARC-c, PoLAR significantly increases the stable rank of the update. Correspondingly, these two tasks are also where PoLAR shows no accuracy advantage relative to LoRA and DoRA. These results support the intuition that a higher stable rank correlates with improved downstream performance. Figs. 4a and 4b (and more in Appendix F.1) extend the analysis over multiple nominal (linear algebraic) ranks: LoRA's stable rank rarely exceeds 2, while PoLAR's increases steadily with the nominal rank, indicating a richer subspace and mirroring the observed accuracy gains. Figs. 4c and 4d further visualize the evolution of the stable rank over the course of training. While the stable rank for LoRA remains relatively constant, PoLAR's stable rank increases throughout training and appears to dynamically vary with the layer type: MLP layers (up and down projection) appear to require a larger stable rank, while that of the value projection layer often comes out lowest (see more examples in Fig. 7).

In Appendix F, we provide further experimental results for the commonsense reasoning benchmark, comparing PoLAR to GaLore (Zhao et al., 2024). In addition, we further scale up the experiment and report detailed results for Gemma-3-12B (Gemma Team, 2025), where PoLAR is observed to improve upon LoRA by more than a percentage point.

## 4.2 Mathematical Problem Solving

Our second set of experiments tests the ability of fine-tuned models to perform multi-step mathematical reasoning. To do so, we apply PoLAR to adapt Gemma-3-27B on MetaMathQA (Yu et al., 2024) and evaluate on GSM8K (Cobbe et al., 2021), a set of 8.5K grade school math word problems,

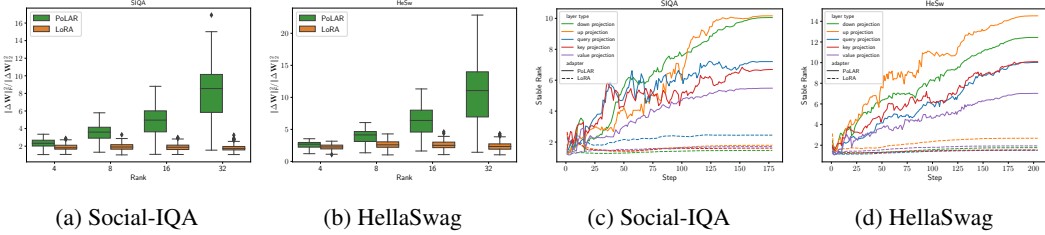

| (a) Social-IQA | (b) HellaSwag | (c) Social-IQA | (d) HellaSwag |

Figure 4: *Left two:* Stable rank distribution of Llama-2-7B adapter layers trained with PoLAR and LoRA and different $r$ on the Social-IQA and HellaSwag datasets. See also Fig. 5. *Right two:* Stable rank dynamics of the fifth transformer block over the course of training by layer type. See also Fig. 7.

Table 3: Performance comparison on GSM8K and MATH. ‡ results are taken from Yu et al. (2024).

| Model | Method | Trainable (%) | GSM8K | MATH |
|---|---|---|---|---|
| GPT-4‡ | - | - | 92.00 | 42.50 |
| Gemma-3-27B-PT | LoRA | 0.2816 | 85.37 | 41.94 |
| | **PoLAR** | 0.2818 | **85.67** | **42.70** |

Table 4: Runtime ($\mu$s) of landing (Land.) and retraction (Retr.) on a H100 GPU with parameter size $(4096, r)$

| $r$ | 4 | 32 | 64 | 256 |
|---|---|---|---|---|
| Retr. | 541 | 1050 | 1916 | 9889 |
| **Land.** | **180** | **186** | **238** | **539** |

and MATH (Hendrycks et al., 2021), a set of 12.5K high school competition mathematics problems. As is done in Yu et al. (2024), a zero-shot prompt is leveraged for evaluation. We report results in Table 3 and remark that PoLAR achieves superior performance compared to LoRA with the margin being larger on the more challenging MATH benchmark. In Appendix F.3, we show that approximate feasibility is maintained throughout training with the manifold constraints fulfilled at convergence.

## 4.3 Natural Language Understanding

Lastly, we demonstrate the applicability of our method beyond auto-regressive language models. To achieve this, we use DeBERTa-V3 (He et al., 2021) as a representative of masked language models (Devlin et al., 2019) and fine-tune it on the GLUE benchmark (Wang et al., 2019). We repeat the experiment with four seeds and report the evaluation results in Table 5 with the standard deviation in the subscript. Comparing PoLAR with LoRA, we note significant improvements across all tasks except for QQP and MRPC where results are comparable. These results highlight that PoLAR can handle both autoregressive as well as masked language model setups.

## 4.4 Practical Runtime Improvement over Retraction

For $\tilde{\mathbf{X}} \in \mathbb{R}^{m \times r}$, a landing step requires $\mathcal{O}(m^2 r)$ FLOPs while a retraction-based routine only requires $\mathcal{O}(mr^2)$ operations. While landing offers no advantage in terms of FLOPs over retraction-based methods, the General Matrix Multiplications (GEMMs) used in a landing step achieve much higher FLOP utilization on GPU hardware than the SVD-based retraction routine. To quantify runtime differences, we benchmark the retraction and landing operations using the most frequently occurring attention and MLP weight dimensions in Llama-2-7B LoRA fine-tuning. As Table 4 shows, even for very small ranks of $r = 4$, which make retraction relatively cheaper, usage of the landing method provides roughly a speedup of $3\times$ with the advantage increasing to $\approx 18\times$ for higher ranks. Appendix F.2 extends the study to a broader range of layer shapes and additionally compares memory and runtime overhead to LoRA and DoRA.

Table 5: Performance of PoLAR on GLUE with DeBERTa-V3-base. We report the matched accuracy for MNLI, Matthew's (Pearson) correlation for CoLA (STS-B), and accuracy for other tasks.

| Name | MNLI | SST-2 | MRPC | CoLA | QNLI | QQP | RTE | STS-B | **Avg.** |
|---|---|---|---|---|---|---|---|---|---|
| LoRA | $90.01_{\pm 0.08}$ | $95.18_{\pm 0.52}$ | $89.71_{\pm 1.34}$ | $68.17_{\pm 1.07}$ | $94.06_{\pm 0.16}$ | $92.47_{\pm 0.03}$ | $84.84_{\pm 1.22}$ | $90.99_{\pm 0.15}$ | 88.18 |
| **PoLAR** | $90.67_{\pm 0.12}$ | $95.96_{\pm 0.22}$ | $89.52_{\pm 0.51}$ | $69.89_{\pm 0.47}$ | $94.37_{\pm 0.13}$ | $92.12_{\pm 0.05}$ | $87.18_{\pm 1.33}$ | $91.83_{\pm 0.06}$ | **88.94** |

## 5 Conclusion and Future Work

Low-rank adaptation has become the workhorse for efficient fine-tuning, yet our analysis revealed that the classical BM parameterization in LoRA often underutilizes its allocated subspace: the stable rank of the learned updates collapses, limiting expressive power. Building on this empirical finding, as well as theoretical insights from a canonical low-rank optimization problem, we introduced PoLAR, a reparameterization inspired by the polar decomposition that is coupled with a landing field optimization procedure. Across backbones from 350M to 27B parameters and tasks that span general language understanding, commonsense reasoning, and multi-step mathematics, PoLAR delivered consistent accuracy gains. The improvements often scale with adapter rank, validating that maintaining a higher stable rank translates into richer, more task-aligned updates. In practice, PoLAR's reliance on nothing more than matrix multiplications implies that it maps cleanly onto GPU hardware, offering a drop-in replacement for existing LoRA pipelines.

Given the low stable rank of the LoRA update, a closer examination of the spectral dynamics of low-rank fine-tuning seems to be a promising direction for future work. While several works already study the spectral evolution of weights during ordinary training of deep neural networks (Boix-Adsera et al., 2023; Martin and Mahoney, 2021; Mousavi-Hosseini et al., 2023), the impact of the explicit low-rank constraint on the spectrum is not yet well understood. An investigation into the dynamics of the spectral distributions of PoLAR could bring more insights into the benefits of the parameterization.

## Acknowledgements

We are grateful to Christopher Criscitiello for insightful discussions and thank the anonymous reviewers for helpful feedback. Kai Lion is supported by Swiss National Science Foundation (SNSF) Sinergia Funding No. 216600. Liang Zhang gratefully acknowledges funding by the Max Planck ETH Center for Learning Systems (CLS). Bingcong Li is supported by SNSF Project Funding No. 200021-207343. Niao He is supported by ETH research grant funded through ETH Zurich Foundations and SNSF Starting Grant. This work was supported as part of the Swiss AI Initiative by a grant from the Swiss National Supercomputing Centre (CSCS) under project ID a105 on Alps.

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

# Supplementary Document for PoLAR

# Contents

# A Limitations, Broader Impact, and Practical Concerns

**Limitations.** In our empirical evaluation of PoLAR we exclusively focused on natural language processing tasks using transformer architectures. We did not evaluate our method on other sequence modeling architectures (e.g., RNNs) or computer vision tasks. Also, we did not cover model sizes beyond 27B. Due to computational cost associated with fine-tuning decently-sized models with 1B+ parameters, we repeat each dataset-model combination only for a single seed. For DeBERTa-v3-base, we repeat training with four different seeds.

**Broader Impact.** The theories and approaches developed are applicable in discriminative and generative use-cases. The proposed algorithmic tool improves fine-tuning performance on discriminative tasks such as sequence classification. Use-cases range from recommender systems to content-moderation pipelines where reliable classification is crucial. For generative downstream tasks, model output should be audited through automated gating or filtering mechanisms.

**Practical Concerns.** In the scenario of serving multiple adapters concurrently (Sheng et al., 2024) PoLAR and LoRA share identical storage and runtime costs at inference, as $\boldsymbol{\Theta}_T$ can be merged into either $\mathbf{X}_T$ or $\mathbf{Y}_T$ once fine-tuning is complete. We note that PoLAR incurs computational overhead during fine-tuning compared to LoRA, due to the landing field computation and the additional $r \times r$ matrix $\boldsymbol{\Theta}_t$. As already mentioned, this runtime overhead compares favorably to a classical Riemannian treatment based on retractions (see Section 4.4). Moreover, we observe runtime benefits over other popular LoRA variants such as DoRA (see Table 6).

# B Additional Related Work

**Parameter-Efficient Fine-Tuning.** The goal of PEFT is to reduce the resource requirement for finetuning LLMs on downstream tasks. Some commonly employed methods include adapters (Houlsby et al., 2019) and prefix tuning (Li and Liang, 2021). The family of LoRA adapters (Hu et al., 2022) recently gained significant popularity. Beyond the works already mentioned in Section 1.1, there are several lines of work focused on improving or understanding different aspects of LoRA. Zhu et al. (2024); Hayou et al. (2024a) contribute to the understanding of the different roles of the low-rank factors. Gu et al. (2023) extend the framework to diffusion models. Xu et al. (2024a); Dettmers et al. (2023) improve memory-efficiency through quantization strategies. Liu et al. (2022) further improve parameter efficiency. Zhang et al. (2024) apply landing methods to LoRA fine-tuning and present LoRA variants that constrain the up-projection to the Oblique and Stiefel manifold. Scale and rotation invariance is considered for efficiently optimizing vanilla LoRA parameterization (Yen et al., 2025; Zhang et al., 2025).

**Optimization for Matrix Factorization.** Existing works mostly handle matrix factorization problems under the BM parameterization, i.e., (3). Such problems are classical examples for understanding the optimization dynamics of LoRA. They involve a complex landscape characterized by nonconvexity, saddle points, and the absence of the PL condition.

Recent works have examined the convergence of several algorithms under the BM parameterization, such as GD, Alternating GD, and ScaledGD (Du et al., 2018; Ward and Kolda, 2023; Li et al., 2025a) in overparameterized settings. Another closely related problem is matrix sensing. The convergence behaviors of different algorithms under the BM parameterization can be found in e.g., (Stöger and Soltanolkotabi, 2021; Zhang et al., 2021; Jin et al., 2023; Xiong et al., 2024).

**Stable Rank.** The stable rank (Rudelson and Vershynin, 2006) is a continuous and perturbation-robust analogue of the classical linear algebraic rank (Ipsen and Saibaba, 2024). It is given by the square of the ratio of the Frobenius and spectral norms. In the context of regularization for convolutional neural networks, constraining the stable rank has been found to curb expressivity, mitigate overfitting, and reduce memorization (Sanyal et al., 2020; Georgiev et al., 2021).

**Other Related Work.** Qiu et al. (2023); Liu et al. (2024b) use a multiplicative orthogonal matrix to fine-tune diffusion models, attempting to preserve the angular relationships between neurons of the pre-trained matrix. This is motivated through the preservation of hyperspherical energy of a weight matrix. Hyperspherical energy measures the distribution of neurons on the unit-norm hypersphere and attains its minimum when neurons are distributed uniformly across the sphere (Liu et al., 2018;

Lin et al., 2020). A low stable rank of the LoRA update can be understood in terms of a state of high hyperspherical energy of the low-rank update.

## C Useful Facts

### C.1 Angles Between Subspaces

Angles between two subspaces are known as principal angles. Suppose that $n \geq p$ and $n \geq q$, and let $\mathcal{U}$ and $\mathcal{V}$ be two linear subspaces of dimension $n \times p$ and $n \times q$. Then the principle angles $\theta_i \in [0, \frac{\pi}{2}], \forall i \leq \min\{p, q\}$ is defined as

$$\theta_1 = \max \left\{ \arccos \frac{\langle \mathbf{u}, \mathbf{v} \rangle}{\|\mathbf{u}\|\|\mathbf{v}\|} \Big| \mathbf{u} \in \mathcal{U}, \mathbf{v} \in \mathcal{V} \right\}$$

$$\theta_i = \max \left\{ \arccos \frac{\langle \mathbf{u}, \mathbf{v} \rangle}{\|\mathbf{u}\|\|\mathbf{v}\|} \Big| \mathbf{u} \in \mathcal{U}, \mathbf{v} \in \mathcal{V}, \mathbf{u} \perp \mathbf{u}_j, \mathbf{v} \perp \mathbf{v}_j, \forall j \in 1, \dots, i-1 \right\}, \forall i \neq 1.$$

There is a well-known relation between principal angles and SVD. Let $\mathbf{U}$ and $\mathbf{V}$ be basis of $\mathcal{U}$ and $\mathcal{V}$ respectively. It can be seen that all the singular values of $\mathbf{U}^\top \mathbf{V}$ belong to $[0, 1]$. Moreover, the principle angles defined above are just the arc-cosine of these singular values (Björck and Golub, 1973). For convenience of this work, *we refer to the singular values of $\mathbf{U}^\top \mathbf{V}$ as principal angles (instead of the arc-cosine of them)*. If the basis $\mathbf{U}$ and $\mathbf{V}$ are both from $\mathsf{St}(m, r)$, we sometimes use the term "alignment", where we say $\mathbf{U}$ and $\mathbf{V}$ are aligned if all the singular values of $\mathbf{U}^\top \mathbf{V}$ are 1; or in other words, they share the same column space.

The principal angles are also related to the geodesic distance on Grassmann manifolds. Oftentimes, people use the term *chordal distance* to refer to $d(\mathbf{U}, \mathbf{V}) = \sqrt{\sum_i \sin^2 \theta_i}$, where $\theta_i$ are principal angles between two subspaces spanned by $\mathbf{U}$ and $\mathbf{V}$. The square of the chordal distance coincides with our notation $\mathsf{Tr}(\mathbf{I} - \mathbf{\Phi}_t \mathbf{\Phi}_t^\top)$, where $\mathbf{\Phi}_t$ is defined in Section 2.2.

### C.2 Other Useful Lemmas

**Lemma 2.** *Given a PSD matrix $\mathbf{A}$, we have that $(\mathbf{I} + \mathbf{A})^{-1} \succeq \mathbf{I} - \mathbf{A}$.*

*Proof.* Simply diagonalizing the LHS and RHS, and using $1/(1 + \lambda) \geq 1 - \lambda, \forall \lambda \geq 0$ gives the result. □

**Lemma 3.** *Suppose that $\mathbf{X} \in \mathsf{St}(m, r)$, $\mathbf{U} \in \mathsf{St}(m, r_A)$ and $r_A \leq r$. Let $\mathbf{U}_\perp \in \mathbb{R}^{m \times (m - r_A)}$ be the orthogonal complement of $\mathbf{U}$. Denote $\mathbf{\Phi} = \mathbf{U}^\top \mathbf{X}$ and $\mathbf{\Omega} = \mathbf{U}_\perp^\top \mathbf{X}$. It is guaranteed that $\sigma_i^2(\mathbf{\Phi}) + \sigma_i^2(\mathbf{\Omega}) = 1$ holds for $i \in \{1, 2, \dots, r\}$.*

*Proof.* We have that

$$\mathbf{I}_r = \mathbf{X}^\top \mathbf{X} = \mathbf{X}^\top \mathbf{I}_m \mathbf{X} = \mathbf{X}^\top [\mathbf{U}, \mathbf{U}_\perp] \begin{bmatrix} \mathbf{U}^\top \\ \mathbf{U}_\perp^\top \end{bmatrix} \mathbf{X} \tag{7}$$

$$= \mathbf{\Phi}^\top \mathbf{\Phi} + \mathbf{\Omega}^\top \mathbf{\Omega}.$$

Equation (7) indicates that $\mathbf{\Psi}^\top \mathbf{\Psi}$ and $\mathbf{\Phi}^\top \mathbf{\Phi}$ commute, i.e.,

$$(\mathbf{\Phi}^\top \mathbf{\Phi})(\mathbf{\Omega}^\top \mathbf{\Omega}) = (\mathbf{\Phi}^\top \mathbf{\Phi})(\mathbf{I}_r - \mathbf{\Phi}^\top \mathbf{\Phi}) = \mathbf{\Phi}^\top \mathbf{\Phi} - \mathbf{\Phi}^\top \mathbf{\Phi} \mathbf{\Phi}^\top \mathbf{\Phi}$$

$$= (\mathbf{I}_r - \mathbf{\Phi}^\top \mathbf{\Phi})(\mathbf{\Phi}^\top \mathbf{\Phi}) = (\mathbf{\Omega}^\top \mathbf{\Omega})(\mathbf{\Phi}^\top \mathbf{\Phi}).$$

The commutativity shows that the eigenspaces of $\mathbf{\Phi}^\top \mathbf{\Phi}$ and $\mathbf{\Omega}^\top \mathbf{\Omega}$ coincide. As a result, we have again from (7) that $\sigma_i^2(\mathbf{\Phi}) + \sigma_i^2(\mathbf{\Omega}) = 1$ for $i \in \{1, 2, \dots, r\}$. The proof is thus completed. □

**Lemma 4.** *Suppose that $\mathbf{P}$ and $\mathbf{Q}$ are $m \times m$ diagonal matrices, and their diagonal entries are non-negative. Let $\mathbf{S}$ be a PD matrix of $m \times m$ with smallest eigenvalue $\lambda_{\min}$, then we have that*

$$\mathsf{Tr}(\mathbf{PSQ}) \geq \lambda_{\min} \mathsf{Tr}(\mathbf{PQ}).$$

*Proof.* Let $p_i$ and $q_i$ be the $(i,i)$-th entry of $\mathbf{P}$ and $\mathbf{Q}$, respectively. Then we have that

$$\mathsf{Tr}(\mathbf{PSQ}) = \sum_i p_i \mathbf{S}_{i,i} q_i \geq \lambda_{\min} \sum_i p_i q_i = \lambda_{\min} \mathsf{Tr}(\mathbf{PQ}) \tag{8}$$

where the inequality above comes from the positive definiteness of $\mathbf{S}$, i.e., $\mathbf{S}_{i,i} = \mathbf{e}_i^\top \mathbf{S} \mathbf{e}_i \geq \lambda_{\min}, \forall i$. $\square$

**Lemma 5.** *Let $\mathbf{A} \in \mathbb{R}^{m \times n}$ be a matrix with full column rank and $\mathbf{B} \in \mathbb{R}^{n \times p}$ be a non-zero matrix. Let $\sigma_{\min}(\cdot)$ be the smallest non-zero singular value. Then it holds that $\sigma_{\min}(\mathbf{AB}) \geq \sigma_{\min}(\mathbf{A})\sigma_{\min}(\mathbf{B})$.*

*Proof.* Using the min-max principle for singular values,

$$\begin{aligned}
\sigma_{\min}(\mathbf{AB}) &= \min_{\|\mathbf{x}\|=1, \mathbf{x} \in \text{ColSpan}(\mathbf{B})} \|\mathbf{ABx}\| \\
&= \min_{\|\mathbf{x}\|=1, \mathbf{x} \in \text{ColSpan}(\mathbf{B})} \left\|\mathbf{A}\frac{\mathbf{Bx}}{\|\mathbf{Bx}\|}\right\| \cdot \|\mathbf{Bx}\| \\
&\overset{(a)}{=} \min_{\|\mathbf{x}\|=1, \|\mathbf{y}\|=1, \mathbf{x} \in \text{ColSpan}(\mathbf{B}), \mathbf{y} \in \text{ColSpan}(\mathbf{B})} \|\mathbf{Ay}\| \cdot \|\mathbf{Bx}\| \\
&\geq \min_{\|\mathbf{y}\|=1, \mathbf{y} \in \text{ColSpan}(\mathbf{B})} \|\mathbf{Ay}\| \cdot \min_{\|\mathbf{x}\|=1, \mathbf{x} \in \text{ColSpan}(\mathbf{B})} \|\mathbf{Bx}\| \\
&\geq \min_{\|\mathbf{y}\|=1} \|\mathbf{Ay}\| \cdot \min_{\|\mathbf{x}\|=1, \mathbf{x} \in \text{ColSpan}(\mathbf{B})} \|\mathbf{Bx}\| \\
&= \sigma_{\min}(\mathbf{A})\sigma_{\min}(\mathbf{B})
\end{aligned}$$

where (a) is by changing of variables, i.e., $\mathbf{y} = \mathbf{Bx}/\|\mathbf{Bx}\|$. $\square$

**Lemma 6** (Theorem 2.2.1 of (Chikuse, 2012)). *If $\mathbf{Z} \in \mathbb{R}^{m \times r}$ has entries drawn iid from Gaussian distribution $\mathcal{N}(0,1)$, then $\mathbf{X} = \mathbf{Z}(\mathbf{Z}^\top \mathbf{Z})^{-1/2}$ is a random matrix uniformly distributed on $\mathsf{St}(m,r)$.*

**Lemma 7** (Vershynin (2010)). *] If $\mathbf{Z} \in \mathbb{R}^{m \times r}$ is a matrix whose entries are independently drawn from $\mathcal{N}(0,1)$. Then for every $\tau \geq 0$, with probability at least $1 - \exp(-\tau^2/2)$, we have*

$$\sigma_1(\mathbf{Z}) \leq \sqrt{m} + \sqrt{r} + \tau.$$

**Lemma 8** (Rudelson and Vershynin (2009)). *If $\mathbf{Z} \in \mathbb{R}^{m \times r}$ is a matrix whose entries are independently drawn from $\mathcal{N}(0,1)$. Suppose that $m \geq r$. Then for every $\tau \geq 0$, we have for some universal constants $C_1 > 0$ and $C_2 > 0$ that*

$$\mathbb{P}\left(\sigma_r(\mathbf{Z}) \leq \tau(\sqrt{m} - \sqrt{r-1})\right) \leq (C_1\tau)^{m-r+1} + \exp(-C_2 r).$$

**Lemma 9.** *If $\mathbf{U} \in \mathsf{St}(m, r_A)$ is a fixed matrix, $\mathbf{X} \in \mathsf{St}(m, r)$ is uniformly sampled from $\mathsf{St}(m, r)$ using methods described in Lemma 6, and $r > r_A$, then we have that with probability at least $1 - \exp(-m/2) - (C_1\tau)^{r-r_A+1} - \exp(-C_2 d)$,*

$$\sigma_{r_A}(\mathbf{U}^\top \mathbf{X}) \geq \frac{\tau(r - r_A + 1)}{6\sqrt{mr}}.$$

*Proof.* Since $\mathbf{X} \in \mathsf{St}(m, r)$ is uniformly sampled from $\mathsf{St}(m, r)$ using methods described in Lemma 6, we can write $\mathbf{X} = \mathbf{Z}(\mathbf{Z}^\top \mathbf{Z})^{-1/2}$, where $\mathbf{Z} \in \mathbb{R}^{m \times r}$ has entries iid sampled from $\mathcal{N}(0,1)$. We thus have

$$\sigma_{r_A}(\mathbf{U}^\top \mathbf{X}) = \sigma_{r_A}\left(\mathbf{U}^\top \mathbf{Z}(\mathbf{Z}^\top \mathbf{Z})^{-1/2}\right).$$

Now consider $\mathbf{U}^\top \mathbf{Z} \in \mathbb{R}^{r_A \times r}$. It is clear that entries of $\mathbf{U}^\top \mathbf{Z}$ are also iid Gaussian random variables $\mathcal{N}(0,1)$. As a consequence of Lemma 8, we have that w.p. at least $1 - (C_1\tau)^{r-r_A+1} - \exp(-C_2 r)$,

$$\sigma_{r_A}\left(\mathbf{U}^\top \mathbf{Z}\right) \geq \tau(\sqrt{r} - \sqrt{r_A - 1}).$$

We also have from Lemma 7 that with probability at least $1 - \exp(-m/2)$

$$\sigma_1(\mathbf{Z}^\top \mathbf{Z}) = \sigma_1^2(\mathbf{Z}) \leq (2\sqrt{m} + \sqrt{r})^2.$$

Taking union bound, we have with probability at least $1 - \exp(-m/2) - (C_1\tau)^{r-r_A+1} - \exp(-C_2 r)$,

$$\sigma_{r_A}(\mathbf{U}^\top \mathbf{X}) \overset{(a)}{\geq} \frac{\sigma_{r_A}(\mathbf{U}^\top \mathbf{Z})}{\sigma_1(\mathbf{Z})} = \frac{\tau(\sqrt{r} - \sqrt{r_A - 1})}{2\sqrt{m} + \sqrt{r}} \geq \frac{\tau(r - r_A + 1)}{3\sqrt{m} \cdot 2\sqrt{r}} = \frac{\tau(r - r_A + 1)}{6\sqrt{mr}} \quad (9)$$

where (a) comes from Lemma 5. $\qquad\qquad\square$

**Lemma 10.** *If* $\mathbf{V} \in \mathsf{St}(n, r_A)$ *is a fixed matrix, and* $\mathbf{Y} \in \mathsf{St}(n, r)$ *is uniformly sampled from* $\mathsf{St}(n, r)$ *using methods described in Lemma 6. Suppose* $r > r_A$. *Then we have that with probability at least* $1 - \exp(-n/2) - (C_1\tau)^{r-r_A+1} - \exp(-C_2 r)$,

$$\sigma_{r_A}(\mathbf{V}^\top \mathbf{Y}) \geq \frac{\tau(r - r_A + 1)}{6\sqrt{nr}}.$$

*Proof.* The proof is omitted since it follows the same steps of Lemma 9. $\qquad\qquad\square$

### C.3 Equivalence of Matrix Factorization and LoRA

Whitening data refers to transforming the data such that the empirical uncentered covariance matrix of the features is identity. Suppose $\mathbf{D} \in \mathbb{R}^{n \times N}$ holds $N$ training examples in its columns with $n$ features each, then whitened data refers to having $\mathbf{D}\mathbf{D}^\top = \mathbf{I}_n$. We follow standard arguments laid out in Arora et al. (2018); Li et al. (2025a) to show that low-rank adaptation on a linear model with whitened data and squared loss can be written as a matrix factorization problem. Consider the minimization of

$$L(\mathbf{X}, \mathbf{Y}) = \|(\mathbf{W}_0 + \mathbf{X}\mathbf{Y}^\top)\mathbf{D} - \mathbf{A}\|_{\mathsf{F}}^2$$

where $\mathbf{A} \in \mathbb{R}^{m \times N}$ holds $m$ labels for each example, $\mathbf{W}_0 \in \mathbb{R}^{m \times n}$ is the pre-trained weight, and $\mathbf{X} \in \mathbb{R}^{m \times r}, \mathbf{Y} \in \mathbb{R}^{n \times r}$ are the weights of LoRA. Rewriting $L(\mathbf{X}, \mathbf{Y})$ yields

$$
\begin{aligned}
L(\mathbf{X}, \mathbf{Y}) &= \|(\mathbf{W}_0 + \mathbf{X}\mathbf{Y}^\top)\mathbf{D} - \mathbf{A}\|_{\mathsf{F}}^2 \\
&= \mathsf{Tr}(((\mathbf{W}_0 + \mathbf{X}\mathbf{Y}^\top)\mathbf{D} - \mathbf{A})((\mathbf{W}_0 + \mathbf{X}\mathbf{Y}^\top)\mathbf{D} - \mathbf{A})^\top) \\
&= \mathsf{Tr}((\mathbf{W}_0 + \mathbf{X}\mathbf{Y}^\top)\mathbf{D}\mathbf{D}^\top(\mathbf{W}_0 + \mathbf{X}\mathbf{Y}^\top)^\top) - \mathsf{Tr}((\mathbf{W}_0 + \mathbf{X}\mathbf{Y}^\top)\mathbf{D}\mathbf{A}^\top) \\
&\qquad - \mathsf{Tr}(\mathbf{A}\mathbf{D}^\top(\mathbf{W}_0 + \mathbf{X}\mathbf{Y}^\top)^\top) + \mathsf{Tr}(\mathbf{A}\mathbf{A}^\top) \\
&\overset{(a)}{=} \mathsf{Tr}((\mathbf{W}_0 + \mathbf{X}\mathbf{Y}^\top)(\mathbf{W}_0 + \mathbf{X}\mathbf{Y}^\top)^\top) - \mathsf{Tr}((\mathbf{W}_0 + \mathbf{X}\mathbf{Y}^\top)\mathbf{D}\mathbf{A}^\top) \\
&\qquad - \mathsf{Tr}(\mathbf{A}\mathbf{D}^\top(\mathbf{W}_0 + \mathbf{X}\mathbf{Y}^\top)^\top) + \mathsf{Tr}(\mathbf{A}\mathbf{A}^\top) \\
&\overset{(b)}{=} \mathsf{Tr}(((\mathbf{W}_0 + \mathbf{X}\mathbf{Y}^\top) - \mathbf{\Lambda})((\mathbf{W}_0 + \mathbf{X}\mathbf{Y}^\top) - \mathbf{\Lambda})^\top) - \mathsf{Tr}(\mathbf{\Lambda}\mathbf{\Lambda}^\top) + \mathsf{Tr}(\mathbf{A}\mathbf{A}^\top)
\end{aligned}
$$

where $(a)$ uses the fact that the data is whitened and $(b)$ defines $\mathbf{\Lambda} = \mathbf{A}\mathbf{D}^\top \in \mathbb{R}^{m \times n}$, i.e., the matrix to be factorized. Thus, we can write

$$
\begin{aligned}
L(\mathbf{X}, \mathbf{Y}) &= \|(\mathbf{W}_0 + \mathbf{X}\mathbf{Y}^\top) - \mathbf{\Lambda}\|_{\mathsf{F}}^2 + c \\
&= \|\mathbf{X}\mathbf{Y}^\top - \mathbf{\Lambda}'\|_{\mathsf{F}}^2 + c
\end{aligned}
$$

for $\mathbf{\Lambda}' := \mathbf{W}_0 - \mathbf{\Lambda}$ and constant $c := -\mathsf{Tr}(\mathbf{\Lambda}\mathbf{\Lambda}^\top) + \mathsf{Tr}(\mathbf{A}\mathbf{A}^\top)$. Using the same arguments, one can frame low-rank adaptation of a linear model with the PoLAR parameterization as the matrix factorization problem given in (4).

## D Proofs for Asymmetric Problems

### D.1 Riemannian Gradients of (4)

One can start with the Euclidean gradient with respect to $\mathbf{X}_t$ as $\tilde{\mathbf{E}}_t = (\mathbf{X}_t\mathbf{\Theta}_t\mathbf{Y}_t^\top - \mathbf{A})\mathbf{Y}_t\mathbf{\Theta}_t^\top = (\mathbf{X}_t\mathbf{X}_t^\top - \mathbf{I}_m)\mathbf{A}\mathbf{Y}_t\mathbf{\Theta}_t^\top$. Note that for $\mathbf{\Theta}_t = \mathbf{X}_t^\top \mathbf{A}\mathbf{Y}_t$ (i.e., $\gamma = 1$) the Euclidean gradient is skew-symmetric such that it is already contained in the tangent space of $\mathsf{St}(m, r)$ at $\mathbf{X}_t$ (i.e., $\mathbf{X}_t^\top \tilde{\mathbf{E}}_t +$

$\tilde{\mathbf{E}}_t^\top \mathbf{X}_t = 0$), yielding equality between the Riemannian gradient $\mathbf{E}_t$ and Euclidean gradient $\tilde{\mathbf{E}}_t$. This equivalence holds regardless of whether the Euclidean or canonical metric is used. In other words, the Riemmanian gradient for $\mathbf{X}_t$ at iteration $t$ is given by

$$\mathbf{E}_t = -(\mathbf{I}_m - \mathbf{X}_t \mathbf{X}_t^\top)\mathbf{A}\mathbf{Y}_t\mathbf{\Theta}_t^\top.$$

Similarly, one can obtain the Riemannian gradient for $\mathbf{Y}_t$ via

$$\mathbf{F}_t = -(\mathbf{I}_n - \mathbf{Y}_t \mathbf{Y}_t^\top)\mathbf{A}^\top \mathbf{X}_t\mathbf{\Theta}_t.$$

### D.2 General Dynamics

Here we derive several equations that are useful *throughout this section.* Note that the choice of learning rate $\gamma = 1$ will be leveraged in some equations.

**Dynamics on $\mathbf{X}_t$, $\mathbf{E}_t$, and $\mathbf{\Phi}_t$.** From the updates in Alg. 1, it is straightforward to arrive

$$\begin{aligned}
\mathbb{R}^{r_A \times r} \ni \mathbf{U}^\top \mathbf{E}_t &= -\mathbf{U}^\top(\mathbf{I}_m - \mathbf{X}_t\mathbf{X}_t^\top)\mathbf{A}\mathbf{Y}_t\mathbf{\Theta}_t^\top \\
&= -\mathbf{U}^\top(\mathbf{I}_m - \mathbf{X}_t\mathbf{X}_t^\top)\mathbf{U}\mathbf{\Sigma}\mathbf{V}^\top\mathbf{Y}_t\mathbf{\Theta}_t^\top \\
&= -(\mathbf{I}_{r_A} - \mathbf{\Phi}_t\mathbf{\Phi}_t^\top)\mathbf{\Sigma}\mathbf{\Psi}_t\mathbf{\Theta}_t^\top.
\end{aligned}$$

Similarly, we also have

$$\begin{aligned}
\mathbb{R}^{r \times r} \ni \mathbf{E}_t^\top\mathbf{E}_t &= \mathbf{\Theta}_t\mathbf{Y}_t^\top\mathbf{A}^\top(\mathbf{I}_m - \mathbf{X}_t\mathbf{X}_t^\top)^2\mathbf{A}\mathbf{Y}_t\mathbf{\Theta}_t^\top \\
&= \mathbf{\Theta}_t\mathbf{Y}_t^\top\mathbf{A}^\top(\mathbf{I}_m - \mathbf{X}_t\mathbf{X}_t^\top)\mathbf{A}\mathbf{Y}_t\mathbf{\Theta}_t^\top \\
&= \mathbf{\Theta}_t\mathbf{\Psi}_t^\top\mathbf{\Sigma}(\mathbf{I}_{r_A} - \mathbf{\Phi}_t\mathbf{\Phi}_t^\top)\mathbf{\Sigma}\mathbf{\Psi}_t\mathbf{\Theta}_t^\top.
\end{aligned}$$

Applying $\sigma_1(\mathbf{\Psi}_t) \le 1$ and $\sigma_1(\mathbf{A}) = \sigma_1(\mathbf{\Sigma}) = 1$ to the equation above, and leveraging $\gamma = 1$ in the update of $\mathbf{\Theta}_t$ (i.e., $\mathbf{\Theta}_t = \mathbf{X}_t^\top\mathbf{A}\mathbf{Y}_t = \mathbf{\Phi}_t^\top\mathbf{\Sigma}\mathbf{\Psi}_t$), we have that

$$\sigma_1(\mathbf{E}_t^\top\mathbf{E}_t) \le \sigma_1^4(\mathbf{\Sigma})\sigma_1(\mathbf{I}_{r_A} - \mathbf{\Phi}_t\mathbf{\Phi}_t^\top) = \sigma_1(\mathbf{I}_{r_A} - \mathbf{\Phi}_t\mathbf{\Phi}_t^\top). \tag{10}$$

And the dynamics on the alignment $\mathbf{\Phi}_t \in \mathbb{R}^{r_A \times r}$ can be written as

$$\begin{aligned}
\mathbf{\Phi}_{t+1} &= \left[\mathbf{\Phi}_t + \eta(\mathbf{I}_{r_A} - \mathbf{\Phi}_t\mathbf{\Phi}_t^\top)\mathbf{\Sigma}\mathbf{\Psi}_t\mathbf{\Theta}_t^\top\right]\left(\mathbf{I}_r + \eta^2\mathbf{E}_t^\top\mathbf{E}_t\right)^{-1/2} \\
&\overset{(a)}{=} \left[\mathbf{I}_{r_A} + \eta(\mathbf{I}_{r_A} - \mathbf{\Phi}_t\mathbf{\Phi}_t^\top)\mathbf{\Sigma}\mathbf{\Psi}_t\mathbf{\Psi}_t^\top\mathbf{\Sigma}\right]\mathbf{\Phi}_t\left(\mathbf{I}_r + \eta^2\mathbf{E}_t^\top\mathbf{E}_t\right)^{-1/2}
\end{aligned}$$

where $(a)$ uses $\mathbf{\Theta}_t = \mathbf{X}_t^\top\mathbf{A}\mathbf{Y}_t = \mathbf{\Phi}_t^\top\mathbf{\Sigma}\mathbf{\Psi}_t$. Hence, we have that

$$\begin{aligned}
&\mathbf{\Phi}_{t+1}\mathbf{\Phi}_{t+1}^\top \\
&= \left[\mathbf{I}_{r_A} + \eta(\mathbf{I}_{r_A} - \mathbf{\Phi}_t\mathbf{\Phi}_t^\top)\mathbf{\Sigma}\mathbf{\Psi}_t\mathbf{\Psi}_t^\top\mathbf{\Sigma}\right]\mathbf{\Phi}_t\left(\mathbf{I}_r + \eta^2\mathbf{E}_t^\top\mathbf{E}_t\right)^{-1}\mathbf{\Phi}_t^\top\left[\mathbf{I}_{r_A} + \eta\mathbf{\Sigma}\mathbf{\Psi}_t\mathbf{\Psi}_t^\top\mathbf{\Sigma}(\mathbf{I}_{r_A} - \mathbf{\Phi}_t\mathbf{\Phi}_t^\top)\right].
\end{aligned} \tag{11}$$

**Dynamics on $\mathbf{Y}_t$, $\mathbf{F}_t$, and $\mathbf{\Psi}_t$.** From the updates in Alg. 1 and similar to the derivation above, we arrive at

$$\begin{aligned}
\mathbb{R}^{r_A \times r} \ni \mathbf{V}^\top\mathbf{F}_t &= -\mathbf{V}^\top(\mathbf{I}_n - \mathbf{Y}_t\mathbf{Y}_t^\top)\mathbf{A}^\top\mathbf{X}_t\mathbf{\Theta}_t \\
&= -\mathbf{V}^\top(\mathbf{I}_n - \mathbf{Y}_t\mathbf{Y}_t^\top)\mathbf{V}\mathbf{\Sigma}\mathbf{U}^\top\mathbf{X}_t\mathbf{\Theta}_t \\
&= -(\mathbf{I}_{r_A} - \mathbf{\Psi}_t\mathbf{\Psi}_t^\top)\mathbf{\Sigma}\mathbf{\Phi}_t\mathbf{\Theta}_t.
\end{aligned}$$

Moreover, we also have

$$\begin{aligned}
\mathbb{R}^{r \times r} \ni \mathbf{F}_t^\top\mathbf{F}_t &= \mathbf{\Theta}_t^\top\mathbf{X}_t^\top\mathbf{A}(\mathbf{I}_n - \mathbf{Y}_t\mathbf{Y}_t^\top)^2\mathbf{A}^\top\mathbf{X}_t\mathbf{\Theta}_t \\
&= \mathbf{\Theta}_t^\top\mathbf{X}_t^\top\mathbf{A}(\mathbf{I}_n - \mathbf{Y}_t\mathbf{Y}_t^\top)\mathbf{A}^\top\mathbf{X}_t\mathbf{\Theta}_t \\
&= \mathbf{\Theta}_t^\top\mathbf{\Phi}_t^\top\mathbf{\Sigma}(\mathbf{I}_{r_A} - \mathbf{\Psi}_t\mathbf{\Psi}_t^\top)\mathbf{\Sigma}\mathbf{\Phi}_t\mathbf{\Theta}_t.
\end{aligned}$$

Applying $\sigma_1(\mathbf{\Phi}_t) \le 1$ and $\sigma_1(\mathbf{A}) = 1$ to the equation above, we have that

$$\sigma_1(\mathbf{F}_t^\top\mathbf{F}_t) \le \sigma_1^4(\mathbf{\Sigma})\sigma_1(\mathbf{I}_{r_A} - \mathbf{\Psi}_t\mathbf{\Psi}_t^\top) = \sigma_1(\mathbf{I}_{r_A} - \mathbf{\Psi}_t\mathbf{\Psi}_t^\top). \tag{12}$$

The alignment $\mathbf{\Psi}_t = \mathbf{V}^\top \mathbf{Y}_t \in \mathbb{R}^{r_A \times r}$ can be tracked via

$$\mathbf{\Psi}_{t+1} = \left[\mathbf{\Psi}_t + \eta(\mathbf{I}_{r_A} - \mathbf{\Psi}_t\mathbf{\Psi}_t^\top)\mathbf{\Sigma}\mathbf{\Phi}_t\mathbf{\Theta}_t\right]\left(\mathbf{I}_r + \eta^2\mathbf{F}_t^\top\mathbf{F}_t\right)^{-1/2}$$

$$\overset{(b)}{=} \left[\mathbf{I}_{r_A} + \eta(\mathbf{I}_{r_A} - \mathbf{\Psi}_t\mathbf{\Psi}_t^\top)\mathbf{\Sigma}\mathbf{\Phi}_t\mathbf{\Phi}_t^\top\mathbf{\Sigma}\right]\mathbf{\Psi}_t\left(\mathbf{I}_r + \eta^2\mathbf{F}_t^\top\mathbf{F}_t\right)^{-1/2}$$

where $(b)$ uses $\mathbf{\Theta}_t = \mathbf{X}_t^\top\mathbf{A}\mathbf{Y}_t = \mathbf{\Phi}_t^\top\mathbf{\Sigma}\mathbf{\Psi}_t$. Finally, we have that

$$\mathbf{\Psi}_{t+1}\mathbf{\Psi}_{t+1}^\top \tag{13}$$
$$= \left[\mathbf{I}_{r_A} + \eta(\mathbf{I}_{r_A} - \mathbf{\Psi}_t\mathbf{\Psi}_t^\top)\mathbf{\Sigma}\mathbf{\Phi}_t\mathbf{\Phi}_t^\top\mathbf{\Sigma}\right]\mathbf{\Psi}_t\left(\mathbf{I}_r + \eta^2\mathbf{F}_t^\top\mathbf{F}_t\right)^{-1}\mathbf{\Psi}_t^\top\left[\mathbf{I}_{r_A} + \eta\mathbf{\Sigma}\mathbf{\Phi}_t\mathbf{\Phi}_t^\top\mathbf{\Sigma}(\mathbf{I}_{r_A} - \mathbf{\Psi}_t\mathbf{\Psi}_t^\top)\right].$$

With these preparations, we are ready to prove our main results.

### D.3  Initialization

**Lemma 11.** *Suppose that $\mathbf{X}_0$ and $\mathbf{Y}_0$ are uniformly sampled from $\mathsf{St}(m, r)$ and $\mathsf{St}(n, r)$, respectively, using methods described in Lemma 6. There exist universal constants $c_1$ and $c_2$ such that whp the following holds*

$$\sigma_{r_A}(\mathbf{\Phi}_0) = \sigma_{r_A}(\mathbf{U}^\top\mathbf{X}_0) \geq \frac{r - r_A + 1}{\sqrt{c_1 mr}} \geq \frac{r - r_A}{\sqrt{c_1 mr}}$$

$$\sigma_{r_A}(\mathbf{\Psi}_0) = \sigma_{r_A}(\mathbf{V}^\top\mathbf{Y}_0) \geq \frac{r - r_A + 1}{\sqrt{c_2 nr}} \geq \frac{r - r_A}{\sqrt{c_2 nr}}.$$

*Proof.* The proofs, the constants $c_1$ and $c_2$, as well as the exact probability follow directly from Lemma 9 and Lemma 10. □

### D.4  Increasing Alignment

Lemma 1 is proved in this subsection, where the detailed proof is divided into two parts.

#### D.4.1  Increasing Alignment Between $\mathbf{X}_t$ and $\mathbf{U}$

**Lemma 12.** *Consider Alg. 1 with $\eta < 1$ and $\gamma = 1$. Let $\beta_t := \sigma_1(\mathbf{I}_{r_A} - \mathbf{\Phi}_t\mathbf{\Phi}_t^\top)$. If it holds that*

$$\frac{2(1 - \eta^2\beta_t)\sigma_{r_A}^2(\mathbf{\Psi}_t)}{\kappa^2}\mathsf{Tr}\left((\mathbf{I}_{r_A} - \mathbf{\Phi}_t\mathbf{\Phi}_t^\top)\mathbf{\Phi}_t\mathbf{\Phi}_t^\top\right) \geq \eta\beta_t\mathsf{Tr}\left(\mathbf{\Phi}_t\mathbf{\Phi}_t^\top\right),$$

*we have $\mathsf{Tr}(\mathbf{\Phi}_{t+1}\mathbf{\Phi}_{t+1}^\top) \geq \mathsf{Tr}(\mathbf{\Phi}_t\mathbf{\Phi}_t^\top)$.*

*Proof.* From (11), we have that

$$\mathbf{\Phi}_{t+1}\mathbf{\Phi}_{t+1}^\top \tag{14}$$
$$= \left[\mathbf{I}_{r_A} + \eta(\mathbf{I}_{r_A} - \mathbf{\Phi}_t\mathbf{\Phi}_t^\top)\mathbf{\Sigma}\mathbf{\Psi}_t\mathbf{\Psi}_t^\top\mathbf{\Sigma}\right]\mathbf{\Phi}_t\left(\mathbf{I}_r + \eta^2\mathbf{E}_t^\top\mathbf{E}_t\right)^{-1}\mathbf{\Phi}_t^\top\left[\mathbf{I}_{r_A} + \eta\mathbf{\Sigma}\mathbf{\Psi}_t\mathbf{\Psi}_t^\top\mathbf{\Sigma}(\mathbf{I}_{r_A} - \mathbf{\Phi}_t\mathbf{\Phi}_t^\top)\right]$$
$$\overset{(a)}{\succeq} \left[\mathbf{I}_{r_A} + \eta(\mathbf{I}_{r_A} - \mathbf{\Phi}_t\mathbf{\Phi}_t^\top)\mathbf{\Sigma}\mathbf{\Psi}_t\mathbf{\Psi}_t^\top\mathbf{\Sigma}\right]\mathbf{\Phi}_t\left(\mathbf{I}_r - \eta^2\mathbf{E}_t^\top\mathbf{E}_t\right)\mathbf{\Phi}_t^\top\left[\mathbf{I}_{r_A} + \eta\mathbf{\Sigma}\mathbf{\Psi}_t\mathbf{\Psi}_t^\top\mathbf{\Sigma}(\mathbf{I}_{r_A} - \mathbf{\Phi}_t\mathbf{\Phi}_t^\top)\right]$$
$$\overset{(b)}{\succeq} (1 - \eta^2\beta_t)\left[\mathbf{I}_{r_A} + \eta(\mathbf{I}_{r_A} - \mathbf{\Phi}_t\mathbf{\Phi}_t^\top)\mathbf{\Sigma}\mathbf{\Psi}_t\mathbf{\Psi}_t^\top\mathbf{\Sigma}\right]\mathbf{\Phi}_t\mathbf{\Phi}_t^\top\left[\mathbf{I}_{r_A} + \eta\mathbf{\Sigma}\mathbf{\Psi}_t\mathbf{\Psi}_t^\top\mathbf{\Sigma}(\mathbf{I}_{r_A} - \mathbf{\Phi}_t\mathbf{\Phi}_t^\top)\right]$$
$$\succeq (1 - \eta^2\beta_t)\left\{\mathbf{\Phi}_t\mathbf{\Phi}_t^\top + \eta(\mathbf{I}_{r_A} - \mathbf{\Phi}_t\mathbf{\Phi}_t^\top)\mathbf{\Sigma}\mathbf{\Psi}_t\mathbf{\Psi}_t^\top\mathbf{\Sigma}\mathbf{\Phi}_t\mathbf{\Phi}_t^\top + \eta\mathbf{\Phi}_t\mathbf{\Phi}_t^\top\mathbf{\Sigma}\mathbf{\Psi}_t\mathbf{\Psi}_t^\top\mathbf{\Sigma}(\mathbf{I}_{r_A} - \mathbf{\Phi}_t\mathbf{\Phi}_t^\top)\right\}$$

where $(a)$ is by Lemma 2; and $(b)$ is by $\left(\mathbf{I}_r - \eta^2\mathbf{E}_t^\top\mathbf{E}_t\right) \succeq (1 - \eta^2\sigma_1(\mathbf{I}_{r_A} - \mathbf{\Phi}_t\mathbf{\Phi}_t^\top))\mathbf{I}_r$ as a result of (10), and we write $\beta_t := \sigma_1(\mathbf{I}_{r_A} - \mathbf{\Phi}_t\mathbf{\Phi}_t^\top)$ for convenience. We also dropped the fourth term $(\mathbf{I}_{r_A} - \mathbf{\Phi}_t\mathbf{\Phi}_t^\top)\mathbf{\Sigma}\mathbf{\Psi}_t\mathbf{\Psi}_t^\top\mathbf{\Sigma}\mathbf{\Phi}_t\mathbf{\Phi}_t^\top\mathbf{\Sigma}_t\mathbf{\Psi}_t\mathbf{\Psi}_t^\top\mathbf{\Sigma}(\mathbf{I}_{r_A} - \mathbf{\Phi}_t\mathbf{\Phi}_t^\top)$ given its PSDness. Note that $\beta_t \in [0, 1]$.

Now let the EVD of $\mathbf{\Phi}_t\mathbf{\Phi}_t^\top = \mathbf{Q}_t\mathbf{\Lambda}_t\mathbf{Q}_t^\top$, where both $\mathbf{Q}_t$ and $\mathbf{\Lambda}_t$ are $r_A \times r_A$ matrices. Note that $\mathbf{0} \preceq \mathbf{\Lambda}_t \preceq \mathbf{I}_{r_A}$. Then we have that

$$\mathsf{Tr}\Big((\mathbf{I}_{r_A} - \mathbf{\Phi}_t\mathbf{\Phi}_t^\top)\mathbf{\Sigma}\mathbf{\Psi}_t\mathbf{\Psi}_t^\top\mathbf{\Sigma}\mathbf{\Phi}_t\mathbf{\Phi}_t^\top\Big) \tag{15}$$

$$= \mathsf{Tr}\Big(\mathbf{Q}_t(\mathbf{I}_{r_A} - \mathbf{\Lambda}_t)\mathbf{Q}_t^\top\mathbf{\Sigma}\mathbf{\Psi}_t\mathbf{\Psi}_t^\top\mathbf{\Sigma}\mathbf{Q}_t\mathbf{\Lambda}_t\mathbf{Q}_t^\top\Big)$$

$$= \mathsf{Tr}\Big((\mathbf{I}_{r_A} - \mathbf{\Lambda}_t)\mathbf{Q}_t^\top\mathbf{\Sigma}\mathbf{\Psi}_t\mathbf{\Psi}_t^\top\mathbf{\Sigma}\mathbf{Q}_t\mathbf{\Lambda}_t\Big)$$

$$\overset{(c)}{\succeq} \frac{\sigma_{r_A}^2(\mathbf{\Psi}_t)}{\kappa^2}\mathsf{Tr}\Big((\mathbf{I}_{r_A} - \mathbf{\Lambda}_t)\mathbf{\Lambda}_t\Big)$$

$$= \frac{\sigma_{r_A}^2(\mathbf{\Psi}_t)}{\kappa^2}\mathsf{Tr}\Big((\mathbf{I}_{r_A} - \mathbf{\Phi}_t\mathbf{\Phi}_t^\top)\mathbf{\Phi}_t\mathbf{\Phi}_t^\top\Big)$$

where $(c)$ is by Lemma 4 and Lemma 5. More precisely, the PSDness of $\mathbf{Q}_t^\top\mathbf{\Sigma}\mathbf{\Psi}_t\mathbf{\Psi}_t^\top\mathbf{\Sigma}\mathbf{Q}_t$ justifies the prerequisites for Lemma 4, and then we use $\sigma_{r_A}(\mathbf{Q}_t^\top\mathbf{\Sigma}\mathbf{\Psi}_t\mathbf{\Psi}_t^\top\mathbf{\Sigma}\mathbf{Q}_t) \geq \sigma_{r_A}^2(\mathbf{\Sigma})\sigma_{r_A}^2(\mathbf{\Psi}_t) = \sigma_{r_A}^2(\mathbf{\Psi}_t)/\kappa^2$.

Taking trace on both sides of (14) and plugging (15) in, we arrive at

$$\frac{\mathsf{Tr}(\mathbf{\Phi}_{t+1}\mathbf{\Phi}_{t+1}^\top)}{1 - \eta^2\beta_t} \geq \mathsf{Tr}(\mathbf{\Phi}_t\mathbf{\Phi}_t^\top) + \frac{2\eta\sigma_{r_A}^2(\mathbf{\Psi}_t)}{\kappa^2}\mathsf{Tr}\Big((\mathbf{I}_{r_A} - \mathbf{\Phi}_t\mathbf{\Phi}_t^\top)\mathbf{\Phi}_t\mathbf{\Phi}_t^\top\Big). \tag{16}$$

Simplifying this inequality gives the results. $\qquad\square$

### D.4.2 Increasing Alignment Between $\mathbf{Y}_t$ and $\mathbf{V}$

We proceed by proving the analogue of Lemma 12 for the alignment between $\mathbf{Y}_t$ and $\mathbf{V}$ by following the same steps.

**Lemma 13.** *Consider Alg. 1 with $\eta < 1$ and $\gamma = 1$. Let $\delta_t := \sigma_1(\mathbf{I}_{r_A} - \mathbf{\Psi}_t\mathbf{\Psi}_t^\top)$. If it holds that*

$$\frac{2(1 - \eta^2\delta_t)\sigma_{r_A}^2(\mathbf{\Phi}_t)}{\kappa^2}\mathsf{Tr}\big((\mathbf{I}_{r_A} - \mathbf{\Psi}_t\mathbf{\Psi}_t^\top)\mathbf{\Psi}_t\mathbf{\Psi}_t^\top\big) \geq \eta\delta_t\mathsf{Tr}\big(\mathbf{\Psi}_t\mathbf{\Psi}_t^\top\big),$$

*we have $\mathsf{Tr}(\mathbf{\Psi}_{t+1}\mathbf{\Psi}_{t+1}^\top) \geq \mathsf{Tr}(\mathbf{\Psi}_t\mathbf{\Psi}_t^\top)$.*

*Proof.* From (13), we have that

$$\mathbf{\Psi}_{t+1}\mathbf{\Psi}_{t+1}^\top \tag{17}$$

$$= \big[\mathbf{I}_{r_A} + \eta(\mathbf{I}_{r_A} - \mathbf{\Psi}_t\mathbf{\Psi}_t^\top)\mathbf{\Sigma}\mathbf{\Phi}_t\mathbf{\Phi}_t^\top\mathbf{\Sigma}\big]\mathbf{\Psi}_t\big(\mathbf{I}_r + \eta^2\mathbf{F}_t^\top\mathbf{F}_t\big)^{-1}\mathbf{\Psi}_t^\top\big[\mathbf{I}_{r_A} + \eta\mathbf{\Sigma}\mathbf{\Phi}_t\mathbf{\Phi}_t^\top\mathbf{\Sigma}(\mathbf{I}_{r_A} - \mathbf{\Psi}_t\mathbf{\Psi}_t^\top)\big]$$

$$\overset{(a)}{\succeq} \big[\mathbf{I}_{r_A} + \eta(\mathbf{I}_{r_A} - \mathbf{\Psi}_t\mathbf{\Psi}_t^\top)\mathbf{\Sigma}\mathbf{\Phi}_t\mathbf{\Phi}_t^\top\mathbf{\Sigma}\big]\mathbf{\Psi}_t\big(\mathbf{I}_r - \eta^2\mathbf{F}_t^\top\mathbf{F}_t\big)\mathbf{\Psi}_t^\top\big[\mathbf{I}_{r_A} + \eta\mathbf{\Sigma}\mathbf{\Phi}_t\mathbf{\Phi}_t^\top\mathbf{\Sigma}(\mathbf{I}_{r_A} - \mathbf{\Psi}_t\mathbf{\Psi}_t^\top)\big]$$

$$\overset{(b)}{\succeq} (1 - \eta^2\delta_t)\big[\mathbf{I}_{r_A} + \eta(\mathbf{I}_{r_A} - \mathbf{\Psi}_t\mathbf{\Psi}_t^\top)\mathbf{\Sigma}\mathbf{\Phi}_t\mathbf{\Phi}_t^\top\mathbf{\Sigma}\big]\mathbf{\Psi}_t\mathbf{\Psi}_t^\top\big[\mathbf{I}_{r_A} + \eta\mathbf{\Sigma}\mathbf{\Phi}_t\mathbf{\Phi}_t^\top\mathbf{\Sigma}(\mathbf{I}_{r_A} - \mathbf{\Psi}_t\mathbf{\Psi}_t^\top)\big]$$

$$\succeq (1 - \eta^2\delta_t)\Big\{\mathbf{\Psi}_t\mathbf{\Psi}_t^\top + \eta(\mathbf{I}_{r_A} - \mathbf{\Psi}_t\mathbf{\Psi}_t^\top)\mathbf{\Sigma}\mathbf{\Phi}_t\mathbf{\Phi}_t^\top\mathbf{\Sigma}\mathbf{\Psi}_t\mathbf{\Psi}_t^\top + \eta\mathbf{\Psi}_t\mathbf{\Psi}_t^\top\mathbf{\Sigma}\mathbf{\Phi}_t\mathbf{\Phi}_t^\top\mathbf{\Sigma}(\mathbf{I}_{r_A} - \mathbf{\Psi}_t\mathbf{\Psi}_t^\top)\Big\}$$

where $(a)$ is by Lemma 2; and $(b)$ is by $\big(\mathbf{I}_r - \eta^2\mathbf{F}_t^\top\mathbf{F}_t\big) \succeq (1 - \eta^2\sigma_1(\mathbf{I}_{r_A} - \mathbf{\Psi}_t\mathbf{\Psi}_t^\top))\mathbf{I}_r$ as a result of (12), and we write $\delta_t := \sigma_1(\mathbf{I}_{r_A} - \mathbf{\Psi}_t\mathbf{\Psi}_t^\top)$ for convenience. Note that $\delta_t \in [0, 1]$.

Now let the SVD of $\mathbf{\Psi}_t\mathbf{\Psi}_t^\top = \mathbf{P}_t\tilde{\mathbf{\Lambda}}_t\mathbf{P}_t^\top$, where both $\mathbf{P}_t$ and $\tilde{\mathbf{\Lambda}}_t$ are $r_A \times r_A$ matrices. Note that $\mathbf{0} \preceq \tilde{\mathbf{\Lambda}}_t \preceq \mathbf{I}_{r_A}$. Then we have that

$$\mathsf{Tr}\Big((\mathbf{I}_{r_A} - \mathbf{\Psi}_t\mathbf{\Psi}_t^\top)\mathbf{\Sigma}\mathbf{\Phi}_t\mathbf{\Phi}_t^\top\mathbf{\Sigma}\mathbf{\Psi}_t\mathbf{\Psi}_t^\top\Big) \tag{18}$$

$$= \mathsf{Tr}\Big(\mathbf{P}_t(\mathbf{I}_{r_A} - \tilde{\mathbf{\Lambda}}_t)\mathbf{P}_t^\top\mathbf{\Sigma}\mathbf{\Phi}_t\mathbf{\Phi}_t^\top\mathbf{\Sigma}\mathbf{P}_t\tilde{\mathbf{\Lambda}}_t\mathbf{P}_t^\top\Big)$$

$$= \mathsf{Tr}\Big((\mathbf{I}_{r_A} - \tilde{\mathbf{\Lambda}}_t)\mathbf{P}_t^\top\mathbf{\Sigma}\mathbf{\Phi}_t\mathbf{\Phi}_t^\top\mathbf{\Sigma}\mathbf{P}_t\tilde{\mathbf{\Lambda}}_t\Big)$$

$$\overset{(c)}{\succeq} \frac{\sigma_{r_A}^2(\mathbf{\Phi}_t)}{\kappa^2}\mathsf{Tr}\Big((\mathbf{I}_{r_A} - \tilde{\mathbf{\Lambda}}_t)\tilde{\mathbf{\Lambda}}_t\Big)$$

$$= \frac{\sigma_{r_A}^2(\mathbf{\Phi}_t)}{\kappa^2}\mathsf{Tr}\Big((\mathbf{I}_{r_A} - \mathbf{\Psi}_t\mathbf{\Psi}_t^\top)\mathbf{\Psi}_t\mathbf{\Psi}_t^\top\Big)$$

where $(c)$ is by Lemma 4 and Lemma 5. More precisely, we use the PSDness of $\mathbf{P}_t^\top \mathbf{\Sigma} \mathbf{\Phi}_t \mathbf{\Phi}_t^\top \mathbf{\Sigma} \mathbf{P}_t$ for applying Lemma 4, and then employ $\sigma_{r_A}(\mathbf{P}_t^\top \mathbf{\Sigma} \mathbf{\Phi}_t \mathbf{\Phi}_t^\top \mathbf{\Sigma} \mathbf{P}_t) \geq \sigma_{r_A}^2(\mathbf{\Sigma}) \sigma_{r_A}^2(\mathbf{\Phi}_t) = \sigma_{r_A}^2(\mathbf{\Phi}_t)/\kappa^2$.

Taking trace on both sides of (17) and plugging (18) in, we arrive at

$$\frac{\mathsf{Tr}(\mathbf{\Psi}_{t+1}\mathbf{\Psi}_{t+1}^\top)}{1 - \eta^2 \delta_t} \geq \mathsf{Tr}(\mathbf{\Psi}_t \mathbf{\Psi}_t^\top) + \frac{2\eta \sigma_{r_A}^2(\mathbf{\Phi}_t)}{\kappa^2} \mathsf{Tr}\Big( (\mathbf{I}_{r_A} - \mathbf{\Psi}_t \mathbf{\Psi}_t^\top)\mathbf{\Psi}_t \mathbf{\Psi}_t^\top \Big). \qquad (19)$$

Simplifying this inequality gives the results. $\qquad \square$

### D.5 Non-Increasing Misalignment

Misalignment of $\mathbf{X}_t$ refers to the principal angles between $\mathbf{X}_t$ and the basis of the orthogonal complement of $\mathbf{U}$. Similarly, one can define the misalignment of $\mathbf{Y}_t$.

#### D.5.1 Non-Increasing Misalignment of $\mathbf{X}_t$

**Lemma 14.** *Denote the orthogonal complement of $\mathbf{U}$ as $\mathbf{U}_\perp \in \mathbb{R}^{m \times (m - r_A)}$. Define the $(m - r_A) \times r$ matrix $\mathbf{\Omega}_t := \mathbf{U}_\perp^\top \mathbf{X}_t$ to characterize the alignment of $\mathbf{X}_t$ and $\mathbf{U}_\perp$. Under the same setting of Lemma 12, we have that $\mathbf{\Omega}_{t+1}\mathbf{\Omega}_{t+1}^\top \preceq \mathbf{\Omega}_t \mathbf{\Omega}_t^\top$. Moreover, if $r_A \leq \frac{m}{2}$, it is guaranteed to have $\sigma_{r_A}^2(\mathbf{\Phi}_{t+1}) \geq \sigma_{r_A}^2(\mathbf{\Phi}_t)$.*

*Proof.* From update (5b), we have that

$$\begin{aligned}
\mathbf{\Omega}_{t+1} &= \mathbf{U}_\perp^\top (\mathbf{X}_t - \eta \mathbf{E}_t)(\mathbf{I}_r + \eta^2 \mathbf{E}_t^\top \mathbf{E}_t)^{-1/2} \\
&= \big( \mathbf{\Omega}_t + \eta \mathbf{U}_\perp^\top (\mathbf{I}_m - \mathbf{X}_t \mathbf{X}_t^\top)\mathbf{A}\mathbf{Y}_t \mathbf{\Theta}_t^\top \big)(\mathbf{I}_r + \eta^2 \mathbf{E}_t^\top \mathbf{E}_t)^{-1/2} \\
&\overset{(a)}{=} \big( \mathbf{\Omega}_t - \eta \mathbf{U}_\perp^\top \mathbf{X}_t \mathbf{\Theta}_t \mathbf{\Theta}_t^\top \big)(\mathbf{I}_r + \eta^2 \mathbf{E}_t^\top \mathbf{E}_t)^{-1/2} \\
&= \mathbf{\Omega}_t \big( \mathbf{I}_r - \eta \mathbf{\Theta}_t \mathbf{\Theta}_t^\top \big)(\mathbf{I}_r + \eta^2 \mathbf{E}_t^\top \mathbf{E}_t)^{-1/2}.
\end{aligned}$$

where in (a) we have used $\mathbf{U}_\perp^\top \mathbf{A} = \mathbf{0}$ and $\mathbf{\Theta}_t = \mathbf{X}_t^\top \mathbf{A}\mathbf{Y}_t$. With this, we can see that

$$\begin{aligned}
\mathbf{\Omega}_{t+1}\mathbf{\Omega}_{t+1}^\top &= \mathbf{\Omega}_t \big( \mathbf{I}_r - \eta \mathbf{\Theta}_t \mathbf{\Theta}_t^\top \big)\big( \mathbf{I}_r + \eta^2 \mathbf{E}_t^\top \mathbf{E}_t \big)^{-1}\big( \mathbf{I}_r - \eta \mathbf{\Theta}_t \mathbf{\Theta}_t^\top \big)\mathbf{\Omega}_t^\top \\
&\preceq \mathbf{\Omega}_t \mathbf{\Omega}_t^\top
\end{aligned}$$

where the last inequality comes from the fact that the three matrices in between are all PSD and their largest eigenvalues are smaller than 1 given our choices of $\eta$. This gives the proof for the first part of this Lemma.

To show $\sigma_{r_A}^2(\mathbf{\Phi}_{t+1}) \geq \sigma_{r_A}^2(\mathbf{\Phi}_t)$, notice that given $2r_A \leq m$, we have from Lemma 3 that $\sigma_{r_A}^2(\mathbf{\Phi}_t) = 1 - \sigma_{r_A}^2(\mathbf{\Omega}_t)$ and $\sigma_{r_A}^2(\mathbf{\Phi}_{t+1}) = 1 - \sigma_{r_A}^2(\mathbf{\Omega}_{t+1})$. The conclusion is straightforward. $\qquad \square$

#### D.5.2 Non-Increasing Misalignment of $\mathbf{Y}_t$

**Lemma 15.** *Denote the orthogonal complement of $\mathbf{V}$ as $\mathbf{V}_\perp \in \mathbb{R}^{n \times (n - r_A)}$. Define the $(n - r_A) \times r$ matrix $\tilde{\mathbf{\Omega}}_t := \mathbf{V}_\perp^\top \mathbf{Y}_t$ to characterize the alignment of $\mathbf{Y}_t$ and $\mathbf{V}_\perp$. Under the same setting of Lemma 13, we have that $\tilde{\mathbf{\Omega}}_{t+1}\tilde{\mathbf{\Omega}}_{t+1}^\top \preceq \tilde{\mathbf{\Omega}}_t \tilde{\mathbf{\Omega}}_t^\top$. Moreover, if $r_A \leq \frac{n}{2}$, it is guaranteed to have $\sigma_{r_A}^2(\mathbf{\Psi}_{t+1}) \geq \sigma_{r_A}^2(\mathbf{\Psi}_t)$.*

*Proof.* From update (5c), we have that

$$\begin{aligned}
\tilde{\mathbf{\Omega}}_{t+1} &= \mathbf{V}_\perp^\top (\mathbf{Y}_t - \eta \mathbf{F}_t)(\mathbf{I}_r + \eta^2 \mathbf{F}_t^\top \mathbf{F}_t)^{-1/2} \\
&= \big( \tilde{\mathbf{\Omega}}_t + \eta \mathbf{V}_\perp^\top (\mathbf{I}_n - \mathbf{Y}_t \mathbf{Y}_t^\top)\mathbf{A}^\top \mathbf{X}_t \mathbf{\Theta}_t \big)(\mathbf{I}_r + \eta^2 \mathbf{F}_t^\top \mathbf{F}_t)^{-1/2} \\
&= \big( \tilde{\mathbf{\Omega}}_t - \eta \mathbf{V}_\perp^\top \mathbf{Y}_t \mathbf{\Theta}_t^\top \mathbf{\Theta}_t \big)(\mathbf{I}_r + \eta^2 \mathbf{F}_t^\top \mathbf{F}_t)^{-1/2} \\
&= \tilde{\mathbf{\Omega}}_t \big( \mathbf{I}_r - \eta \mathbf{\Theta}_t^\top \mathbf{\Theta}_t \big)(\mathbf{I}_r + \eta^2 \mathbf{F}_t^\top \mathbf{F}_t)^{-1/2}.
\end{aligned}$$

With this, we can see that

$$\begin{aligned}
\tilde{\mathbf{\Omega}}_{t+1}\tilde{\mathbf{\Omega}}_{t+1}^\top &= \tilde{\mathbf{\Omega}}_t \big( \mathbf{I}_r - \eta \mathbf{\Theta}_t^\top \mathbf{\Theta}_t \big)\big( \mathbf{I}_r + \eta^2 \mathbf{F}_t^\top \mathbf{F}_t \big)^{-1}\big( \mathbf{I}_r - \eta \mathbf{\Theta}_t^\top \mathbf{\Theta}_t \big)\tilde{\mathbf{\Omega}}_t^\top \\
&\preceq \tilde{\mathbf{\Omega}}_t \tilde{\mathbf{\Omega}}_t^\top
\end{aligned}$$

where the last inequality comes from the fact that the three matrices in between are all PSD and their largest eigenvalues are smaller than 1 given our choice of $\eta$.

To show $\sigma_{r_A}^2(\boldsymbol{\Psi}_{t+1}) \geq \sigma_{r_A}^2(\boldsymbol{\Psi}_t)$, notice that given $2r_A \leq m$, we have from Lemma 3 that $\sigma_{r_A}^2(\boldsymbol{\Psi}_t) = 1 - \sigma_{r_A}^2(\tilde{\boldsymbol{\Omega}}_t)$ and $\sigma_{r_A}^2(\boldsymbol{\Psi}_{t+1}) = 1 - \sigma_{r_A}^2(\tilde{\boldsymbol{\Omega}}_{t+1})$. The conclusion is straightforward. □

### D.6 Convergence of $\mathsf{Tr}(\boldsymbol{\Phi}_t\boldsymbol{\Phi}_t^\top)$ and $\mathsf{Tr}(\boldsymbol{\Psi}_t\boldsymbol{\Psi}_t^\top)$

#### D.6.1 Dynamics of $\mathsf{Tr}(\boldsymbol{\Phi}_t\boldsymbol{\Phi}_t^\top)$

**Lemma 16.** *Suppose that $r_A \leq \frac{n}{2}$, and let $\rho := \min\{\frac{1}{m}, \frac{(r-r_A)^2}{mr}\}$. Choosing $\eta = \mathcal{O}\big(\frac{\rho(r-r_A)^2}{r^2\kappa^2 m}\big)$ and $\gamma = 1$, Alg. 1 guarantees that after at most $T = \mathcal{O}\big(\frac{r_A r^3 \kappa^4 m^2}{\rho^2(r-r_A)^4} + \frac{m^2 r^3 \kappa^4}{\rho(r-r_A)^4}\log\frac{1}{\epsilon}\big)$ steps $\mathsf{Tr}(\mathbf{I}_{r_A} - \boldsymbol{\Phi}_t\boldsymbol{\Phi}_t^\top) \leq \epsilon$.*

*Proof.* By rewriting (16), we arrive at

$$\mathsf{Tr}(\boldsymbol{\Phi}_{t+1}\boldsymbol{\Phi}_{t+1}^\top) - \mathsf{Tr}(\boldsymbol{\Phi}_t\boldsymbol{\Phi}_t^\top) \tag{20}$$
$$\geq \frac{2\eta\sigma_{r_A}^2(\boldsymbol{\Psi}_t)}{\kappa^2}(1 - \eta^2\beta_t)\mathsf{Tr}\big((\mathbf{I}_{r_A} - \boldsymbol{\Phi}_t\boldsymbol{\Phi}_t^\top)\boldsymbol{\Phi}_t\boldsymbol{\Phi}_t^\top\big) - \eta^2\beta_t\mathsf{Tr}(\boldsymbol{\Phi}_t\boldsymbol{\Phi}_t^\top).$$

Based on (20), we discuss the convergence in three different regimes.

**Phase I.** $\mathsf{Tr}(\mathbf{I}_{r_A} - \boldsymbol{\Phi}_t\boldsymbol{\Phi}_t^\top) \geq r_A - 0.5$. This is the initial phase, and the condition is equivalent to $\mathsf{Tr}(\boldsymbol{\Phi}_t\boldsymbol{\Phi}_t^\top) \leq 0.5$. For notational convenience, let the SVD of $\boldsymbol{\Phi}_t\boldsymbol{\Phi}_t^\top = \mathbf{Q}_t\boldsymbol{\Lambda}_t\mathbf{Q}_t^\top$. Given these conditions, it can be seen that $\sigma_{r_A}(\mathbf{I}_{r_A} - \boldsymbol{\Phi}_t\boldsymbol{\Phi}_t^\top) = \sigma_{r_A}(\mathbf{I}_{r_A} - \boldsymbol{\Lambda}_t) \geq 0.5$. Recalling that $\beta_t = \sigma_1(\mathbf{I}_{r_A} - \boldsymbol{\Phi}_t\boldsymbol{\Phi}_t^\top) \leq 1$, we can simplify (20) as

$$\mathsf{Tr}(\boldsymbol{\Phi}_{t+1}\boldsymbol{\Phi}_{t+1}^\top) - \mathsf{Tr}(\boldsymbol{\Phi}_t\boldsymbol{\Phi}_t^\top) \geq \frac{2\eta\sigma_{r_A}^2(\boldsymbol{\Psi}_t)}{\kappa^2}(1 - \eta^2)\mathsf{Tr}\big((\mathbf{I}_{r_A} - \boldsymbol{\Lambda}_t)\boldsymbol{\Lambda}_t\big) - \eta^2\mathsf{Tr}(\boldsymbol{\Phi}_t\boldsymbol{\Phi}_t^\top)$$
$$\overset{(a)}{\geq} \frac{\eta\sigma_{r_A}^2(\boldsymbol{\Psi}_t)}{\kappa^2}(1 - \eta^2)\mathsf{Tr}(\boldsymbol{\Lambda}_t) - \eta^2\mathsf{Tr}(\boldsymbol{\Phi}_t\boldsymbol{\Phi}_t^\top)$$
$$\overset{(b)}{\geq} \frac{\eta(r - r_A)^2}{\kappa^2 c_2 nr}(1 - \eta^2)\mathsf{Tr}(\boldsymbol{\Phi}_t\boldsymbol{\Phi}_t^\top) - \eta^2\mathsf{Tr}(\boldsymbol{\Phi}_t\boldsymbol{\Phi}_t^\top)$$

where $(a)$ uses $\sigma_{r_A}(\mathbf{I}_{r_A} - \boldsymbol{\Lambda}_t) \geq 0.5$; $(b)$ uses Lemmas 15 and 11, which jointly imply that $\sigma_{r_A}^2(\boldsymbol{\Psi}_t) \geq \sigma_{r_A}^2(\boldsymbol{\Psi}_0) \geq (r - r_A)^2/(c_2nr)$ for some universal constant $c_2$ defined in Lemma 11. Rearranging the terms, we arrive at

$$\mathsf{Tr}(\boldsymbol{\Phi}_{t+1}\boldsymbol{\Phi}_{t+1}^\top) \geq \left(1 + \frac{\eta(r - r_A)^2}{\kappa^2 c_2 nr}(1 - \eta^2) - \eta^2\right)\mathsf{Tr}(\boldsymbol{\Phi}_t\boldsymbol{\Phi}_t^\top)$$

which is linearly increasing once the term in parentheses is greater than 1. This amounts to choosing a small enough $\eta$, i.e., $\eta \leq \mathcal{O}\big(\frac{(r-r_A)^2}{\kappa^2 nr}\big)$.

**Phase II.** $0.5 < \mathsf{Tr}(\mathbf{I}_{r_A} - \boldsymbol{\Phi}_t\boldsymbol{\Phi}_t^\top) < r_A - 0.5$. Suppose that $\mathsf{Tr}\big((\mathbf{I}_{r_A} - \boldsymbol{\Lambda}_t)\boldsymbol{\Lambda}_t\big) \geq \rho$, for some $\rho > 0$ to be discussed shortly. Let $\eta = \mathcal{O}\big(\frac{\rho(r-r_A)^2}{r^2\kappa^2 m}\big)$ and $\eta \leq 0.5$, it is straightforward to have

$$\mathsf{Tr}(\boldsymbol{\Phi}_{t+1}\boldsymbol{\Phi}_{t+1}^\top) - \mathsf{Tr}(\boldsymbol{\Phi}_t\boldsymbol{\Phi}_t^\top) \geq \frac{2\eta(r - r_A)^2}{\kappa^2 c_2 nr}(1 - \eta^2)\mathsf{Tr}\big((\mathbf{I}_{r_A} - \boldsymbol{\Lambda}_t)\boldsymbol{\Lambda}_t\big) - \eta^2 r_A \tag{21}$$
$$\geq \frac{2\eta(r - r_A)^2}{\kappa^2 c_2 mr}(1 - \eta^2)\mathsf{Tr}\big((\mathbf{I}_{r_A} - \boldsymbol{\Lambda}_t)\boldsymbol{\Lambda}_t\big) - \eta^2 r$$
$$\geq \mathcal{O}\Big(\frac{\rho^2(r - r_A)^4}{r^3\kappa^4 m^2}\Big) := \Delta_1.$$

Note that the $\mathcal{O}(\cdot)$ notation ignores the dependence on constants including $c_1$ and $c_2$. This means that per step, $\mathsf{Tr}(\boldsymbol{\Phi}_t\boldsymbol{\Phi}_t^\top)$ increases at least by $\Delta_1$. Consequently, after at most $(r_A - 1)/\Delta_1 = \mathcal{O}\big(r_A r^3 \kappa^4 m^2/(\rho^2(r - r_A)^4)\big)$ iterations, RGD leaves Phase II.

Next, we show that $\rho \geq \mathcal{O}(\min\{\frac{1}{m}, \frac{(r-r_A)^2}{mr}\})$. Notice that $\text{Tr}((\mathbf{I}_{r_A} - \boldsymbol{\Lambda}_t)\boldsymbol{\Lambda}_t) \geq \sum_{i=1}^{r_A} \sigma_i^2(\boldsymbol{\Phi}_t)(1 - \sigma_i^2(\boldsymbol{\Phi}_t)) \geq \sigma_{r_A}^2(\boldsymbol{\Phi}_t)(1 - \sigma_{r_A}^2(\boldsymbol{\Phi}_t)) \geq \mathcal{O}(\min\{\frac{1}{m}, \frac{(r-r_A)^2}{mr}\})$, where the last inequality comes from the facts that i) for $x \in [a, b]$ with $0 < a < 0.5 < b < 1$, the smallest value of $x(1-x)$ is $\min\{a(1-a), b(1-b)\}$; and, ii) $\sigma_{r_A}^2(\boldsymbol{\Phi}_t)$ belongs to interval $[a, b]$ with $a = \mathcal{O}(\frac{(r-r_A)^2}{mr})$ and $b = \frac{r_A - 0.5}{r_A} = 1 - \frac{1}{2r_A} \leq 1 - \frac{1}{m}$. Lemmas 14 and 11 are adopted to calculate $a$, that is, $\sigma_{r_A}^2(\boldsymbol{\Phi}_t) \geq \sigma_{r_A}^2(\boldsymbol{\Phi}_0) = \mathcal{O}((r - r_A)^2/mr)$.

**Phase III.** $\text{Tr}(\mathbf{I}_{r_A} - \boldsymbol{\Phi}_t\boldsymbol{\Phi}_t^\top) \leq 0.5$. This is a regime near the optimum. An implication of this phase is that $\text{Tr}(\boldsymbol{\Phi}_t\boldsymbol{\Phi}_t^\top) \geq r_A - 0.5$. Given that the singular values of $\boldsymbol{\Phi}_t\boldsymbol{\Phi}_t^\top$ belong to $[0, 1]$, it can be seen that $\sigma_{r_A}(\boldsymbol{\Phi}_t\boldsymbol{\Phi}_t^\top) = \sigma_{r_A}(\boldsymbol{\Lambda}_t) \geq 0.5$. Together with $\beta_t \leq 0.5$ in this scenario, we can simplify (20) as

$$
\begin{aligned}
&\text{Tr}(\boldsymbol{\Phi}_{t+1}\boldsymbol{\Phi}_{t+1}^\top) - \text{Tr}(\boldsymbol{\Phi}_t\boldsymbol{\Phi}_t^\top) \\
&\geq \frac{2\eta\sigma_{r_A}^2(\boldsymbol{\Psi}_t)}{\kappa^2}\left(1 - \frac{\eta^2}{2}\right)\text{Tr}((\mathbf{I}_{r_A} - \boldsymbol{\Lambda}_t)\boldsymbol{\Lambda}_t) - \eta^2\beta_t\text{Tr}(\boldsymbol{\Phi}_t\boldsymbol{\Phi}_t^\top) \\
&\overset{(c)}{\geq} \frac{\eta(r-r_A)^2}{c_2 nr\kappa^2}\left(1 - \frac{\eta^2}{2}\right)\text{Tr}(\mathbf{I}_{r_A} - \boldsymbol{\Lambda}_t) - \eta^2\beta_t r_A \\
&= \frac{\eta(r-r_A)^2}{c_2 nr\kappa^2}\left(1 - \frac{\eta^2}{2}\right)\text{Tr}(\mathbf{I}_{r_A} - \boldsymbol{\Phi}_t\boldsymbol{\Phi}_t^\top) - \eta^2\beta_t r_A \\
&\overset{(d)}{\geq} \frac{\eta(r-r_A)^2}{c_2 nr\kappa^2}\left(1 - \frac{\eta^2}{2}\right)\text{Tr}(\mathbf{I}_{r_A} - \boldsymbol{\Phi}_t\boldsymbol{\Phi}_t^\top) - \eta^2 r_A\text{Tr}(\mathbf{I}_{r_A} - \boldsymbol{\Phi}_t\boldsymbol{\Phi}_t^\top)
\end{aligned}
$$

where $(c)$ uses $\sigma_{r_A}(\boldsymbol{\Lambda}_t) \geq 0.5$; and $(d)$ comes from $\beta_t \leq \text{Tr}(\mathbf{I}_{r_A} - \boldsymbol{\Phi}_t\boldsymbol{\Phi}_t^\top)$. This further implies that

$$
\begin{aligned}
&\text{Tr}(\mathbf{I}_{r_A} - \boldsymbol{\Phi}_{t+1}\boldsymbol{\Phi}_{t+1}^\top) - \text{Tr}(\mathbf{I}_{r_A} - \boldsymbol{\Phi}_t\boldsymbol{\Phi}_t^\top) \\
&\leq -\frac{\eta(r-r_A)^2}{c_2 nr\kappa^2}\left(1 - \frac{\eta^2}{2}\right)\text{Tr}(\mathbf{I}_{r_A} - \boldsymbol{\Phi}_t\boldsymbol{\Phi}_t^\top) + \eta^2 r_A\text{Tr}(\mathbf{I}_{r_A} - \boldsymbol{\Phi}_t\boldsymbol{\Phi}_t^\top).
\end{aligned}
$$

Reorganizing the terms, we arrive at

$$
\text{Tr}(\mathbf{I}_{r_A} - \boldsymbol{\Phi}_{t+1}\boldsymbol{\Phi}_{t+1}^\top) \leq \left(1 - \frac{\eta(r-r_A)^2}{c_2 nr\kappa^2}\left(1 - \frac{\eta^2}{2}\right) + \eta^2 r_A\right)\text{Tr}(\mathbf{I}_{r_A} - \boldsymbol{\Phi}_t\boldsymbol{\Phi}_t^\top). \tag{22}
$$

This indicates a linear rate until we achieve optimality once $\eta$ is chosen sufficiently small.

Note that our choice of $\eta$ ensures the conditions in Lemma 12 are satisfied, indicating that an increase of $\text{Tr}(\boldsymbol{\Phi}_t\boldsymbol{\Phi}_t^\top)$ per iteration is guaranteed. This means that $\text{Tr}(\boldsymbol{\Phi}_t\boldsymbol{\Phi}_t^\top)$ traverses Phase I, II, and III consecutively. Combining these three phases together gives the claimed complexity bound. $\qquad\square$

### D.6.2  Dynamics of $\text{Tr}(\boldsymbol{\Psi}_t\boldsymbol{\Psi}_t^\top)$

**Lemma 17.** *Suppose that $r_A \leq \frac{n}{2}$, and let $\rho := \min\{\frac{1}{m}, \frac{(r-r_A)^2}{mr}\}$. Choosing $\eta = \mathcal{O}(\frac{\rho(r-r_A)^2}{r^2\kappa^2 m})$ and $\gamma = 1$, Alg.1 guarantees that after at most $T = \mathcal{O}(\frac{r_A r^3\kappa^4 m^2}{\rho^2(r-r_A)^4} + \frac{m^2 r^3\kappa^4}{\rho(r-r_A)^4}\log\frac{1}{\epsilon})$ steps $\text{Tr}(\mathbf{I}_{r_A} - \boldsymbol{\Psi}_t\boldsymbol{\Psi}_t^\top) \leq \epsilon$.*

*Proof.* By rewriting (19), we arrive at

$$
\begin{aligned}
&\text{Tr}(\boldsymbol{\Psi}_{t+1}\boldsymbol{\Psi}_{t+1}^\top) - \text{Tr}(\boldsymbol{\Psi}_t\boldsymbol{\Psi}_t^\top) \\
&\geq \frac{2\eta\sigma_{r_A}^2(\boldsymbol{\Phi}_t)}{\kappa^2}(1 - \eta^2\delta_t)\text{Tr}((\mathbf{I}_{r_A} - \boldsymbol{\Psi}_t\boldsymbol{\Psi}_t^\top)\boldsymbol{\Psi}_t\boldsymbol{\Psi}_t^\top) - \eta^2\delta_t\text{Tr}(\boldsymbol{\Psi}_t\boldsymbol{\Psi}_t^\top).
\end{aligned} \tag{23}
$$

Based on (23), we discuss the convergence in three different regimes.

**Phase I.** $\text{Tr}(\mathbf{I}_{r_A} - \boldsymbol{\Psi}_t\boldsymbol{\Psi}_t^\top) \geq r_A - 0.5$. This is the initial phase, and the condition is equivalent to $\text{Tr}(\boldsymbol{\Psi}_t\boldsymbol{\Psi}_t^\top) \leq 0.5$. For notational convenience let the SVD of $\boldsymbol{\Psi}_t\boldsymbol{\Psi}_t^\top = \mathbf{P}_t\tilde{\boldsymbol{\Lambda}}_t\mathbf{P}_t^\top$. Given these

conditions, it can be seen that $\sigma_{r_A}(\mathbf{I}_{r_A} - \mathbf{\Psi}_t\mathbf{\Psi}_t^\top) = \sigma_{r_A}(\mathbf{I}_{r_A} - \tilde{\mathbf{\Lambda}}_t) \geq 0.5$. Together with $\delta_t \leq 1$ (recall that $\delta_t = \sigma_1(\mathbf{I}_{r_A} - \mathbf{\Psi}_t\mathbf{\Psi}_t^\top)$), we can simplify (23) as

$$
\begin{aligned}
\mathsf{Tr}(\mathbf{\Psi}_{t+1}\mathbf{\Psi}_{t+1}^\top) - \mathsf{Tr}(\mathbf{\Psi}_t\mathbf{\Psi}_t^\top) &\geq \frac{2\eta\sigma_{r_A}^2(\mathbf{\Phi}_t)}{\kappa^2}(1-\eta^2)\mathsf{Tr}\big((\mathbf{I}_{r_A} - \tilde{\mathbf{\Lambda}}_t)\tilde{\mathbf{\Lambda}}_t\big) - \eta^2\mathsf{Tr}(\mathbf{\Psi}_t\mathbf{\Psi}_t^\top) \\
&\overset{(a)}{\geq} \frac{\eta\sigma_{r_A}^2(\mathbf{\Phi}_t)}{\kappa^2}(1-\eta^2)\mathsf{Tr}(\tilde{\mathbf{\Lambda}}_t) - \eta^2\mathsf{Tr}(\mathbf{\Psi}_t\mathbf{\Psi}_t^\top) \\
&\overset{(b)}{\geq} \frac{\eta(r-r_A)^2}{\kappa^2 c_1 mr}(1-\eta^2)\mathsf{Tr}(\mathbf{\Psi}_t\mathbf{\Psi}_t^\top) - \eta^2\mathsf{Tr}(\mathbf{\Psi}_t\mathbf{\Psi}_t^\top)
\end{aligned}
$$

where $(a)$ uses $\sigma_{r_A}(\mathbf{I}_{r_A} - \tilde{\mathbf{\Lambda}}_t) \geq 0.5$; $(b)$ uses Lemmas 14 and 11, which jointly imply that $\sigma_{r_A}^2(\mathbf{\Phi}_t) \geq \sigma_{r_A}^2(\mathbf{\Phi}_0) \geq (r-r_A)^2/(c_1 mr)$ for some universal constant $c_1$ in defined in Lemma 11. Rearranging the terms, we arrive at

$$
\mathsf{Tr}(\mathbf{\Psi}_{t+1}\mathbf{\Psi}_{t+1}^\top) \geq \left(1 + \frac{\eta(r-r_A)^2}{\kappa^2 c_1 mr}(1-\eta^2) - \eta^2\right)\mathsf{Tr}(\mathbf{\Psi}_t\mathbf{\Psi}_t^\top)
$$

which is linearly increasing once the term in parentheses is greater than 1. This amounts to choosing a small enough $\eta$, i.e., $\eta \leq \mathcal{O}\big(\frac{(r-r_A)^2}{\kappa^2 mr}\big)$.

**Phase II.** $0.5 < \mathsf{Tr}(\mathbf{I}_{r_A} - \mathbf{\Psi}_t\mathbf{\Psi}_t^\top) < r_A - 0.5$. Suppose that $\mathsf{Tr}\big((\mathbf{I}_{r_A} - \mathbf{\Lambda}_t)\mathbf{\Lambda}_t\big) \geq \rho$, for some $\rho$ to be discussed shortly. Choosing $\eta \leq 0.5$, and $\eta = \mathcal{O}\big(\frac{\rho(r-r_A)^2}{r^2\kappa^2 m}\big)$, it is straightforward to have

$$
\begin{aligned}
\mathsf{Tr}(\mathbf{\Psi}_{t+1}\mathbf{\Psi}_{t+1}^\top) - \mathsf{Tr}(\mathbf{\Psi}_t\mathbf{\Psi}_t^\top) &\geq \frac{2\eta(r-r_A)^2}{\kappa^2 c_1 mr}(1-\eta^2)\mathsf{Tr}\big((\mathbf{I}_{r_A} - \tilde{\mathbf{\Lambda}}_t)\tilde{\mathbf{\Lambda}}_t\big) - \eta^2 r_A \qquad (24) \\
&\geq \frac{2\eta(r-r_A)^2}{\kappa^2 c_1 mr}(1-\eta^2)\mathsf{Tr}\big((\mathbf{I}_{r_A} - \tilde{\mathbf{\Lambda}}_t)\tilde{\mathbf{\Lambda}}_t\big) - \eta^2 r \\
&\geq \mathcal{O}\Big(\frac{\rho^2(r-r_A)^4}{r^3\kappa^4 m^2}\Big) := \Delta_2.
\end{aligned}
$$

Note that the $\mathcal{O}(\cdot)$ notation ignores the dependence on constants including $c_1$ and $c_2$. This means that per step, $\mathsf{Tr}(\mathbf{\Psi}_t\mathbf{\Psi}_t^\top)$ at least increases by $\Delta_2$. Consequently, after at most $(r_A - 1)/\Delta_2 = \mathcal{O}(r_A r^3\kappa^4 m^2/\rho^2(r - r_A)^4)$ iterations, RGD leaves Phase II.

Next, we show that $\rho \geq \mathcal{O}(\min\{\frac{1}{n}, \frac{(r-r_A)^2}{nr}\}) \geq \mathcal{O}(\min\{\frac{1}{m}, \frac{(r-r_A)^2}{mr}\})$. Notice that $\mathsf{Tr}\big((\mathbf{I}_{r_A} - \tilde{\mathbf{\Lambda}}_t)\tilde{\mathbf{\Lambda}}_t\big) \geq \sum_{i=1}^{r_A}\sigma_i^2(\mathbf{\Psi}_t)(1 - \sigma_i^2(\mathbf{\Psi}_t)) \geq \sigma_{r_A}^2(\mathbf{\Psi}_t)(1 - \sigma_{r_A}^2(\mathbf{\Psi}_t)) \geq \mathcal{O}(\min\{\frac{1}{n}, \frac{(r-r_A)^2}{nr}\})$, where the last inequality comes from the facts that i) for $x \in [a, b]$ with $0 < a < 0.5 < b < 1$, the smallest value of $x(1-x)$ is $\min\{a(1-a), b(1-b)\}$; and, ii) $\sigma_{r_A}^2(\mathbf{\Psi}_t)$ belongs to interval $[a, b]$ with $a = \mathcal{O}(\frac{(r-r_A)^2}{nr})$ and $b = \frac{r_A - 0.5}{r_A} = 1 - \frac{1}{2r_A} \leq 1 - \frac{1}{n}$. Lemmas 15 and 11 are adopted to calculate $a$, that is, $a = \sigma_{r_A}^2(\mathbf{\Psi}_t) \geq \sigma_{r_A}^2(\mathbf{\Psi}_0) = \mathcal{O}((r - r_A)^2/nr)$.

**Phase III.** $\mathsf{Tr}(\mathbf{I}_{r_A} - \mathbf{\Psi}_t\mathbf{\Psi}_t^\top) \leq 0.5$. This is a regime near the optimum. An implication of this phase is that $\mathsf{Tr}(\mathbf{\Psi}_t\mathbf{\Psi}_t^\top) \geq r_A - 0.5$. Given that the singular values of $\mathbf{\Psi}_t\mathbf{\Psi}_t^\top$ belong to $[0, 1]$, it can be seen that $\sigma_{r_A}(\mathbf{\Psi}_t\mathbf{\Psi}_t^\top) = \sigma_{r_A}(\tilde{\mathbf{\Lambda}}_t) \geq 0.5$. Together with $\delta_t \leq 0.5$ in this scenario, we can simplify (20) as

$$
\begin{aligned}
&\mathsf{Tr}(\mathbf{\Psi}_{t+1}\mathbf{\Psi}_{t+1}^\top) - \mathsf{Tr}(\mathbf{\Psi}_t\mathbf{\Psi}_t^\top) \\
&\geq \frac{2\eta\sigma_{r_A}^2(\mathbf{\Phi}_t)}{\kappa^2}\left(1 - \frac{\eta^2}{2}\right)\mathsf{Tr}\big((\mathbf{I}_{r_A} - \tilde{\mathbf{\Lambda}}_t)\tilde{\mathbf{\Lambda}}_t\big) - \eta^2\delta_t\mathsf{Tr}(\mathbf{\Psi}_t\mathbf{\Psi}_t^\top) \\
&\overset{(c)}{\geq} \frac{\eta(r-r_A)^2}{c_1 mr\kappa^2}\left(1 - \frac{\eta^2}{2}\right)\mathsf{Tr}(\mathbf{I}_{r_A} - \tilde{\mathbf{\Lambda}}_t) - \eta^2\delta_t r_A \\
&= \frac{\eta(r-r_A)^2}{c_1 mr\kappa^2}\left(1 - \frac{\eta^2}{2}\right)\mathsf{Tr}(\mathbf{I}_{r_A} - \mathbf{\Psi}_t\mathbf{\Psi}_t^\top) - \eta^2\delta_t r_A \\
&\overset{(d)}{\geq} \frac{\eta(r-r_A)^2}{c_1 mr\kappa^2}\left(1 - \frac{\eta^2}{2}\right)\mathsf{Tr}(\mathbf{I}_{r_A} - \mathbf{\Psi}_t\mathbf{\Psi}_t^\top) - \eta^2 r_A\mathsf{Tr}(\mathbf{I}_{r_A} - \mathbf{\Psi}_t\mathbf{\Psi}_t^\top)
\end{aligned}
$$

where $(c)$ comes from $\sigma_{r_A}(\tilde{\mathbf{\Lambda}}_t) \geq 0.5$, as well as $\sigma_{r_A}^2(\mathbf{\Phi}_t) \geq \sigma_{r_A}^2(\mathbf{\Phi}_0) \geq (r - r_A)^2/(c_1 mr)$; and $(d)$ uses $\delta_t \leq \mathsf{Tr}(\mathbf{I}_{r_A} - \mathbf{\Psi}_t\mathbf{\Psi}_t^\top)$. This further implies that

$$
\mathsf{Tr}(\mathbf{I}_{r_A} - \mathbf{\Psi}_{t+1}\mathbf{\Psi}_{t+1}^\top) - \mathsf{Tr}(\mathbf{I}_{r_A} - \mathbf{\Psi}_t\mathbf{\Psi}_t^\top)
$$
$$
\leq -\frac{\eta(r - r_A)^2}{c_1 mr\kappa^2}\left(1 - \frac{\eta^2}{2}\right)\mathsf{Tr}(\mathbf{I}_{r_A} - \mathbf{\Psi}_t\mathbf{\Psi}_t^\top) + \eta^2 r_A\mathsf{Tr}(\mathbf{I}_{r_A} - \mathbf{\Psi}_t\mathbf{\Psi}_t^\top).
$$

Reorganizing the terms, we arrive at

$$
\mathsf{Tr}(\mathbf{I}_{r_A} - \mathbf{\Psi}_{t+1}\mathbf{\Psi}_{t+1}^\top) \leq \left(1 - \frac{\eta(r - r_A)^2}{c_1 mr\kappa^2}\left(1 - \frac{\eta^2}{2}\right) + \eta^2 r_A\right)\mathsf{Tr}(\mathbf{I}_{r_A} - \mathbf{\Psi}_t\mathbf{\Psi}_t^\top).
$$

This indicates a linear rate until we achieve optimality once $\eta$ is chosen sufficiently small.

Note that our choice of $\eta$ ensures the conditions in Lemma 13 are satisfied. In other words, increasing $\mathsf{Tr}(\mathbf{\Psi}_t\mathbf{\Psi}_t^\top)$ across $t$ is guaranteed. This means that $\mathsf{Tr}(\mathbf{\Psi}_t\mathbf{\Psi}_t^\top)$ will traverse Phase I, II, and III consecutively. Combining these three phases together gives the claimed complexity bound. $\qquad\square$

### D.7 Convergence of $\mathbf{\Theta}_t$

**Lemma 18.** *Suppose that at iteration $t$, Alg. 1 with $\gamma = 1$ satisfies $\mathsf{Tr}(\mathbf{I}_{r_A} - \mathbf{\Phi}_t\mathbf{\Phi}_t^\top) \leq \rho_1$ and $\mathsf{Tr}(\mathbf{I}_{r_A} - \mathbf{\Psi}_t\mathbf{\Psi}_t^\top) \leq \rho_2$. It is guaranteed to have $f(\mathbf{X}_t, \mathbf{Y}_t, \mathbf{\Theta}_t) = \mathcal{O}(\rho_1 + \rho_2)$.*

*Proof.* Recall that $\gamma = 1$ implies $\mathbf{\Theta}_t = \mathbf{X}_t^\top \mathbf{A}\mathbf{Y}_t$, we thus have that

$$
\begin{aligned}
\|\mathbf{X}_t\mathbf{\Theta}_t\mathbf{Y}_t^\top - \mathbf{A}\|_\mathsf{F} &= \|\mathbf{X}_t\mathbf{X}_t^\top\mathbf{A}\mathbf{Y}_t\mathbf{Y}_t^\top - \mathbf{A}\|_\mathsf{F}\\
&= \|\mathbf{X}_t\mathbf{X}_t^\top\mathbf{A}\mathbf{Y}_t\mathbf{Y}_t^\top - \mathbf{A}\mathbf{Y}_t\mathbf{Y}_t^\top + \mathbf{A}\mathbf{Y}_t\mathbf{Y}_t^\top - \mathbf{A}\|_\mathsf{F}\\
&\leq \|(\mathbf{X}_t\mathbf{X}_t^\top - \mathbf{I}_m)\mathbf{A}\mathbf{Y}_t\mathbf{Y}_t^\top\|_\mathsf{F} + \|\mathbf{A}(\mathbf{Y}_t\mathbf{Y}_t^\top - \mathbf{I}_n)\|_\mathsf{F}\\
&\overset{(a)}{\leq} \|(\mathbf{X}_t\mathbf{X}_t^\top - \mathbf{I}_m)\mathbf{U}\|_\mathsf{F}\|\mathbf{\Sigma}\mathbf{V}^\top\mathbf{Y}_t\mathbf{Y}_t^\top\| + \|\mathbf{U}\mathbf{\Sigma}\|\|\mathbf{V}^\top(\mathbf{Y}_t\mathbf{Y}_t^\top - \mathbf{I}_n)\|_\mathsf{F}\\
&\leq \|\mathbf{\Sigma}\|\|(\mathbf{I}_m - \mathbf{X}_t\mathbf{X}_t^\top)\mathbf{U}\|_\mathsf{F} + \|\mathbf{\Sigma}\|\|\mathbf{V}^\top(\mathbf{I}_n - \mathbf{Y}_t\mathbf{Y}_t^\top)\|_\mathsf{F}
\end{aligned}
$$

where $(a)$ uses the compact SVD of $\mathbf{A} = \mathbf{U}\mathbf{\Sigma}\mathbf{V}^\top$. Now we have that

$$
\begin{aligned}
\|(\mathbf{I}_m - \mathbf{X}_t\mathbf{X}_t^\top)\mathbf{U}\|_\mathsf{F}^2 &= \mathsf{Tr}\left(\mathbf{U}^\top(\mathbf{I}_m - \mathbf{X}_t\mathbf{X}_t^\top)(\mathbf{I}_m - \mathbf{X}_t\mathbf{X}_t^\top)\mathbf{U}\right)\\
&= \mathsf{Tr}(\mathbf{I}_{r_A} - \mathbf{\Phi}_t\mathbf{\Phi}_t^\top) \leq \rho_1.
\end{aligned}
$$

Similarly, we also have

$$
\begin{aligned}
\|\mathbf{V}^\top(\mathbf{I}_n - \mathbf{Y}_t\mathbf{Y}_t^\top)\|_\mathsf{F}^2 &= \mathsf{Tr}\left(\mathbf{V}^\top(\mathbf{I}_n - \mathbf{Y}_t\mathbf{Y}_t^\top)(\mathbf{I}_n - \mathbf{Y}_t\mathbf{Y}_t^\top)\mathbf{V}\right)\\
&= \mathsf{Tr}(\mathbf{I}_{r_A} - \mathbf{\Psi}_t\mathbf{\Psi}_t^\top) \leq \rho_2.
\end{aligned}
$$

Combining these inequalities, we have that

$$
\|\mathbf{X}_t\mathbf{\Theta}_t\mathbf{Y}_t^\top - \mathbf{A}\|_\mathsf{F}^2 \overset{(b)}{\leq} 2\|\mathbf{\Sigma}\|^2\|(\mathbf{I}_m - \mathbf{X}\mathbf{X}^\top)\mathbf{U}\|_\mathsf{F}^2 + 2\|\mathbf{\Sigma}\|^2\|\mathbf{V}^\top(\mathbf{I}_n - \mathbf{Y}_t\mathbf{Y}_t^\top)\|_\mathsf{F}^2 = \mathcal{O}(\rho_1 + \rho_2)
$$

where $(b)$ uses $(a + b)^2 \leq 2a^2 + 2b^2$. This finishes the proof. $\qquad\square$

### D.8 Proof of Theorem 1

*Proof.* The proof is straightforward. We apply Lemma 16 and Lemma 17 to show that within $\mathcal{O}\left(\frac{m^2 r^3 r_A\kappa^4}{\rho^2(r - r_A)^4} + \frac{m^2 r^3\kappa^4}{\rho(r - r_A)^4}\log\frac{1}{\epsilon}\right)$ iterations, we have $\mathsf{Tr}(\mathbf{I}_{r_A} - \mathbf{\Psi}_t\mathbf{\Psi}_t^\top) \leq \epsilon$ and $\mathsf{Tr}(\mathbf{I}_{r_A} - \mathbf{\Phi}_t\mathbf{\Phi}_t^\top) \leq \epsilon$. Then, Lemma 18 is adopted to reach the conclusion. $\qquad\square$

# E Theoretical Extensions to Symmetric Matrix Factorization

This section extends our theoretical results to symmetric matrix factorization problems. While slightly deviating from the LoRA application, we still include these results because we believe the improvement of the $\epsilon$ and $\kappa$ dependence is important in its own right.

**Algorithm 3** RGD for PoLAR-parameterized (25)

---

**Require:** Learning rates $\eta, \gamma$; sample $\mathbf{X}_0$ uniformly from $\mathsf{St}(m, r)$.
   **for** $t = 0, \dots, T - 1$ **do**
      Find $\mathbf{\Theta}_t$ via (26a)
      Obtain Riemannian gradient $\mathbf{G}_t$ using (26b)
      Update $\mathbf{X}_{t+1}$ with (26c)
   **end for**

---

Let $\mathbf{B} \in \mathbb{R}^{m \times m}$ be a symmetric positive semi-definite matrix with $\mathsf{rank}(\mathbf{B}) = r_B$. Its compact SVD is given by $\mathbf{B} = \mathbf{U}\mathbf{\Sigma}\mathbf{U}^\top$, where $\mathbf{U} \in \mathbb{R}^{m \times r_B}$ and $\mathbf{\Sigma} \in \mathbb{R}^{r_B \times r_B}$. Without loss of generality, we assume $\sigma_1(\mathbf{\Sigma}) = 1$ and $\sigma_{r_B}(\mathbf{\Sigma}) = 1/\kappa$ with $\kappa$ denoting the condition number.

## E.1 Detailed Setups and Main Results

In the same spirit of the asymmetric problems (4), the standard BM factorization is reformulated as

$$\min_{\mathbf{X}, \mathbf{\Theta}} f_{\mathsf{sym}}(\mathbf{X}, \mathbf{\Theta}) = \frac{1}{2}\|\mathbf{X}\mathbf{\Theta}\mathbf{X}^\top - \mathbf{B}\|_{\mathsf{F}}^2, \quad \text{s.t.} \quad \mathbf{X} \in \mathsf{St}(m, r), \mathbf{\Theta} \in \mathbb{R}^{r \times r}. \tag{25}$$

We again consider the overparameterized setting where $r > r_B$.

The algorithm to solve (25) shares the same principles as Alg. 1. At iteration $t$, we first apply GD to update $\mathbf{\Theta}$ using $\nabla_{\mathbf{\Theta}} f(\mathbf{X}_t, \mathbf{\Theta}_{t-1})$, which gives

$$\mathbf{\Theta}_t = (1 - \gamma)\mathbf{\Theta}_{t-1} + \gamma \mathbf{X}_t^\top \mathbf{B}\mathbf{X}_t. \tag{26a}$$

Next, the Riemannian gradient (of $\mathbf{X}_t$) calculated at $(\mathbf{X}_t, \mathbf{\Theta}_t)$ is given by

$$\mathbf{G}_t = -(\mathbf{I}_m - \mathbf{X}_t\mathbf{X}_t^\top)\mathbf{B}\mathbf{X}_t\mathbf{X}_t^\top\mathbf{B}\mathbf{X}_t. \tag{26b}$$

With $\mathbf{G}_t$ and polar retraction, the update on $\mathbf{X}_t$ is given by

$$\mathbf{X}_{t+1} = (\mathbf{X}_t - \eta\mathbf{G}_t)(\mathbf{I}_r + \eta^2\mathbf{G}_t^\top\mathbf{G}_t)^{-1/2}. \tag{26c}$$

The detailed algorithm is summarized in Alg. 3.

**Main Results for the Overparameterized and Symmetric Problem.** We show that the PoLAR parameterization optimized with Alg. 3 can exponentially improve the dependence of $\epsilon$ compared with GD applied on BM.

**Theorem 2.** *Suppose that $r_B \leq \frac{m}{2}$, and let $\rho := \min\{\frac{1}{m}, \frac{(r-r_B)^2}{mr}\}$. Choosing $\eta = \mathcal{O}\big(\frac{\rho(r-r_B)^2}{r^2\kappa^2 m}\big)$ and $\gamma = 1$, Alg. 3 guarantees $f_{sym}(\mathbf{X}_T, \mathbf{\Theta}_T) \leq \epsilon$ for all $T \geq \mathcal{O}\big(\frac{r_B r^3 \kappa^4 m^2}{\rho^2(r-r_B)^4} + \frac{mr^2\kappa^4}{\rho(r-r_B)^2}\log\frac{1}{\epsilon}\big)$.*

Note that a lower bound of $\Omega(\frac{\kappa^2}{\sqrt{\epsilon}})$ applies to GD on the overparameterized matrix factorization problem with BM parameterization (Xiong et al., 2024). This shows that our PoLAR parameterization *exponentially* improves the $\epsilon$ dependence, transforming a sublinear rate into a linear one. Moreover, it can be seen that the convergence of Alg. 3 improves with overparameterization, since $r = cr_B$ induces a tighter dependence on $r$ compared to $r = r_B + c$ for proper $c > 0$.

## E.2 Proof of Theorem 2

### E.2.1 Dynamics

The alignment matrix is defined as $\mathbf{\Phi}_t := \mathbf{U}^\top\mathbf{X}_t \in \mathbb{R}^{r_B \times r}$. Next we derive several equations and inequalities that help to establish the main theorem. Note that the choice of learning rate $\gamma = 1$ will be leveraged in some equations. We start with

$$\mathbb{R}^{r \times r} \ni \mathbf{G}_t^\top\mathbf{G}_t = \mathbf{X}_t^\top\mathbf{B}\mathbf{X}_t\mathbf{X}_t^\top\mathbf{B}(\mathbf{I}_m - \mathbf{X}_t\mathbf{X}_t^\top)^2\mathbf{B}\mathbf{X}_t\mathbf{X}_t^\top\mathbf{B}\mathbf{X}_t$$
$$= \mathbf{X}_t^\top\mathbf{B}\mathbf{X}_t\mathbf{X}_t^\top\mathbf{B}(\mathbf{I}_m - \mathbf{X}_t\mathbf{X}_t^\top)\mathbf{B}\mathbf{X}_t\mathbf{X}_t^\top\mathbf{B}\mathbf{X}_t$$
$$= \mathbf{\Theta}_t\mathbf{\Phi}_t^\top\mathbf{\Sigma}(\mathbf{I}_{r_B} - \mathbf{\Phi}_t\mathbf{\Phi}_t^\top)\mathbf{\Sigma}\mathbf{\Phi}_t\mathbf{\Theta}_t$$

where we use $\mathbb{R}^{r \times r} \ni \mathbf{\Theta}_t = \mathbf{X}_t^\top \mathbf{B} \mathbf{X}_t = \mathbf{\Phi}_t^\top \mathbf{\Sigma} \mathbf{\Phi}_t$. Applying $\sigma_1(\mathbf{\Phi}_t) \leq 1$ and $\sigma_1(\mathbf{B}) = 1$ to the equation above, we have that

$$\sigma_1(\mathbf{G}_t^\top \mathbf{G}_t) \leq \sigma_1^4(\mathbf{\Sigma})\sigma_1(\mathbf{I}_{r_B} - \mathbf{\Phi}_t \mathbf{\Phi}_t^\top) = \sigma_1(\mathbf{I}_{r_B} - \mathbf{\Phi}_t \mathbf{\Phi}_t^\top). \tag{27}$$

We also have that

$$\begin{aligned}
\mathbf{\Phi}_{t+1} &= \left[ \mathbf{\Phi}_t + \eta(\mathbf{I}_{r_B} - \mathbf{\Phi}_t \mathbf{\Phi}_t^\top)\mathbf{\Sigma}\mathbf{\Phi}_t\mathbf{\Theta}_t \right]\left( \mathbf{I}_r + \eta^2 \mathbf{G}_t^\top \mathbf{G}_t \right)^{-1/2} \\
&= \left[ \mathbf{I}_{r_B} + \eta(\mathbf{I}_{r_B} - \mathbf{\Phi}_t \mathbf{\Phi}_t^\top)\mathbf{\Sigma}\mathbf{\Phi}_t\mathbf{\Phi}_t^\top \mathbf{\Sigma} \right]\mathbf{\Phi}_t\left( \mathbf{I}_r + \eta^2 \mathbf{G}_t^\top \mathbf{G}_t \right)^{-1/2}.
\end{aligned}$$

This gives that

$$\begin{aligned}
&\mathbf{\Phi}_{t+1}\mathbf{\Phi}_{t+1}^\top \tag{28} \\
&= \left[ \mathbf{I}_{r_B} + \eta(\mathbf{I}_{r_B} - \mathbf{\Phi}_t \mathbf{\Phi}_t^\top)\mathbf{\Sigma}\mathbf{\Phi}_t\mathbf{\Phi}_t^\top \mathbf{\Sigma} \right]\mathbf{\Phi}_t\left( \mathbf{I}_r + \eta^2 \mathbf{G}_t^\top \mathbf{G}_t \right)^{-1}\mathbf{\Phi}_t^\top \left[ \mathbf{I}_{r_B} + \eta\mathbf{\Sigma}\mathbf{\Phi}_t\mathbf{\Phi}_t^\top \mathbf{\Sigma}(\mathbf{I}_{r_B} - \mathbf{\Phi}_t \mathbf{\Phi}_t^\top) \right].
\end{aligned}$$

With these preparations, we are ready to prove our main results.

### E.2.2   Initialization

**Lemma 19.** *Suppose that $\mathbf{X}_0$ is uniformly sampled from $\mathsf{St}(m, r)$ using methods described in Lemma 6. There exists a universal constant $c_1$ such that whp the following hods*

$$\sigma_{r_B}(\mathbf{\Phi}_0) = \sigma_{r_B}(\mathbf{U}^\top \mathbf{X}_0) \geq \frac{r - r_B + 1}{\sqrt{c_1 mr}} \geq \frac{r - r_B}{\sqrt{c_1 mr}}.$$

*Proof.* The proofs, as well as the constants $c_1$, and the exact probability, follow directly from Lemma 9. $\qquad \square$

### E.2.3   Increasing Alignment in Symmetric Problems

**Lemma 20.** *Let $\beta_t := \sigma_1(\mathbf{I}_{r_B} - \mathbf{\Phi}_t \mathbf{\Phi}_t^\top)$. Assuming $\eta < 1$, $\gamma = 1$, and*

$$\frac{2(1 - \eta^2\beta_t)\sigma_{r_B}^2(\mathbf{\Phi}_t)}{\kappa^2}\mathsf{Tr}\left( (\mathbf{I}_{r_B} - \mathbf{\Phi}_t \mathbf{\Phi}_t^\top)\mathbf{\Phi}_t \mathbf{\Phi}_t^\top \right) \geq \eta\beta_t\mathsf{Tr}\left( \mathbf{\Phi}_t \mathbf{\Phi}_t^\top \right)$$

*holds, we have $\mathsf{Tr}(\mathbf{\Phi}_{t+1}\mathbf{\Phi}_{t+1}^\top) \geq \mathsf{Tr}(\mathbf{\Phi}_t \mathbf{\Phi}_t^\top)$.*

*Proof.* From (28), we have that

$$\begin{aligned}
&\mathbf{\Phi}_{t+1}\mathbf{\Phi}_{t+1}^\top \\
&= \left[ \mathbf{I}_{r_B} + \eta(\mathbf{I}_{r_B} - \mathbf{\Phi}_t \mathbf{\Phi}_t^\top)\mathbf{\Sigma}\mathbf{\Phi}_t\mathbf{\Phi}_t^\top \mathbf{\Sigma} \right]\mathbf{\Phi}_t\left( \mathbf{I}_r + \eta^2 \mathbf{G}_t^\top \mathbf{G}_t \right)^{-1}\mathbf{\Phi}_t^\top \left[ \mathbf{I}_{r_B} + \eta\mathbf{\Sigma}\mathbf{\Phi}_t\mathbf{\Phi}_t^\top \mathbf{\Sigma}(\mathbf{I}_{r_B} - \mathbf{\Phi}_t \mathbf{\Phi}_t^\top) \right] \\
&\overset{(a)}{\succeq} \left[ \mathbf{I}_{r_B} + \eta(\mathbf{I}_{r_B} - \mathbf{\Phi}_t \mathbf{\Phi}_t^\top)\mathbf{\Sigma}\mathbf{\Phi}_t\mathbf{\Phi}_t^\top \mathbf{\Sigma} \right]\mathbf{\Phi}_t\left( \mathbf{I}_r - \eta^2 \mathbf{G}_t^\top \mathbf{G}_t \right)\mathbf{\Phi}_t^\top \left[ \mathbf{I}_{r_B} + \eta\mathbf{\Sigma}\mathbf{\Phi}_t\mathbf{\Phi}_t^\top \mathbf{\Sigma}(\mathbf{I}_{r_B} - \mathbf{\Phi}_t \mathbf{\Phi}_t^\top) \right] \\
&\overset{(b)}{\succeq} (1 - \eta^2\beta_t)\left[ \mathbf{I}_{r_B} + \eta(\mathbf{I}_{r_B} - \mathbf{\Phi}_t \mathbf{\Phi}_t^\top)\mathbf{\Sigma}\mathbf{\Phi}_t\mathbf{\Phi}_t^\top \mathbf{\Sigma} \right]\mathbf{\Phi}_t\mathbf{\Phi}_t^\top \left[ \mathbf{I}_{r_B} + \eta\mathbf{\Sigma}\mathbf{\Phi}_t\mathbf{\Phi}_t^\top \mathbf{\Sigma}(\mathbf{I}_{r_B} - \mathbf{\Phi}_t \mathbf{\Phi}_t^\top) \right]
\end{aligned}$$

where $(a)$ is by Lemma 2; and $(b)$ is by $\left( \mathbf{I}_r - \eta^2 \mathbf{G}_t^\top \mathbf{G}_t \right) \succeq (1 - \eta^2\sigma_1(\mathbf{I}_{r_B} - \mathbf{\Phi}_t \mathbf{\Phi}_t^\top))\mathbf{I}$ as a result of (27), and we write $\beta_t := \sigma_1(\mathbf{I}_{r_B} - \mathbf{\Phi}_t \mathbf{\Phi}_t^\top)$ for convenience. Note that $\beta_t \in [0, 1]$.

Now let the thin SVD of $\mathbf{\Phi}_t$ be $\mathbf{Q}_t\mathbf{\Lambda}_t\mathbf{P}_t^\top$, where $\mathbf{Q}_t \in \mathbb{R}^{r_B \times r_B}$, $\mathbf{\Lambda}_t \in \mathbb{R}^{r_B \times r_B}$, and $\mathbf{P}_t \in \mathbb{R}^{r \times r_B}$. This gives that

$$\begin{aligned}
&\mathbf{\Phi}_{t+1}\mathbf{\Phi}_{t+1}^\top \\
&\overset{(c)}{\succeq} (1 - \eta^2\beta_t)\mathbf{Q}_t\left[ \mathbf{I}_{r_B} + \eta(\mathbf{I}_{r_B} - \mathbf{\Lambda}_t^2)\mathbf{S}_t\mathbf{\Lambda}_t^2\mathbf{S}_t \right]\mathbf{\Lambda}_t^2\left[ \mathbf{I}_{r_B} + \eta\mathbf{S}_t\mathbf{\Lambda}_t^2\mathbf{S}_t(\mathbf{I}_{r_B} - \mathbf{\Lambda}_t^2) \right]\mathbf{Q}_t^\top \\
&\succeq (1 - \eta^2\beta_t)\mathbf{Q}_t\left[ \mathbf{\Lambda}_t^2 + \eta(\mathbf{I}_{r_B} - \mathbf{\Lambda}_t^2)\mathbf{S}_t\mathbf{\Lambda}_t^2\mathbf{S}_t\mathbf{\Lambda}_t^2 + \eta\mathbf{\Lambda}_t^2\mathbf{S}_t\mathbf{\Lambda}_t^2\mathbf{S}_t(\mathbf{I}_{r_B} - \mathbf{\Lambda}_t^2) \right]\mathbf{Q}_t^\top
\end{aligned}$$

where in $(c)$ we denote $\mathbf{S}_t := \mathbf{Q}_t^\top \boldsymbol{\Sigma} \mathbf{Q}_t$ which is PD. Noticing that $1 - \eta^2 \beta_t \geq 0$, and taking the trace on both sides, we have that

$$\frac{1}{1 - \eta^2 \beta_t} \mathsf{Tr}(\boldsymbol{\Phi}_{t+1} \boldsymbol{\Phi}_{t+1}^\top) \tag{29}$$

$$\geq \mathsf{Tr}(\boldsymbol{\Phi}_t \boldsymbol{\Phi}_t^\top) + \eta \mathsf{Tr}\big((\mathbf{I}_{r_B} - \boldsymbol{\Lambda}_t^2) \mathbf{S}_t \boldsymbol{\Lambda}_t^2 \mathbf{S}_t \boldsymbol{\Lambda}_t^2 + \boldsymbol{\Lambda}_t^2 \mathbf{S}_t \boldsymbol{\Lambda}_t^2 \mathbf{S}_t (\mathbf{I}_{r_B} - \boldsymbol{\Lambda}_t^2)\big)$$

$$\overset{(d)}{\geq} \mathsf{Tr}(\boldsymbol{\Phi}_t \boldsymbol{\Phi}_t^\top) + \frac{2\eta \sigma_{r_B}(\boldsymbol{\Lambda}_t^2)}{\kappa^2} \mathsf{Tr}\big((\mathbf{I}_{r_B} - \boldsymbol{\Lambda}_t^2) \boldsymbol{\Lambda}_t^2\big)$$

$$= \mathsf{Tr}(\boldsymbol{\Phi}_t \boldsymbol{\Phi}_t^\top) + \frac{2\eta \sigma_{r_B}(\boldsymbol{\Lambda}_t^2)}{\kappa^2} \mathsf{Tr}\big((\mathbf{I}_{r_B} - \boldsymbol{\Phi}_t \boldsymbol{\Phi}_t^\top) \boldsymbol{\Phi}_t \boldsymbol{\Phi}_t^\top\big)$$

where $(d)$ is by Lemma 4 and Lemma 5. More precisely, we use $\sigma_{r_B}(\mathbf{S}_t \boldsymbol{\Lambda}_t^2 \mathbf{S}_t) \geq \sigma_{r_B}^2(\mathbf{S}_t) \sigma_{r_B}(\boldsymbol{\Lambda}_t^2) = \sigma_{r_B}(\boldsymbol{\Lambda}_t^2)/\kappa^2$. Simplifying the inequality finishes the proof. $\qquad \square$

### E.2.4 Non-Increasing Misalignment in Symmetric Problems

**Lemma 21.** *Denote the orthogonal complement of $\mathbf{U}$ be $\mathbf{U}_\perp \in \mathbb{R}^{m \times (m - r_B)}$. Define the $(m - r_B) \times r$ matrix $\boldsymbol{\Psi}_t := \mathbf{U}_\perp^\top \mathbf{X}_t$ to characterize the alignment of $\mathbf{X}_t$ and $\mathbf{U}_\perp$. Under the same setting of Lemma 20, we have that $\boldsymbol{\Psi}_{t+1} \boldsymbol{\Psi}_{t+1}^\top \preceq \boldsymbol{\Psi}_t \boldsymbol{\Psi}_t^\top$. Moreover, if $r_B \leq \frac{m}{2}$, it is guaranteed to have $\sigma_{r_B}^2(\boldsymbol{\Phi}_{t+1}) \geq \sigma_{r_B}^2(\boldsymbol{\Phi}_t)$.*

*Proof.* From update (26c), we have that

$$\boldsymbol{\Psi}_{t+1} = \mathbf{U}_\perp^\top (\mathbf{X}_t - \eta \mathbf{G}_t)(\mathbf{I}_r + \eta^2 \mathbf{G}_t^\top \mathbf{G}_t)^{-1/2}$$

$$= \big(\boldsymbol{\Psi}_t + \eta \mathbf{U}_\perp^\top (\mathbf{I}_m - \mathbf{X}_t \mathbf{X}_t^\top) \mathbf{B} \mathbf{X}_t \mathbf{X}_t^\top \mathbf{B} \mathbf{X}_t\big)(\mathbf{I}_r + \eta^2 \mathbf{G}_t^\top \mathbf{G}_t)^{-1/2}$$

$$= \big(\boldsymbol{\Psi}_t - \eta \mathbf{U}_\perp^\top \mathbf{X}_t \boldsymbol{\Theta}_t^2\big)(\mathbf{I}_r + \eta^2 \mathbf{G}_t^\top \mathbf{G}_t)^{-1/2}$$

$$= \boldsymbol{\Psi}_t \big(\mathbf{I}_r - \eta \boldsymbol{\Theta}_t^2\big)(\mathbf{I}_r + \eta^2 \mathbf{G}_t^\top \mathbf{G}_t)^{-1/2}.$$

With this, we can see that

$$\boldsymbol{\Psi}_{t+1} \boldsymbol{\Psi}_{t+1}^\top = \boldsymbol{\Psi}_t \big(\mathbf{I}_r - \eta \boldsymbol{\Theta}_t^2\big)(\mathbf{I}_r + \eta^2 \mathbf{G}_t^\top \mathbf{G}_t)^{-1} \big(\mathbf{I}_r - \eta \boldsymbol{\Theta}_t^2\big) \boldsymbol{\Psi}_t^\top$$

$$\preceq \boldsymbol{\Psi}_t \boldsymbol{\Psi}_t^\top$$

where the last inequality comes from the fact that the three matrices in between are all PSD and their largest eigenvalues are all smaller than 1 given our choices of $\eta$. This gives the proof for the first part of this lemma.

To show $\sigma_{r_B}^2(\boldsymbol{\Phi}_{t+1}) \geq \sigma_{r_B}^2(\boldsymbol{\Phi}_t)$, notice that given $2r_B \leq m$, we have from Lemma 3 that $\sigma_{r_B}^2(\boldsymbol{\Phi}_t) = 1 - \sigma_{r_B}^2(\boldsymbol{\Omega}_t)$ and $\sigma_{r_B}^2(\boldsymbol{\Phi}_{t+1}) = 1 - \sigma_{r_B}^2(\boldsymbol{\Psi}_{t+1})$. The conclusion is straightforward. $\qquad \square$

### E.2.5 Convergence

The proof of Theorem 2 is a direct result of the following Lemmas.

**Lemma 22.** *Suppose that $r_B \leq \frac{m}{2}$, and let $\rho := \min\{\frac{1}{m}, \frac{(r - r_B)^2}{mr}\}$. Choosing $\eta = \mathcal{O}\big(\frac{\rho(r - r_B)^2}{r^2 \kappa^2 m}\big)$ and $\gamma = 1$, we have that after at most $\mathcal{O}\big(\frac{r_B r^3 \kappa^4 m^2}{\rho^2 (r - r_B)^4} + \frac{mr^2 \kappa^4}{\rho(r - r_B)^2} \log \frac{1}{\epsilon}\big)$ steps, $\mathsf{Tr}(\mathbf{I}_{r_B} - \boldsymbol{\Phi}_t \boldsymbol{\Phi}_t^\top) \leq \epsilon$.*

*Proof.* By rewriting (29), we arrive at

$$\mathsf{Tr}(\boldsymbol{\Phi}_{t+1} \boldsymbol{\Phi}_{t+1}^\top) - \mathsf{Tr}(\boldsymbol{\Phi}_t \boldsymbol{\Phi}_t^\top) \tag{30}$$

$$\geq \frac{2\eta \sigma_{r_B}(\boldsymbol{\Lambda}_t^2)}{\kappa^2} (1 - \eta^2 \beta_t) \mathsf{Tr}\big((\mathbf{I}_{r_B} - \boldsymbol{\Lambda}_t^2) \boldsymbol{\Lambda}_t^2\big) - \eta^2 \beta_t \mathsf{Tr}(\boldsymbol{\Phi}_t \boldsymbol{\Phi}_t^\top).$$

Based on (30), we discuss the convergence in three different regimes.

**Phase I.** $\mathsf{Tr}(\mathbf{I}_{r_B} - \boldsymbol{\Phi}_t \boldsymbol{\Phi}_t^\top) \geq r_B - 0.5$. This is the initial phase, and the condition is equivalent to $\mathsf{Tr}(\boldsymbol{\Phi}_t \boldsymbol{\Phi}_t^\top) \leq 0.5$. Given these conditions, it can be seen that $\sigma_{\min}(\mathbf{I}_{r_B} - \boldsymbol{\Phi}_t \boldsymbol{\Phi}_t^\top) = \sigma_{\min}(\mathbf{I}_{r_B} - $

$\mathbf{\Lambda}_t^2) \geq 0.5$. Together with $\beta_t \leq 1$ (recall that $\beta_t := \sigma_1(\mathbf{I}_{r_B} - \mathbf{\Phi}_t\mathbf{\Phi}_t^\top)$), we can simplify (30) as

$$\mathsf{Tr}(\mathbf{\Phi}_{t+1}\mathbf{\Phi}_{t+1}^\top) - \mathsf{Tr}(\mathbf{\Phi}_t\mathbf{\Phi}_t^\top) \geq \frac{2\eta\sigma_{r_B}(\mathbf{\Lambda}_t^2)}{\kappa^2}(1-\eta^2)\mathsf{Tr}\big((\mathbf{I}_{r_B} - \mathbf{\Lambda}_t^2)\mathbf{\Lambda}_t^2\big) - \eta^2\mathsf{Tr}(\mathbf{\Phi}_t\mathbf{\Phi}_t^\top)$$

$$\overset{(a)}{\geq} \frac{\eta\sigma_{r_B}(\mathbf{\Lambda}_t^2)}{\kappa^2}(1-\eta^2)\mathsf{Tr}(\mathbf{\Lambda}_t^2) - \eta^2\mathsf{Tr}(\mathbf{\Phi}_t\mathbf{\Phi}_t^\top)$$

$$\overset{(b)}{\geq} \frac{\eta(r-r_B)^2}{\kappa^2 c_1 mr}(1-\eta^2)\mathsf{Tr}(\mathbf{\Phi}_t\mathbf{\Phi}_t^\top) - \eta^2\mathsf{Tr}(\mathbf{\Phi}_t\mathbf{\Phi}_t^\top)$$

where $(a)$ uses $\sigma_{\min}(\mathbf{I}_{r_B} - \mathbf{\Lambda}_t^2) \geq 0.5$; $(b)$ uses Lemmas 21 and 19, which implies that $\sigma_{r_B}(\mathbf{\Lambda}_t^2) \geq \sigma_{r_B}^2(\mathbf{\Phi}_0) \geq (r-r_B)^2/(c_1 mr)$ for some universal constant $c_1$ defined in Lemma 19. Rearranging the terms, we arrive at

$$\mathsf{Tr}(\mathbf{\Phi}_{t+1}\mathbf{\Phi}_{t+1}^\top) \geq \left(1 + \frac{\eta(r-r_B)^2}{\kappa^2 c_1 mr}(1-\eta^2) - \eta^2\right)\mathsf{Tr}(\mathbf{\Phi}_t\mathbf{\Phi}_t^\top)$$

which is linearly increasing once the term in parentheses is greater than 1. This amounts to choosing a small enough $\eta$, e.g., $\eta \leq \mathcal{O}(\frac{(r-r_B)^2}{\kappa^2 mr})$.

**Phase II.** $0.5 < \mathsf{Tr}(\mathbf{I}_{r_B} - \mathbf{\Phi}_t\mathbf{\Phi}_t^\top) < r_B - 0.5$. Suppose that $\mathsf{Tr}\big((\mathbf{I}_{r_B} - \mathbf{\Lambda}_t^2)\mathbf{\Lambda}_t^2\big) \geq \rho$, for some $\rho$ to be discussed shortly. Choosing $\eta \leq 0.5$, and $\eta = \mathcal{O}(\frac{\rho(r-r_B)^2}{r^2\kappa^2 m})$, it is straightforward to have

$$\mathsf{Tr}(\mathbf{\Phi}_{t+1}\mathbf{\Phi}_{t+1}^\top) - \mathsf{Tr}(\mathbf{\Phi}_t\mathbf{\Phi}_t^\top) \geq \frac{2\eta(r-r_B)^2}{\kappa^2 c_1 mr}(1-\eta^2)\mathsf{Tr}\big((\mathbf{I}_{r_B} - \mathbf{\Lambda}_t^2)\mathbf{\Lambda}_t^2\big) - \eta^2 r_B$$

$$\geq \frac{2\eta(r-r_B)^2}{\kappa^2 c_1 mr}(1-\eta^2)\mathsf{Tr}\big((\mathbf{I}_{r_B} - \mathbf{\Lambda}_t^2)\mathbf{\Lambda}_t^2\big) - \eta^2 r$$

$$\geq \mathcal{O}\left(\frac{\rho^2(r-r_B)^4}{r^3\kappa^4 m^2}\right) := \Delta.$$

This means that per step, $\mathsf{Tr}(\mathbf{\Phi}_t\mathbf{\Phi}_t^\top)$ increases at least by $\Delta$. Consequently, after at most $(r_B - 1)/\Delta = \mathcal{O}(r_B r^3 \kappa^4 m^2/\rho^2(r-r_B)^4)$ iterations, RGD leaves Phase II.

Next, we show that $\rho \geq \mathcal{O}(\min\{\frac{1}{m}, \frac{(r-r_B)^2}{mr}\})$. Notice that $\mathsf{Tr}\big((\mathbf{I}_{r_B} - \mathbf{\Lambda}_t^2)\mathbf{\Lambda}_t^2\big) \geq \sum_{i=1}^{r_B}\sigma_i^2(\mathbf{\Phi}_t)(1 - \sigma_i^2(\mathbf{\Phi}_t)) \geq \sigma_{r_B}^2(\mathbf{\Phi}_t)(1-\sigma_{r_B}^2(\mathbf{\Phi}_t)) \geq \mathcal{O}(\min\{\frac{1}{m}, \frac{(r-r_B)^2}{mr}\})$, where the last inequality comes from the facts that i) for $x \in [a,b]$ with $0 < a < 0.5 < b < 1$, the smallest value of $x(1-x)$ is $\min\{a(1-a), b(1-b)\}$; and, ii) $\sigma_{r_B}^2(\mathbf{\Phi}_t)$ belongs to interval $[a,b]$ with $a = \mathcal{O}(\frac{(r-r_B)^2}{mr})$ and $b = \frac{r_B - 0.5}{r_B} = 1 - \frac{1}{2r_B} \leq 1 - \frac{1}{m}$. Lemmas 21 and 19 are adopted to calculate $a$, that is, $\sigma_{r_B}^2(\mathbf{\Phi}_t) \geq \sigma_{r_B}^2(\mathbf{\Phi}_0) = \mathcal{O}((r-r_B)^2/mr)$.

**Phase III.** $\mathsf{Tr}(\mathbf{I}_{r_B} - \mathbf{\Phi}_t\mathbf{\Phi}_t^\top) \leq 0.5$. This is a regime near the optimum. An implication of this phase is that $\mathsf{Tr}(\mathbf{\Phi}_t\mathbf{\Phi}_t^\top) \geq r_B - 0.5$. Given that the singular values of $\mathbf{\Phi}_t\mathbf{\Phi}_t^\top$ belong to $[0,1]$, it can be seen that $\sigma_{r_B}(\mathbf{\Phi}_t\mathbf{\Phi}_t^\top) = \sigma_{r_B}(\mathbf{\Lambda}_t^2) \geq 0.5$. Together with $\beta_t \leq 0.5$ in this scenario, we can simplify (30) as

$$\mathsf{Tr}(\mathbf{\Phi}_{t+1}\mathbf{\Phi}_{t+1}^\top) - \mathsf{Tr}(\mathbf{\Phi}_t\mathbf{\Phi}_t^\top)$$

$$\geq \frac{2\eta\sigma_{r_B}(\mathbf{\Lambda}_t^2)}{\kappa^2}\left(1 - \frac{\eta^2}{2}\right)\mathsf{Tr}\big((\mathbf{I}_{r_B} - \mathbf{\Lambda}_t^2)\mathbf{\Lambda}_t^2\big) - \eta^2\beta_t\mathsf{Tr}(\mathbf{\Phi}_t\mathbf{\Phi}_t^\top)$$

$$\overset{(c)}{\geq} \frac{\eta}{2\kappa^2}\left(1 - \frac{\eta^2}{2}\right)\mathsf{Tr}(\mathbf{I}_{r_B} - \mathbf{\Lambda}_t^2) - \eta^2\beta_t r_B$$

$$= \frac{\eta}{2\kappa^2}\left(1 - \frac{\eta^2}{2}\right)\mathsf{Tr}(\mathbf{I}_{r_B} - \mathbf{\Phi}_t\mathbf{\Phi}_t^\top) - \eta^2\beta_t r_B$$

$$\overset{(d)}{\geq} \frac{\eta}{2\kappa^2}\left(1 - \frac{\eta^2}{2}\right)\mathsf{Tr}(\mathbf{I}_{r_B} - \mathbf{\Phi}_t\mathbf{\Phi}_t^\top) - \eta^2 r_B\mathsf{Tr}(\mathbf{I}_{r_B} - \mathbf{\Phi}_t\mathbf{\Phi}_t^\top)$$

where $(c)$ applies $\sigma_{r_B}(\mathbf{\Lambda}_t^2) \geq 0.5$ twice; and $(d)$ follows from $\beta_t \leq \mathsf{Tr}(\mathbf{I}_{r_B} - \mathbf{\Phi}_t\mathbf{\Phi}_t^\top)$. This further implies that

$$\mathsf{Tr}(\mathbf{I}_{r_B} - \mathbf{\Phi}_{t+1}\mathbf{\Phi}_{t+1}^\top) - \mathsf{Tr}(\mathbf{I}_{r_B} - \mathbf{\Phi}_t\mathbf{\Phi}_t^\top)$$
$$\leq -\frac{\eta}{2\kappa^2}\left(1 - \frac{\eta^2}{2}\right)\mathsf{Tr}(\mathbf{I}_{r_B} - \mathbf{\Phi}_t\mathbf{\Phi}_t^\top) + \eta^2 r_B \mathsf{Tr}(\mathbf{I}_{r_B} - \mathbf{\Phi}_t\mathbf{\Phi}_t^\top).$$

Reorganizing the terms, we arrive at

$$\mathsf{Tr}(\mathbf{I}_{r_B} - \mathbf{\Phi}_{t+1}\mathbf{\Phi}_{t+1}^\top) \leq \left(1 - \frac{\eta}{2\kappa^2}\left(1 - \frac{\eta^2}{2}\right) + \eta^2 r_B\right)\mathsf{Tr}(\mathbf{I}_{r_B} - \mathbf{\Phi}_t\mathbf{\Phi}_t^\top).$$

This indicates a linear rate until we achieve optimality once $\eta$ is chosen sufficiently small.

Note that our choice of $\eta$ ensures the conditions in Lemma 20 are satisfied, guaranteeing an increase of $\mathsf{Tr}(\mathbf{\Phi}_t\mathbf{\Phi}_t^\top)$ in every iteration. This means that the $\mathsf{Tr}(\mathbf{\Phi}_t\mathbf{\Phi}_t^\top)$ will traverse Phase I, II, and III consecutively. Combining these three phases together gives the claimed complexity bound. $\qquad\square$

**Lemma 23.** *Alg.* 3 *with* $\gamma = 1$ *guarantees that* $f(\mathbf{X}_t, \mathbf{\Theta}_t) = \mathcal{O}(\epsilon)$ *if* $\mathsf{Tr}(\mathbf{I}_{r_B} - \mathbf{\Phi}_t\mathbf{\Phi}_t^\top) \leq \epsilon$ *is satisfied.*

*Proof.* We ignore the subscript in $\mathbf{X}_t$ and $\mathbf{\Theta}_t = \mathbf{X}_t^\top \mathbf{B}\mathbf{X}_t$ for simplicity. We have that

$$\|\mathbf{X}\mathbf{\Theta}\mathbf{X}^\top - \mathbf{B}\|_\mathsf{F} = \|\mathbf{X}\mathbf{X}^\top \mathbf{B}\mathbf{X}\mathbf{X}^\top - \mathbf{B}\|_\mathsf{F}$$
$$= \|\mathbf{X}\mathbf{X}^\top \mathbf{B}\mathbf{X}\mathbf{X}^\top - \mathbf{B}\mathbf{X}\mathbf{X}^\top + \mathbf{B}\mathbf{X}\mathbf{X}^\top - \mathbf{B}\|_\mathsf{F}$$
$$\leq \|(\mathbf{X}\mathbf{X}^\top - \mathbf{I}_m)\mathbf{B}\mathbf{X}\mathbf{X}^\top\|_\mathsf{F} + \|\mathbf{B}(\mathbf{X}\mathbf{X}^\top - \mathbf{I}_m)\|_\mathsf{F}$$
$$\overset{(a)}{\leq} \|(\mathbf{X}\mathbf{X}^\top - \mathbf{I}_m)\mathbf{U}\|_\mathsf{F}\|\mathbf{\Sigma}\mathbf{U}^\top\mathbf{X}\mathbf{X}^\top\| + \|\mathbf{U}\mathbf{\Sigma}\|\|\mathbf{U}^\top(\mathbf{X}\mathbf{X}^\top - \mathbf{I}_m)\|_\mathsf{F}$$
$$\overset{(b)}{\leq} 2\|\mathbf{\Sigma}\|\|(\mathbf{I}_m - \mathbf{X}\mathbf{X}^\top)\mathbf{U}\|_\mathsf{F}.$$

Now since we have that

$$\|(\mathbf{I}_m - \mathbf{X}\mathbf{X}^\top)\mathbf{U}\|_\mathsf{F}^2 = \mathsf{Tr}\left(\mathbf{U}^\top(\mathbf{I}_m - \mathbf{X}\mathbf{X}^\top)(\mathbf{I}_m - \mathbf{X}\mathbf{X}^\top)\mathbf{U}\right)$$
$$= \mathsf{Tr}(\mathbf{I}_{r_B} - \mathbf{\Phi}_t\mathbf{\Phi}_t^\top) \leq \epsilon$$

Combining above inequalities, we have that

$$\|\mathbf{X}\mathbf{\Theta}\mathbf{X}^\top - \mathbf{B}\|_\mathsf{F}^2 \leq 4\|\mathbf{\Sigma}\|^2\|(\mathbf{I}_m - \mathbf{X}\mathbf{X}^\top)\mathbf{U}\|_\mathsf{F}^2 \leq 4\epsilon.$$

This finishes the proof. $\qquad\square$

# F    Additional Experiments

## F.1    Additional Analysis on Stable Rank

In addition to the low stable rank in commonsense reasoning tasks shown in Fig. 2, we analyze the *official* LoRA checkpoints from Hu et al. (2022) for DeBERTa-XXL.[4] The results are shown in Fig. 6. It demonstrates that for most tasks the median stable rank of LoRA is in $[1.30, 1.40]$ even if $r$ is chosen as 16 in LoRA. This implies that for half of the layers the stable rank is at the lower end, leading to low directional diversity as outlined in the main text.

We also provide more comprehensive results on the relationship between the stable and nominal rank for LoRA and PoLAR fine-tuned on commonsense reasoning tasks. Fig. 5 confirms the argument made in the main text that LoRA suffers from a low effective rank which is addressed by our PoLAR parameterization. This applies to all commonsense reasoning datasets considered with the exception of ARC-e and ARC-c. Moreover, the stable rank of PoLAR increases with $r$ on all datasets, yet the trend is unclear for LoRA.

---

[4]Available at `https://github.com/microsoft/LoRA`.

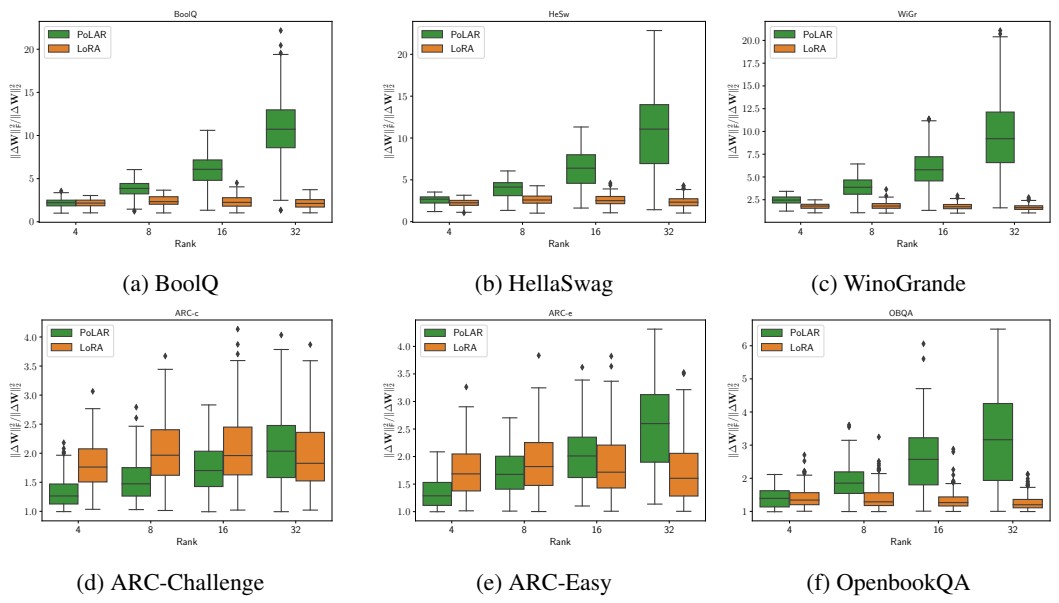

(a) BoolQ        (b) HellaSwag        (c) WinoGrande

(d) ARC-Challenge        (e) ARC-Easy        (f) OpenbookQA

Figure 5: Stable rank of $\Delta\mathbf{W}$ for Llama2-7B fine-tuned on commonsense reasoning tasks.

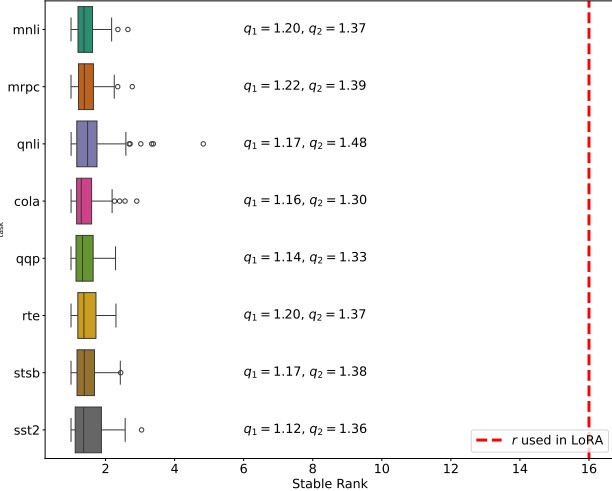

Figure 6: Stable rank of the official DeBERTa-XXL LoRA checkpoints trained on GLUE tasks with $r = 16$. $q_1$ and $q_2$ denote the first quartile and median, respectively.

In Fig. 7, we show the evolution of $\mathsf{sr}(\Delta\mathbf{W})$ over the course of training for PoLAR and LoRA at $r = 32$. While the stable rank for LoRA does not deviate significantly from the stable rank early in training, PoLAR's stable rank appears to dynamically vary with the layer type: more often than not the MLP layers (up and down projection) appear to require a larger stable rank, while the stable rank of the value projection layer often comes out at the lowest.

### F.2    Runtime Comparison

**Benchmarking against Retraction.**    We provide the experimental details for the runtime comparison of a landing step and retraction step based on polar retraction in Table 4. For the retraction step, we re-implement the polar retraction from Boumal et al. (2014) in PyTorch and compare it to our parameter update based on a landing step. We use `torch.compile` to pre-compile each function and run the benchmarking on a single NVIDIA H100 device. We sample measurements until the ratio of interquartile range to median is below $0.15$ and report the median. We provide detailed results for

Table 6: Reserved memory, average step time, and total time of training different low-rank adapters for Llama-2-7B on PIQA with $r = 32$. A single NVIDIA H100 GPU was used.

| Method | Mem. (GB) | Avg. Step Time (s) | Total Time (min) |
|--------|-----------|--------------------|--------------------|
| LoRA   | 52.37     | 2.72               | 28.74              |
| DoRA   | 67.05     | 5.71               | 60.38              |
| PoLAR  | 52.68     | 3.13               | 33.13              |

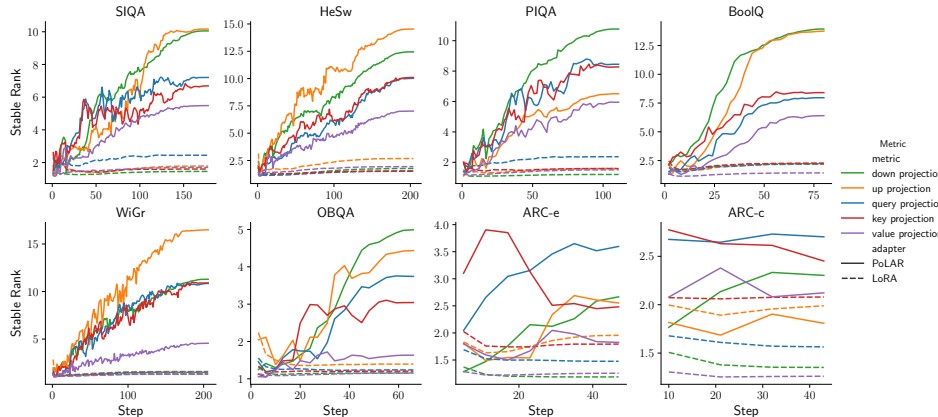

Figure 7: Stable rank evolution of the low-rank updates for the layers of the 5th transformer block of Llama-2-7B.

additional values of $m$ in Table 7 below, where the choice of $m$ comes from the dimensions of linear layers of Llama models.

**Benchmarking against LoRA/DoRA.** We compare runtime and memory usage of PoLAR to that of LoRA and DoRA by reporting average step time and reserved memory usage in Table 6. We record memory usage with the PyTorch profiler and measure step time using the HuggingFace Trainer. We remark that PoLAR only slightly increases memory usage relative to LoRA. The step time overhead of PoLAR over LoRA is at approximately 15%.

### F.3 Approximate Feasibility

In Fig. 8, we display the Frobenius norm distance to the Stiefel manifold for $\mathbf{X}$ and $\mathbf{Y}$ of PoLAR as well as $\mathbf{Z}_1$ and $\mathbf{Z}_2$ of LoRA. The displayed values refer to averages across all layers. The plot provides empirical evidence that PoLAR ensures (i) approximate feasibility of $\mathbf{X}$ and $\mathbf{Y}$ throughout training and (ii) convergence to the Stiefel manifold.

Table 7: Retraction (Retr.) and landing (Land.) times (in $\mu s$) for different $(m, r)$.

| $m$ | | 4 | 32 | 64 | 256 |
|-----|-------|------|------|------|-------|
|     |       |      | $r$  |      |       |
| 768 | Retr. | 507  | 1160 | 2178 | 8317  |
|     | Land. | 167  | 100  | 102  | 109   |
| 4096 | Retr. | 541 | 1050 | 1916 | 9889  |
|      | Land. | 180 | 186  | 238  | 539   |
| 11008 | Retr. | 606 | 1312 | 2237 | 11653 |
|       | Land. | 1785 | 1233 | 1632 | 3606 |

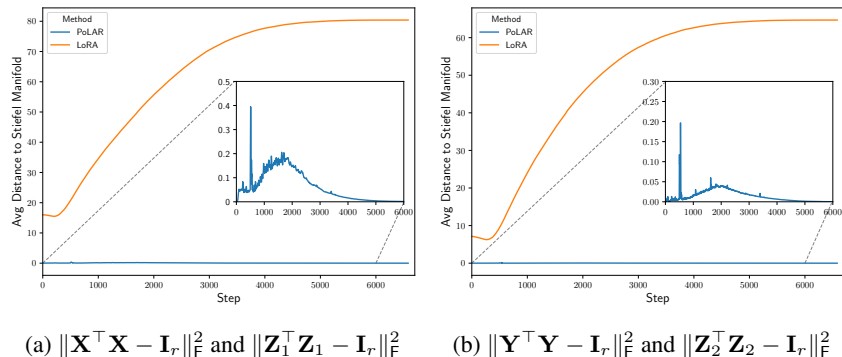

(a) $\|\mathbf{X}^\top \mathbf{X} - \mathbf{I}_r\|_\mathsf{F}^2$ and $\|\mathbf{Z}_1^\top \mathbf{Z}_1 - \mathbf{I}_r\|_\mathsf{F}^2$      (b) $\|\mathbf{Y}^\top \mathbf{Y} - \mathbf{I}_r\|_\mathsf{F}^2$ and $\|\mathbf{Z}_2^\top \mathbf{Z}_2 - \mathbf{I}_r\|_\mathsf{F}^2$

Figure 8: Average distance to Stiefel manifold across all LoRA and PoLAR modules in Gemma-3-27B on MetaMathQA.

Table 8: Performance on commonsense reasoning tasks with Llama-2-7B using GaLore and PoLAR for rank 16 in a single-task setup. HeSw refers to HellaSwag and WiGr to WinoGrande.

|  | $\eta$ | BoolQ | PIQA | SIQA | HeSw | WiGr | ARC-e | ARC-c | OBQA | Avg. |
|---|---|---|---|---|---|---|---|---|---|---|
| GaLore | $1 \times 10^{-5}$ | 81.93 | 80.20 | 56.40 | 80.23 | 81.53 | 80.22 | 47.53 | 44.80 | 69.10 |
|  | $4 \times 10^{-5}$ | 86.48 | 82.81 | 61.26 | 83.84 | 78.85 | 83.08 | 54.44 | 57.80 | 73.57 |
| PoLAR |  | 88.01 | 82.92 | 59.93 | 82.68 | 81.77 | 81.82 | 57.51 | 57.40 | 74.00 |

## F.4 Comparison with GaLore

We compare PoLAR to GaLore (Zhao et al., 2024) in Table 8. We adopt our commonsense reasoning training setup from Table 2 with rank 16. For GaLore's hyperparameters, we take inspiration from their fine-tuning experiments in Appendix D and set the scale to 2. We set the update projection gap s.t. the projections are updated once 15% increments of training are completed and tune the learning rate $\eta \in \{1 \times 10^{-5}, 4 \times 10^{-5}\}$.

## F.5 Gemma-3-12B on Commonsense Reasoning

In Table 9, we evaluate PoLAR on Gemma-3 on the commonsense reasoning benchmark, replicating the setting described Appendix G.1 with the difference of setting $\lambda = 5 \times 10^{-3}$, $r = 8$, and using Gemma-3-12B.

## G  Experimental Setup

Experiments are performed on either of NVIDIA GH200 and NVIDIA H100 GPUs. We use *PyTorch* (Paszke et al., 2019) for all experiments.

Table 9: Performance on commonsense-reasoning tasks with Gemma-3-12B using LoRA and PoLAR at rank 8 for different learning rates ($\eta$). HeSw refers to HellaSwag and WiGr to WinoGrande.

|  | $\eta$ | BoolQ | PIQA | SIQA | HeSw | WiGr | ARC-e | ARC-c | OBQA | Avg. |
|---|---|---|---|---|---|---|---|---|---|---|
| LoRA | $4 \times 10^{-4}$ | 90.03 | 84.33 | 59.77 | 86.25 | 79.40 | 88.55 | 70.90 | 63.60 | 77.86 |
|  | $8 \times 10^{-4}$ | 90.43 | 82.43 | 57.68 | 83.70 | 77.35 | 87.79 | 69.28 | 63.00 | 76.46 |
| PoLAR | $4 \times 10^{-4}$ | 89.72 | 85.53 | 61.87 | 87.19 | 86.50 | 90.11 | 65.53 | 66.00 | 79.06 |
|  | $8 \times 10^{-4}$ | 89.54 | 85.58 | 61.87 | 86.58 | 86.50 | 90.61 | 66.81 | 65.80 | 79.16 |

### G.1 Commonsense Reasoning

We consider the following tasks: BoolQ (Clark et al., 2019), PIQA (Bisk et al., 2019), SIQA (Sap et al., 2019), HellaSwag (Zellers et al., 2019), WinoGrande (Sakaguchi et al., 2019), ARC-e and ARC-c (Clark et al., 2018), and OpenbookQA (Mihaylov et al., 2018). To facilitate reproducibility, we use Eleuther-AI's *lm-evaluation-harness* (Biderman et al., 2024) and report the accuracy based on multiple-choice log-likelihood evaluation, i.e., we select the answer choice with the highest conditional log-likelihood as the predicted answer. For datasets with answer choices of varying length (PIQA, ARC-e, ARC-c, OpenbookQA, and Hellaswag), we perform byte-length normalization of the log-likelihood scores to remove any bias due to the answer length.

For the results in Table 10, we train for 5 epochs on each task with batch size 128 and choose the learning rate within $\{4 \times 10^{-4}, 8 \times 10^{-4}, 4 \times 10^{-3}\}$. We tune $\lambda \in \{10^{-3}, 5 \times 10^{-3}\}$ for PoLAR and set $\alpha = 32$. We choose the combination that performs best on average throughout all datasets (see Table 10) and report these.

For the comparison in Table 1, we train again on each task for 5 epochs with $\eta = 4 \times 10^{-3}$ and set $\lambda = 0.1$ for all datasets and settings.

### G.2 Mathematical Reasoning

For the mathematical reasoning experiment, we tune the learning rate grid in $\{2 \times 10^{-4}, 4 \times 10^{-4}, 6 \times 10^{-4}\}$. We train for 2 epochs on MetaMathQA (Yu et al., 2024) using rank 16, batch size 128, and tune $\lambda \in \{10^{-3}, 5 \times 10^{-3}\}$. Further following Yu et al. (2024), we do not use few-shot prompting, as the zero-shot setup reduces inference costs and tends to work better for fine-tuned models (see Appendix B in Yu et al. (2024)). We use the following Alpaca-based prompt (Taori et al., 2023) for training and evaluation:

```
"Below is an instruction that describes a task.  Write a
response that appropriately completes the request.\n\n###
Instruction:\n{instruction}\n\n### Response:\n".
```
We evaluate on GSM8K (Cobbe et al., 2021) and MATH (Hendrycks et al., 2021) using lm-evaluation-harness and the above prompt.

### G.3 GLUE

We set the rank $r = 16$ and batch size 32 for all methods. Additionally, we fix $\alpha = 8$ and $\lambda = 1$. For both methods, we do a grid-search of the learning rate $\eta \in \{4 \times 10^{-4}, 8 \times 10^{-4}, 1 \times 10^{-3}\}$ and report the set of runs that yields best performance across all datasets. We report the average accuracy across four seeds.

### G.4 Matrix Factorization

Here, we provide the experimental details for Fig. 3. We compare Riemannian Gradient Descent (RGD) with the PoLAR parameterization (PO) to Gradient Descent (GD) with the Burer-Monteiro parameterization (BM). We set $\mathbf{A} \in \mathbb{R}^{m \times n}$ with $r_A = 4$ evenly spaced singular values in the interval $[1, \kappa]$ with $\kappa \in \{10, 100\}$. For $\kappa = 10$, we set $\eta = 10^{-3}$ and $\gamma = 1$ for PO + RGD and $\eta = 10^{-3}$ for BM + GD. For $\kappa = 100$, we use $\eta = 10^{-4}$ for both methods. In the symmetric case, we use again $\eta = 10^{-3}$ for $\kappa = 10$ and $\eta = 10^{-5}$ for $\kappa = 100$.

### G.5 Licenses

Our evaluations are carried out on commonly-used datasets in the literature.

**Language Understanding.** GLUE (Wang et al., 2019) is designed to provide a general-purpose evaluation of language understanding. Those adopted in our work include MNLI (inference, Williams et al. (2018)), SST-2 (sentiment analysis, Socher et al. (2013)), MRPC (paraphrase detection, Dolan and Brockett (2005)), CoLA (linguistic acceptability, Warstadt et al. (2019)), QNLI (inference Rajpurkar et al. (2018)), QQP[5] (question-answering), RTE[6] (inference), and STS-B (textual similarity, Cer et al. (2017)). These datasets are released under different permissive licenses.

---

[5] https://quoradata.quora.com/First-Quora-Dataset-Release-Question-Pairs
[6] https://paperswithcode.com/dataset/rte

Table 10: Learning rate and regularization lambda values for different ranks and adapters.

| Rank | Adapter | Learning Rate | $\lambda$ | $\alpha$ |
|------|---------|---------------|-----------|----------|
| 4    | DoRA    | $4 \times 10^{-4}$ |           | 32 |
|      | LoRA    | $4 \times 10^{-4}$ |           | 32 |
|      | PoLAR   | $4 \times 10^{-3}$ | $5 \times 10^{-3}$ | 32 |
| 8    | DoRA    | $4 \times 10^{-4}$ |           | 32 |
|      | LoRA    | $4 \times 10^{-4}$ |           | 32 |
|      | PoLAR   | $4 \times 10^{-3}$ | $5 \times 10^{-3}$ | 32 |
| 16   | DoRA    | $4 \times 10^{-4}$ |           | 32 |
|      | LoRA    | $4 \times 10^{-4}$ |           | 32 |
|      | PoLAR   | $4 \times 10^{-3}$ | $5 \times 10^{-3}$ | 32 |
| 32   | DoRA    | $4 \times 10^{-4}$ |           | 32 |
|      | LoRA    | $4 \times 10^{-4}$ |           | 32 |
|      | PoLAR   | $4 \times 10^{-3}$ | $5 \times 10^{-3}$ | 32 |

**Commonsense Reasoning.** These datasets are a collection tasks that require commonsense reasoning to answer. The considered datasets include WinoGrande (Sakaguchi et al., 2019), PIQA (Bisk et al., 2019), Social Interaction QA (SIQA) (Sap et al., 2019), HellaSwag (Zellers et al., 2019), ARC-easy, ARC-challenge (Chollet, 2019) and OpenbookQA (Mihaylov et al., 2018). These datasets are released under different permissive licenses.

**Mathematical Reasoning.** For mathematical problems, we consider GSM8K (Cobbe et al., 2021) dataset that consists of high quality linguistically diverse school math problems created by human problem writers. MATH (Hendrycks et al., 2021) contains high-school competition mathematics problems from different areas of mathematics with varying degrees of difficulty. We also adopt MetaMathQA dataset (Yu et al., 2024), which is constructed through bootstrapping mathematical questions by rewriting the question from multiple perspectives. All three datasets are under the MIT license.

**Models.** We summarize licenses and terms of use of the models employed in this work. All model checkpoints are obtained from HuggingFace, if not stated otherwise. DeBERTa-v3-Base and DeBERTa-v2-XXL are under the MIT License. Llama-2 models are under the Llama 2 community license agreement.[7] Gemma models are under the Gemma terms of use.[8] The official LoRA checkpoints for DeBERTa-XXL available on Github are under the MIT license.[9]

---

[7] https://huggingface.co/meta-llama/Llama-2-7b-chat-hf/blob/main/LICENSE.txt
[8] https://ai.google.dev/gemma/terms
[9] https://github.com/microsoft/LoRA/releases/tag/DeBERTa

