# OpenReview forum: "PoLAR: Polar-Decomposed Low-Rank Adapter Representation"
_NeurIPS.cc/2025/Conference — NeurIPS 2025 poster_

### Official Review · Reviewer_guGQ · 2025-06-30

**Clarity:** 3
**Significance:** 1
**Originality:** 2
**Rating:** 4
**Confidence:** 3

**Summary:**

This paper introduces PoLAR, a new method for PEFT of LLMs that leverages the polar-like decomposition of LoRA matrices into unitary and a low-dimensional components. The authors provide theoretical motivation with rigorous analysis and demonstrate empirical gains over strong LoRA-based baselines across various model sizes and tasks.

**Questions:**

1. A comment:
The authors reference Zeng and Lee (2024) to motivate the expressiveness of low-rank adaptation, noting that LoRA can approximate any target transformer under mild conditions. However, the cited result crucially depends on using relatively high ranks—specifically, LoRA rank being at least half the embedding dimension—which makes the theoretical result more about approximation capacity than practical efficiency. In contrast, this paper aims to improve fine-tuning efficiency by operating in a very low-rank regime, which is far from the high-rank setting assumed in Zeng and Lee. Therefore, the claim that “the low-rank space offers sufficient expressiveness” is not directly supported by that citation.

2. In the theoretical analysis (e.g., lines 168–170), the convergence is derived under the assumption of zero training loss, which implies that the model operates in a high-rank regime—something the authors also acknowledge. However, this seems inconsistent with the practical low-rank settings  used in PEFT. Could the authors clarify how much insight the analysis in the full-rank or zero-loss regime provides for understanding the behavior of LoRA or PoLAR in low-rank regimes commonly used in practice?

3. In Theorem 1, the expression uses T \geq O(\cdot), but the Big-O notation already implies an upper bound. Would it not be more precise to write T = O(\cdot) or clarify the inequality with appropriate asymptotic notation?
Similarly, in the supplementary material (line 760, Condition C4), the equality marked as (a) is said to follow from the data being whitened. However, whitening ensures the identity covariance only in expectation, not deterministically. Shouldn’t this step be marked as an approximation, rather than a strict equality? While these may be minor points, they raise concerns about the precision and rigor of the theoretical claims in a paper that is otherwise theory-driven.


4. In Algorithm 1, the matrix \Theta is updated at the beginning of each iteration, whereas in Algorithm 2, it is updated at the very end. This inconsistency raises a question: what would happen if we swapped the update order in Algorithm 2 to match that of Algorithm 1? Is there a theoretical or empirical reason for this discrepancy? Some clarification would help understand whether the difference is deliberate or incidental.

5. In Table 2, both LoRA and PoLAR show lower average performance at rank 32 compared to some smaller ranks. This seems counterintuitive since higher ranks should, in principle, offer more capacity. Can the authors comment on why this drop occurs? Additionally, did you train all configurations (across different ranks) with the same number of epochs? If so, could underfitting or overfitting at higher ranks be a factor?

**Ethical Concerns:**

["NO or VERY MINOR ethics concerns only"]

**Final Justification:**

I appreciate the authors’ response, so increased the score.

**Limitations:**

The authors briefly mention some limitations of their approach, such as the orthogonal constraint’s potential optimization challenges, but they do not clearly discuss broader limitations or possible negative societal impacts. It would strengthen the paper if they included a more explicit reflection on the practical constraints of PoLAR (eg. the selection of \lambda) and any potential risks in real-world deployment scenarios.

**Paper Formatting Concerns:**

No formatting concern for the paper.

**Quality:**

3

**Strengths And Weaknesses:**

Strengths
1. The paper is well-organized and clearly written, with effective use of figures to illustrate key concepts and results.
2.  The authors present a theoretical connection between PEFT and matrix factorization. They analyze convergence properties under this framework in a rigorous way.
3. The proposed “landing trick” serves as a computationally efficient alternative to the more expensive Stiefel retraction, enabling the enforcement of orthonormality constraints with minimal overhead.
4. The experimental results include both small- and large-scale language models, demonstrating the scalability of the approach across model sizes.

Weaknesses:
1. The theoretical analysis on matrix factorization does not clearly translate into deeper understanding of PEFT or address any of the known theoretical challenges in the field. The connection feels superficial and lacks strong justification for its practical relevance.
2. The performance improvements shown in Table 2 and other benchmarks are modest. Given the competitiveness of the PEFT field, it is unclear whether polar decomposition offers a meaningful advantage over existing methods such as LoRA or QLoRA. More extensive ablation or comparisons would help clarify the benefits.

---

> ### Author Rebuttal · Authors · 2025-07-30
>
> We appreciate your thorough evaluation and your constructive feedback, which greatly helps us to further refine the work.
>
> **W1 The theoretical analysis on matrix factorization does not clearly translate into deeper understanding of PEFT or address any of the known theoretical challenges in the field. The connection feels superficial and lacks strong justification for its practical relevance.**
>
> **R1** We agree that there is a gap between theory and practice in the field of parameter-efficient fine-tuning, and the relevance of theoretical results for practice might sometimes be limited. However, we would like to highlight that our theoretical analysis on the matrix factorization problem, which is equivalent to a linear LoRA adaptation problem under whitened data, can be interpreted as a basic threshold for efficient low-rank techniques. If one can obtain provable benefits on such a problem, it is an indication that the technique might provide meaningful benefits in a practically relevant setup. As such, our theoretical analysis provides value not in that has immediate implications for state-of-the-art fine-tuning pipelines, but in that it highlights that PoLAR provably overcomes a basic requirement of providing benefits in an analytically tractable scenario.
>
> **W2 The performance improvements shown in Table 2 and other benchmarks are modest. Given the competitiveness of the PEFT field, it is unclear whether polar decomposition offers a meaningful advantage over existing methods such as LoRA or QLoRA. More extensive ablation or comparisons would help clarify the benefits.**
>
> **R2** Performance gains in LLM evaluation are highly dependent on evaluation framework used and the
> magnitude of differences can wildly differ, making evaluation an active area of research. We opted for
> _lm-eval-harness_ [1] with log-likelihood based evaluation which is a widely recognized and reproducible way to
> evaluate models. This evaluation methodology is in line with the
> multiple-choice question answering format of the commonsense reasoning benchmark and results in more conservative estimates of improvements.
>
>
> **Q1 A comment: The authors reference Zeng and Lee (2024) to motivate the expressiveness of low-rank adaptation, noting that LoRA can approximate any target transformer under mild conditions. However, the cited result crucially depends on using relatively high ranks—specifically, LoRA rank being at least half the embedding dimension—which makes the theoretical result more about approximation capacity than practical efficiency. In contrast, this paper aims to improve fine-tuning efficiency by operating in a very low-rank regime, which is far from the high-rank setting assumed in Zeng and Lee. Therefore, the claim that “the low-rank space offers sufficient expressiveness” is not directly supported by that citation.**
>
> **A1** We thank the reviewer for this interesting commment and agree that the current phrasing of the rank requirement as being mild should be improved by providing a more detailed contextualization. The cited work [2] expresses the required LoRA rank in terms of what they refer to as _rank-based functionality gap_, which can be informally understood as measuring the closeness between the frozen model and the target model. The worst-case in which this measure is equal to the embedding dimension is only relevant in the scenario where the frozen and target network are completely unrelated (e.g., independent random initialization). In any practical fine-tuning scenario, we expect the frozen and the target model to be closer than this worst case, decreasing the rank-based functionality gap and thereby the required rank to achieve desired approximation capacity. The empirical applicability of these theoretical findings is validated in Appendix G.9 of [2]. We will revise the draft accordingly by providing further context for the supporting claim made.
>
> **Q2 In the theoretical analysis (e.g., lines 168–170), the convergence is derived under the assumption of zero training loss, which implies that the model operates in a high-rank regime—something the authors also acknowledge. However, this seems inconsistent with the practical low-rank settings used in PEFT. Could the authors clarify how much insight the analysis in the full-rank or zero-loss regime provides for understanding the behavior of LoRA or PoLAR in low-rank regimes commonly used in practice?**
>
> **A2** The significance of the high-rank analysis hinges on the relation between the rank of a hypothetical "ground-truth" update (think of $r_A$) and the chosen rank $r$ of the adapter. The premise of the entire low-rank adaptation field is that $r_A$ is sufficiently low (think $r_A < r$ or at least $r_A \approx r$), making the case for the applicability of the high-rank, overparameterized regime in which $r > r_A$. An almost verbatim repetition of it is contained in the original LoRA paper, where it is hypothesized "that the change in weights during model adaptation also has a low 'intrinsic rank'" [3].
> In addition, there are some practical fine-tuning scenarios in which an approximately zero-loss, say below $10^{-5}$, is commonly encountered. This is for instance the case of fine-tuning 7B models with LoRA on some of the GLUE tasks.
>
>
> **Q3 In Theorem 1, the expression uses $T \geq O(\cdot)$, but the Big-O notation already implies an upper bound. Would it not be more precise to write $T = O(\cdot)$ or clarify the inequality with appropriate asymptotic notation? Similarly, in the supplementary material (line 760, Condition C4), the equality marked as (a) is said to follow from the data being whitened. However, whitening ensures the identity covariance only in expectation, not deterministically. Shouldn’t this step be marked as an approximation, rather than a strict equality? While these may be minor points, they raise questions about the precision and rigor of the theoretical claims in a paper that is otherwise theory-driven.**
>
> **A3** We apologize for the confusion on these two matters and address them in order. (i) We will improve the notation from "for all $T \geq O(\cdot)$" to "for all $T \geq c f(\cdot)$" where $c$ is some constant and $f(\cdot)$ the convergence rate. We remark though that writing $T = O(\cdot)$ wouldn't suffice here, as Theorem 1 implies a strictly stronger result. It not only implies the existential bound $T=O(\cdot)$, but also provides last-iterate convergence which is a result of PoLAR's monotone error. (ii) Whitening the data ensures that the empirical uncentered covariance matrix is exactly identity; this _holds deterministically_ and not just in expectation. The corresponding whitening operator can always be obtained from the eigendecomposition of the uncentered covariance matrix of the "raw" (unwhitened) data. We outline the procedure below. Following the notation in Appendix C.4, let $\tilde{D} \in \mathbb{R}^{n \times N}$ be the raw (unwhitened) data whose columns are training examples. Its empirical covariance matrix is given by $\tilde{\Sigma} = \tilde{D} \tilde{D}^\top$. From the eigendecomposition of $\tilde{\Sigma} = Q \Lambda Q^\top$, we obtain the whitening operator $W := \tilde{\Sigma}^{-1/2} = Q \Lambda^{-1/2}Q^T$. Multiplying the raw data with $W$ yields the whitened data $D := W\tilde{D}$. This ensures $DD^{\top} = W\tilde{D}\tilde{D}^\top W^\top = I_{n}$ exactly.
>
> **Q4 In Algorithm 1, the matrix $\Theta$ is updated at the beginning of each iteration, whereas in Algorithm 2, it is updated at the very end. This inconsistency raises a question: what would happen if we swapped the update order in Algorithm 2 to match that of Algorithm 1? Is there a theoretical or empirical reason for this discrepancy? Some clarification would help understand whether the difference is deliberate or incidental.**
>
> **A4** We remark that the order of the updates in Algorithm 2 does not matter, as the updates only depend on parameters from the previous iteration. In other words, the updates of Algorithm 2 can be thought of happening simultaneously rather than alternatingly. This is different from the update procedure in Algorithm 1 where the order matters due to the alternating procedure. The reason for this discrepancy is of theoretical nature, as the alternating update greatly facilitates the analysis.
>
> **Q5 In Table 2, both LoRA and PoLAR show lower average performance at rank 32 compared to some smaller ranks. This seems counterintuitive since higher ranks should, in principle, offer more capacity. Can the authors comment on why this drop occurs? Additionally, did you train all configurations (across different ranks) with the same number of epochs? If so, could underfitting or overfitting at higher ranks be a factor?**
>
> **A5** All configurations were trained with the same number of epochs. We agree that the lower average performance at rank 32 appears counterintuitive  and we hypothesize that the underlying reason for this phenomenon to occur is overfitting, as the rank of the optimal weight update could lie in-between 16 and 32.
>
> **References**
>
> [1] Biderman, Stella, et al. "Lessons from the Trenches on Reproducible Evaluation of Language Models". arXiv. 2024.
>
> [2] Zeng, Yuchen, et al. "The Expressive Power of Low-Rank Adaptation". ICLR. 2024.
>
> [3] Hu, Edward J., et al. "LoRA: Low-Rank Adaptation of Large Language Models." ICLR. 2022.

---

> ### Author Response · Authors · 2025-08-05
>
> Dear Reviewer, thank you once again for your detailed review and your insightful comments. Given that the Author-Reviewer discussion period comes to an end, we would like to take this opportunity to ask whether our rebuttal fully addressed your concerns.

---

> > ### Comment · Reviewer_guGQ · 2025-08-06
> >
> > Thank you for the detailed response. I will keep my current score but would be OK if the paper is accepted.
> >
> > > Whitening the data ensures that the empirical uncentered covariance matrix is exactly the identity; this holds deterministically and not just in expectation.
> >
> > Regarding this point, I’m still a bit confused. In the manuscript, it doesn’t seem clearly stated that whitening targets the _empirical_ covariance matrix;  rather, it seems to refer to the covariance in terms of the expectation. If the manuscript indeed explicitly states it’s the empirical version, please feel free to ignore this comment, and thank you again for the clarification.
> >
> > As for point W1, I remain unconvinced that the theoretical analysis offers a substantial contribution to the understanding of fine-tuning. I also find it unclear how directly the analysis connects to fine-tuning in practice. but, I acknowledge this is a subjective judgment, and I am no longer using this as a decisive factor in evaluating the paper.  **I will leave it to the AC and other reviewers to weigh this aspect.**

---

> > > ### Author Response · Authors · 2025-08-06
> > >
> > > Thank you for your reply and your consideration.
> > >
> > > > > Whitening the data ensures that the empirical uncentered covariance matrix is exactly the identity; this holds deterministically and not just in expectation.
> > > >
> > > > Regarding this point, I’m still a bit confused. In the manuscript, it doesn’t seem clearly stated that whitening targets the empirical covariance matrix; rather, it seems to refer to the covariance in terms of the expectation. If the manuscript indeed explicitly states it’s the empirical version, please feel free to ignore this comment, and thank you again for the clarification.
> > >
> > > We apologize for not making this clear. We will state explicitly that the whitening procedure refers to the empirical covariance matrix, and will also cite prior work that adopts the same strategy, e.g., [4].
> > >
> > >
> > > > As for point W1, I remain unconvinced that the theoretical analysis offers a substantial contribution to the understanding of fine-tuning. I also find it unclear how directly the analysis connects to fine-tuning in practice. but, I acknowledge this is a subjective judgment, and I am no longer using this as a decisive factor in evaluating the paper. I will leave it to the AC and other reviewers to weigh this aspect.
> > >
> > > Our analysis uses a prototypical problem that not only captures the low-rank structure relevant to LoRA/PoLAR, but also the bilinear form. The goal is (i) to show that PoLAR provably solves this instance and (ii) to demonstrate that, PoLAR converges faster than LoRA in this setting, indicating theoretical potential for more challenging regimes.
> > > - **Shared architecture.** The model preserves the bilinearity and low-rankness that drive the non-convexity in LoRA.
> > > - **Traceable baseline.** To compare PoLAR with LoRA rigorously, we require a setting where LoRA’s dynamics/rates can be tracked; this narrows down the viable choices, as general fine-tuning dynamics are notoriously hard to analyze.
> > >
> > > While we share the interest in understanding the dynamics of fine-tuning, we hope that our work can serve as a first step: methods that do not succeed for this prototypical problem are unlikely to extend to more theoretically involved regimes.
> > >
> > > **References**
> > >
> > > [4] Arora, Sanjeev, et al. "A Convergence Analysis of Gradient Descent for Deep Linear Neural Networks." ICML. 2018

---

### Official Review · Reviewer_1G8C · 2025-07-02

**Clarity:** 3
**Significance:** 3
**Originality:** 4
**Rating:** 5
**Confidence:** 4

**Summary:**

The work introduces a new method for fine-tuning large language models that addresses LoRA's underuse of the allocated rank, leading to low stable rank and collapsed update directions. PoLAR uses polar decomposition to factor low-rank updates into orthogonal direction matrices and a scale matrix, improving expressiveness. Combined with a Riemannian optimization strategy that avoids costly retractions, PoLAR achieves faster convergence and better GPU efficiency. PoLAR outperforms LoRA and DoRA  (the most recent reproducible SoTA) across models and tasks by maintaining higher stable rank.

**Questions:**

In Figure 5, the stable rank of LoRA seems to have a wider range compared to PoLAR different from other datasets. You mention this in L1060, but I couldn't find an explanation or hypothesis about it. Why do you think PoLAR doesn't provide any improvements here?

**Ethical Concerns:**

["NO or VERY MINOR ethics concerns only"]

**Final Justification:**

Considering the clarifications and additional experimental results the authors have provided in their answer, I'll keep my score the same (5: Accept).

I don't consider the weaknesses raised by the last reviewer to be actual weaknesses (performance and evaluation benchmarks), and I believe the authors' answer addresses these raised questions adequately. Same for weaknesses raised by other reviewers about GaLore, since GaLore is significantly different in terms of their goal and methodology.

**Limitations:**

yes

**Quality:**

3

**Strengths And Weaknesses:**

Strengths:
- Comprehensive and detailed theoretical grounding.
- The method scales well with model sizes.
- The implementation of the landing method seems stable and significantly improves convergence time.

Weaknesses:
- Optimizing over Steifel manifolds still introduces overhead and significant complexity over DoRA. This is considering that the performance improvements are noticeable but not necessarily significant in the tested benchmarks.
- It would also be interesting to see the difference in the jump in stable rank from this work compared to multi-lora applications. I say this because multi-lora methods bump up the rank by adding multiple matrices together, which could result in a rank that is higher than the sum of all ranks without changing anything about the typical adapter.
- The state of the art being compared against is a bit limited. I understand that in most of the tested tasks, AdaLoRA barely performs better than DoRA, but both AdaLoRA and LoRA+ are missing.
- Addressing the diversity collapse with this work leads to better stable rank within the adapter. Interpreting this from a LoRA’s perspective, the singular value spectrum is more uniform as opposed to being peaky. Other works like DoRA and LoRA+ have directly or indirectly addressed this issue, and their claims were mostly directed towards matching the training dynamics of full fine-tuning. Similar analyses would be interesting to see if the same behavior is seen from PoLAR.

Nitpick:
The references to the Appendix in the main text are too frequent, and it disturbs the flow of reading.

---

> ### Author Rebuttal · Authors · 2025-07-30
>
> We sincerely acknowledge your careful analysis of our work and thank you for recognizing its strengths.
>
> **W1 Optimizing over Stiefel manifolds still introduces overhead and significant complexity over DoRA. This is considering that the performance improvements are noticeable but not necessarily significant in the tested benchmarks.**
>
> **R1** Optimizing over the Stiefel manifold does add overhead. Nevertheless, the runtime of PoLAR compares favorably to that of DoRA (see the table below and Table 6 in the Appendix). To see why that is, consider a linear layer with weight $W\in \mathbb{R}^{m \times n}$ and $m>n$. As outlined in Alg. 2 and discussed in Section 4.4, PoLAR requires $O(m^2r)$ additional operations to obtain the landing field on top of the layer's gradient. In contrast, the gradient projection in Eq. 6 of DoRA [1], requires $O(m^2n)$ computations. Similarly, in the forward pass, DoRA requires the explicit construction of the low-rank weight change to compute the columnwise norm of the updated layer. This raises the complexity of the forward pass from $O(mr)$ for LoRA/PoLAR to $O(mnr)$ for DoRA.
>
> | Method    | Total Time (min)  |
> | ------    | ----------------: |
> | LoRA      | 28.74             |
> | DoRA      | 60.38             |
> | PoLAR     | 33.13             |
>
>
> **W2 It would also be interesting to see the difference in the jump in stable rank from this work compared to multi-lora applications. I say this because multi-lora methods bump up the rank by adding multiple matrices together, which could result in a rank that is higher than the sum of all ranks without changing anything about the typical adapter.**
>
> **R2** It is indeed interesting to see whether Multi-LoRA approaches that sequentially fit multiple low-rank updates obtain similar improvements in stable rank as PoLAR does. We find it plausible that using sequential fitting of PoLAR adapters could further elevate the stable rank, since LoRA's update often exhibits low stable rank compared to PoLAR.
>
> **W3 The state of the art being compared against is a bit limited. I understand that in most of the tested tasks, AdaLoRA barely performs better than DoRA, but both AdaLoRA and LoRA+ are missing.**
>
> **R3** Note that Table 1 includes a comparison to a variant of PoLAR that brings it closer to AdaLoRA by using the Euclidean gradient to update the parameters as well as a diagonal $\Theta$. This variant can be interpreted as a memory-matched variant of AdaLoRA that does not redistribute ranks among matrices and does not rely on overbudgeting during training (see Section 4.1 in [2]). We consider this to be a fair baseline, as AdaLoRA itself matches the parameter count to LoRA based on a final rank-budget of all adapters, but starts training with $1.5\times$ as many parameters. This overbudgeting increases memory load throughout training and makes a fair comparison hard. For this reason, we decided to benchmark PoLAR against the reduced variant described above.
>
>
> **W4 Addressing the diversity collapse with this work leads to better stable rank within the adapter. Interpreting this from a LoRA’s perspective, the singular value spectrum is more uniform as opposed to being peaky. Other works like DoRA and LoRA+ have directly or indirectly addressed this issue, and their claims were mostly directed towards matching the training dynamics of full fine-tuning. Similar analyses would be interesting to see if the same behavior is seen from PoLAR.**
>
> **R4** Thank you for raising this interesting point. While DoRA and LoRA+ indeed discuss concepts such as directional diversity or efficient feature learning, we do not see direct or indirect hints at how the spectrum of the low-rank update behaves.
>
> Based on your comment regarding the alignment with full fine-tuning, we ran further experiments to see whether PoLAR also exhibits a closer alignment with the training dynamics of full fine-tuning. To do so, we ran full fine-tuning on the commonsense reasoning benchmark and analyzed the stable rank distribution of $\Delta W = W' - W$ where $W'$ and $W$ refer to the fine-tuned and pre-trained weights, respectively. We tune the learning rate in $\lbrace 1 \times 10^{-5}, 4 \times 10^{-5} \rbrace$ and report the three quartiles of the stable rank distribution across layers below.
>
> $\eta = 1 \times 10^{-5}$
>
> | Percentile | BoolQ | PIQA | SIQA | HeSw | WiGr | ARC‑e | ARC‑c | OBQA | Avg |
> |-----------:|------:|-----:|-----:|-----:|-----:|------:|------:|-----:|-----:|
> | 0.25 | 65.38 | 76.9 | 50.15 | 80.28 | 50.1 | 27.47 | 20.87 | 23.53 | 49.34 |
> | 0.5  | 104.88 | 104.84 | 74.82 | 114.24 | 69.47 | 43.64 | 33.87 | 33.43 | 72.40 |
> | 0.75 | 150.88 | 132.42 | 91.56 | 151.8 | 89.96 | 58.7 | 44.36 | 42.76 | 95.31 |
>
>
> $\eta = 4 \times 10^{-5}$
>
> | Percentile | BoolQ | PIQA | SIQA | HeSw | WiGr | ARC‑e | ARC‑c | OBQA | Avg |
> |-----------:|------:|-----:|-----:|-----:|-----:|------:|------:|-----:|-----:|
> | 0.25 | 78.90 | 59.28 | 53.16 | 83.53 | 54.27 | 19.60 | 12.77 | 26.84 | 48.54 |
> | 0.5  | 106.37 | 70.55 | 60.76 | 106.61 | 60.84 | 29.71 | 21.52 | 34.78 | 61.39 |
> | 0.75 | 126.42 | 82.59 | 76.32 | 134.89 | 73.77 | 34.55 | 26.44 | 39.42 | 74.30 |
>
>
> Referring to the results $\eta=1\times 10^{-5}$, the average of the median stable rank across datasets is $72.4$ and the average first quartile is $49.3$, implying that most weight updates exhibit a relatively high stable rank. Therefore, the higher stable rank obtained by PoLAR can be interpreted from the angle of closer alignment with full fine-tuning.
>
> **Q1 In Figure 5, the stable rank of LoRA seems to have a wider range compared to PoLAR different from other datasets. You mention this in L1060, but I couldn’t find an explanation or hypothesis about it. Why do you think PoLAR doesn’t provide any improvements here?**
>
> **A1** While we can only provide a speculative answer, we hypothesize that this is related to the training set size and task format. ARC-c and ARC-e [3] are by some margin the smallest datasets in the benchmark with roughly 1k and 2k training examples, respectively. We suspect that this dataset size is not sufficiently large s.t. meaningful, orthogonal directions can be found. Moreover, the ARC datasets differ from the other datasets in that they do not provide additional per-choice context to answer the question. In contrast, the other datasets do incorporate some form of minimal context: HellaSwag has a context field; BoolQ provides a passage; Social-IQA gives a context for the question; WinoGrande provides the full sentence with a blank; OBQA provides related facts; and PIQA frames the question as a goal.
>
> **References**
>
> [1] Liu, Shih-Yang, et al. "DoRA: Weight-Decomposed Low-Rank Adaptation." ICML. 2024.
>
> [2] Zhang, Qingru, et al. "Adaptive Budget Allocation for Parameter-Efficient Fine-Tuning." ICLR. 2023.
>
> [3] Clark, Peter, et al. "Think you have Solved Question Answering? Try ARC, the AI2 Reasoning Challenge.", arXiv. 2018.

---

> > ### Comment · Reviewer_1G8C · 2025-08-04
> >
> > Thank you for the clarifications and additional experiments. All my concerns have been addressed and I'm keeping my score the same.

---

### Official Review · Reviewer_DiUw · 2025-07-02

**Clarity:** 3
**Significance:** 2
**Originality:** 3
**Rating:** 5
**Confidence:** 4

**Summary:**

This paper introduces PoLAR, a parameter-efficient fine-tuning (PEFT) method designed to enhance the directional diversity of weight updates ($\Delta W$) in LoRA, thereby addressing the low stable rank issue. Experimental results demonstrate that PoLAR achieves consistent accuracy improvements across diverse datasets and base models.

**Questions:**

1. How does PoLAR compare to GaLore and other methods addressing directional diversity (*e.g.*, SVD-based approaches)? Please provide a discussion on similarities and distinctions.

2. Is it possible to provide a convergence analysis for the alternating update algorithm used in PoLAR’s implementation, given that the theoretical analysis relies on Riemannian gradient descent?

3. How does the proposed method performs compared to methods like GaLore, Fira, SLTrain, and full-rank fine-tuning?

4. Does PoLAR offer any memory efficiency advantages over LoRA, or are there plans to address this limitation in future work?

**Ethical Concerns:**

["NO or VERY MINOR ethics concerns only"]

**Final Justification:**

The authors' rebuttal has effectively addressed my concerns regarding the related work, theoretical analysis, and experimental results. Accordingly, I have revised my overall score to Accept.

**Limitations:**

yes

**Paper Formatting Concerns:**

No formatting issues were identified.

**Quality:**

3

**Strengths And Weaknesses:**

**Strengths**

1. **Clear motivations.** The paper effectively identifies the directional diversity collapse in low-rank adapters, articulates its impact, and explains how PoLAR addresses this issue.
2. **Theoretically sound analysis.** The study provides a comprehensive theoretical framework, including problem definition and convergence guarantees, strengthening the method’s foundation.
3. **Practical implementation.** This paper introduces an alternating update method to solve the low-rank approximation problem with reduced computational requirement.

**Weaknesses**

1. **Limited discussions of related  work.** One major reason for the low directional diversity of LoRA may be attributed to the non-orthogonal projection. There are other related work that attempts to solve the same problem, such as SVD-based GaLore (Zhao et al., 2024) and other related works. The discussion on the similarity and distinctions between the proposed method and GaLore-like methods are necessary.
2. **Misalignment between theory and practice.** While convergence is analyzed under Riemannian gradient descent, the practical implementation relies on an alternating update algorithm. An analysis of the convergence properties of this algorithm would bridge this gap.
3. **Insufficient experiments.** The experiments compare PoLAR only with LoRA and DoRA. Including additional methods like GaLore, Fira (Chen et al., 2024), SLTrain (Han et al., 2024), and full-rank fine-tuning would provide a more comprehensive evaluation.
4. **Limited memory efficiency.** Unlike recent PEFT methods that prioritize both parameter and memory efficiency, PoLAR does not demonstrate memory advantages over LoRA, potentially limiting its practical applicability.
5. **The organization and presentations can be further improved.**
   - Key content, such as Appendix C.1, is relegated to the appendices despite its importance and should be integrated into the main text. Frequent references to appendices disrupt the reading flow.
   - [Minor] The core idea’s presentation could be more concise, avoiding excessive use of technical terms and complex concepts.

**References**

[R1] X. Chen, et al. (2024). Fira: Can We Achieve Full-rank Training of LLMs Under Low-rank Constraint? *arXiv preprint: 2410.01623*.

[R2] A. Han, J. Li, W. Huang, M. Hong, & A. Takeda, (2024). SLTrain: A Sparse Plus Low-Rank Approach for Parameter and Memory Efficient Pretraining. *NeurIPS 2024*.

---

> ### Author Rebuttal · Authors · 2025-07-30
>
> We are thankful for your suggestions and insightful comments, bringing up important points which help us in sharpening our contribution.
>
> **Q1 How does PoLAR compare to GaLore and other methods addressing directional diversity (e.g., SVD-based approaches)? Please provide a discussion on similarities and distinctions.**
>
> **A1** We thank the reviewer for bringing this up and agree that there are indeed some similarities to GaLore [1]. First and foremost, PoLAR as well as GaLore see low-rank structure during training as an opportunity to improve memory efficiency. In addition, both setups use semi-orthogonal matrices for compression and decompression operations. Nevertheless, there are also substantial differences; PoLAR imposes low-rank structure on the weight updates whereas GaLore restricts the gradients driving those updates to be low-rank. Moreover, the orthogonality of the subspace is enforced through different means and the frequency of subspace changes differs. GaLore updates the subspace only periodically via an SVD of the gradient, which introduces substantial overhead. PoLAR, on the other hand, updates the subspace at every iteration through the use of the landing field, which can be obtained using matrix multiplications only. Moreover, there are substantial differences in storing fine-tuned models; PoLAR natively stores a compressed representation requiring storage of $(m+n+r)r$ parameters. In contrast, GaLore requires storage of $mn$ parameters.
>
> **Q2 Is it possible to provide a convergence analysis for the alternating update algorithm used in PoLAR’s implementation, given that the theoretical analysis relies on Riemannian gradient descent?**
>
> **A2** We apologize for not making it clear in the main text. The theoretical analysis uses alternating updates (see Alg. 1 and Eq. 5a-5c). The updates in Alg. 2 (PoLAR), on the other hand, can be thought of happening simultaneously rather than alternatingly. The reason for this discrepancy is of theoretical nature, as the alternating update greatly facilitates the analysis. Nevertheless, we do believe that our analysis can be extended to cover the updates used in the practical variant.
>
> **Q3 How does the proposed method performs compared to methods like GaLore, Fira, SLTrain, and full-rank fine-tuning?**
>
> **A3** We provide results for GaLore+AdamW and full-rank fine-tuning below. We adopt our commonsense reasoning training setup from Table 2 with rank $16$. For GaLore's hyperparameters, we take inspiration from their fine-tuning experiments in Appendix D [1] and set the scale to $2$. We set the update projection gap s.t. the projections are updated once 15\% increments of training are completed and tune the learning rate $\eta \in \lbrace 1 \times 10^{-5}, 4 \times 10^{-5} \rbrace$. We additionally show results for PoLAR at rank $16$ for ease of reference.
>
>  |       | $\eta$    | BoolQ | PIQA | SIQA | HeSw | WiGr | ARC‑e | ARC‑c | OBQA | Avg  |
>  |-------|-------|-------|------|------|------|------|-------|-------|------|------|
>  | GaLore | $1 \times 10^{-5}$ | 81.93 | 80.2 | 56.4 | 80.23 | 81.53 | 80.22 | 47.53 | 44.8 | 69.1 |
>  | GaLore | $4 \times 10^{-5}$ | 86.48 | 82.81 | 61.26 | 83.84 | 78.85 | 83.08 | 54.44 | 57.8 | 73.57 |
>  | Full-FT | $1 \times 10^{-5}$ | 84.8  | 81.72 | 59.21 | 81.71 | 81.06 | 82.95 | 53.67 | 51.2 | 72.04 |
>  | Full-FT | $4 \times 10^{-5}$ | 87.68 | 80.09 | 55.63 | 80.93 | 76.56 | 81.36 | 58.02 | 54.8 | 71.88 |
>  | PoLAR  |                     |88.01  | 82.92 | 59.93 | 82.68 | 81.77 | 81.82 | 57.51 | 57.40 | 74.00 |
>
> We remark that the average performance of GaLore at $4 \times 10^{-5}$ comes close to the performance of PoLAR at rank $16$ while full fine-tuning appears to lag behind.
>
>
> **Q4 Does PoLAR offer any memory efficiency advantages over LoRA, or are there plans to address this limitation in future work?**
>
> **A4** PoLAR's memory footprint is mostly comparable to that of LoRA, as it only adds $r^2$ additional trainable parameters to the procedure. We report peak memory usage for Llama2-7B on PIQA in Table 6 of Appendix F.2 and show the relevant values here for convenience.
>
>  | Method    | Memory (GB)  |
>  | ------    | ----------------: |
>  | LoRA      | 52.37             |
>  | PoLAR     | 52.68             |
>
> The memory footprint of PoLAR could theoretically be further reduced through a reparameterization of the direction matrices living on the Stiefel manifold. While our current $m \times r$ form exhibits an avoidable overhead of $r + {r \choose 2}$ (see, e.g., Section 3.3.2 in [2]), we do not think that these rather small gains in memory warrant the extra computational cost and complexity associated with it. The bulk of the memory is still consumed by the model itself with memory complexity $O(mn)$, while the low-rank factors only consume $O((m+n)r)$. In the table above PoLAR has approximately 56M trainable low-rank parameters with each of them consuming 12 bytes (parameter and gradient with 2 bytes each and Adam states with 4 bytes each), totalling less than 1GB of memory.
>
> **W1 Limited discussions of related work. One major reason for the low directional diversity of LoRA may be attributed to the non-orthogonal projection. There are other related work that attempts to solve the same problem, such as SVD-based GaLore (Zhao et al., 2024) and other related works. The discussion on the similarity and distinctions between the proposed method and GaLore-like methods are necessary.**
>
> **R1** We agree that GaLore-like methods share some similarities, but also exhibit significant differences. We included a discussion of those in **A1** and will also integrate it in a revised version.
>
> **W2 Misalignment between theory and practice. While convergence is analyzed under Riemannian gradient descent, the practical implementation relies on an alternating update algorithm. An analysis of the convergence properties of this algorithm would bridge this gap.**
>
> **R2** As in our reply to **Q2**, we apologize for not making it clear that our practical variant does not use the alternating update procedure which was used in the theoretical analysis. Note that the use of the alternating RGD procedure in the practical variant would double the computational cost, as it requires two forward and backward passes per optimizer step. Moreover, Table 4 shows the runtime advantage of the practical variant over RGD.
>
> **W3 Insufficient experiments. The experiments compare PoLAR only with LoRA and DoRA. Including additional methods like GaLore, Fira (Chen et al., 2024), SLTrain (Han et al., 2024), and full-rank fine-tuning would provide a more comprehensive evaluation.**
>
> **R3** We provide results for GaLore and full fine-tuning in **A3**.
>
> **W4 Limited memory efficiency. Unlike recent PEFT methods that prioritize both parameter and memory efficiency, PoLAR does not demonstrate memory advantages over LoRA, potentially limiting its practical applicability.**
>
> **R4** As mentioned in **A4**, PoLAR maintains the memory footprint of LoRA, taking out the additional $O(r^2)$ memory incurred by the $\Theta$ matrix.
>
> **W5 The organization and presentations can be further improved. Key content, such as Appendix C.1, is relegated to the appendices despite its importance and should be integrated into the main text. Frequent references to appendices disrupt the reading flow. [Minor] The core idea’s presentation could be more concise, avoiding excessive use of technical terms and complex concepts.**
>
> **R5** We thank the reviewer for this important feedback on the overall presentation of our work. We agree that the definition of the polar decomposition, a key concept in our work, should have a more prominent position in the main text, and we will change it accordingly. Also, we will reduce the number of references to the appendix.
>
> **References**
>
> [1] Zhao, Jiawei, et al. "GaLore: Memory-Efficient LLM Training by Gradient Low-Rank Projection." ICML. 2024.
>
> [2] Absil, Pierre-Antoine, et al. "Optimization Algorithms on Matrix Manifolds." 2008.
>
> [3] Ponkshe, Kaustubh, et al. "Initialization using Update Approximation is a Silver Bullet for Extremely Efficient Low-Rank Fine-Tuning." arXiv. 2025

---

> ### Author Response · Authors · 2025-08-05
>
> Dear Reviewer, we would like to thank you again for your time and your helpful feedback. As the Author-Reviewer discussion period ends soon, we would like to ask whether you have any remaining concerns.

---

> > ### Comment · Reviewer_DiUw · 2025-08-06
> >
> > I appreciate the authors' comprehensive responses, which have effectively addressed my concerns. Accordingly, I will raise my score.

---

### Official Review · Reviewer_35pC · 2025-07-04

**Clarity:** 4
**Significance:** 3
**Originality:** 4
**Rating:** 5
**Confidence:** 4

**Summary:**

The authors start out with looking at PEFT -- parameter efficient fine-tuning. Their key enabling insight, which serves as strong motivation for this work, is that vanilla LoRA adapters demonstrate low stable rank, and severe underutilization of available algebraic rank and, thus, expressivity. Thus, while structurally, capacity is available, it is not utilized due to demonstrated directionality collapse (DC). What's interesting is that the effective stable rank over time converges to 1. Armed with this insight, the authors propose to factor low rank updates (ie \delta{W} updates to W_0) into direction matrices and a separate scale matrix. This parameterization is then suported by a Riemannian optimization. As a result of this, the authors demonstrate exponentially faster convergence rate compared to vanilla LoRA! Findings generalize across tasks, models, and model sizes. Impressive work.

**Questions:**

* would you be able to demonstrate your proposed technique on the Silver Bullet, which reduces a set of FT-trainable parameters to R (rxr), instead of the typical A and B LoRA matrices. This reduction to R may lead to higher stable rank. I'm wondering whether Silver Bullet led to improving/mitigating directional collapse as well.
* To what extent is PoLAR sensitive to initialization? Conversely, if we use the best possible initialization, does this help mitigate the stable rank or DC in LoRA? In other words, can initialization help with this problem, or is it fundamentally an issue with the optimization step and, therefore, truly requires proposed decomposition?

[1] Initialization using Update Approximation is a Silver Bullet for Extremely Efficient Low-Rank Fine-Tuning, https://arxiv.org/abs/2411.19557. Submitted on 29 Nov 2024 (v1).

**Ethical Concerns:**

["NO or VERY MINOR ethics concerns only"]

**Limitations:**

yes

**Quality:**

4

**Strengths And Weaknesses:**

**Strengths**

* very good presentation. Well written, thoughtful, thorough disposition throughout the paper. I enjoyed reading it. Best in the pile.
* the proposed technique is mathematically sound
* strong key enabling insight, showing directional collapse in standard LoRA adapters as well as very low stable rank and, therefore, poor utilization of available expressivity. The fact that varying r typically doesn't help is strongly aligned with this insight.
* findings generalize across tasks, models, and even model sizes (up to 27B)


**Weaknesses**
* some important LoRA initialization work hasn't been cited [1]. It would also be great to see if PoLAR can generalize to this type of LoRA parameterization (see questions).

[1] Initialization using Update Approximation is a Silver Bullet for Extremely Efficient Low-Rank Fine-Tuning, https://arxiv.org/abs/2411.19557. Submitted on 29 Nov 2024 (v1).

---

> ### Author Rebuttal · Authors · 2025-07-30
>
> We are grateful for your insightful comments and positive feedback, which guided further improvements to our work.
>
> **Q1 would you be able to demonstrate your proposed technique on the Silver Bullet, which reduces a set of FT-trainable parameters to R (rxr), instead of the typical A and B LoRA matrices. This reduction to R may lead to higher stable rank. I'm wondering whether Silver Bullet led to improving/mitigating directional collapse as well.**
>
> **A1** We thank the reviewer for bringing this interesting related work to our attention. The parameterization does indeed share similarities, as it contains a trainable $r \times r$ matrix and two semi-orthogonal factors. However, we remark that freezing the semi-orthogonal factors might cause a loss in expressivity compared to PoLAR, as the row and column spaces of the low-rank update remain fixed. As we discuss in **A2**, the increase in stable rank occurs during training, s.t. the question of whether a trainable $r \times r$ matrix alone achieves similar benefits as PoLAR boils down to whether the parameterization alone already achieves those benefits or the trainability of the direction factors is required as well. Apart from that, we find it unlikely that a better initialization alone (without any constraints on the factors throughout training) will yield similar results; see the discussion in **A2**.
>
> **Q2 To what extent is PoLAR sensitive to initialization? Conversely, if we use the best possible initialization, does this help mitigate the stable rank or DC in LoRA? In other words, can initialization help with this problem, or is it fundamentally an issue with the optimization step and, therefore, truly requires proposed decomposition?**
>
> **A2** We did not test alternative initialization schemes and how these would relate to the stable rank dynamics. Note though that Fig. 7 demonstrates that the increase in stable rank appears throughout training, rather than at initialization. Therefore, we are unsure whether a more advanced initialization alone mitigates the problem of low stable rank. Other than that, the initialization scheme proposed by LoRA-SB could further improve our PoLAR procedure, as the Silver Bullet initialization is informed by the SVD of the gradient evaluated at the pre-trained model. Going beyond our initialization that is randomly sampled from the Stiefel manifold, the information contained in the SB-basis might provide complementary benefits. A thorough investigation on whether these purported benefits materialize empirically is an interesting direction for follow-up work. We will include a discussion of LoRA-SB in a revised version.
>
> **W1 some important LoRA initialization work hasn't been cited [1]. It would also be great to see if PoLAR can generalize to this type of LoRA parameterization (see questions).**
>
> **R1** We refer to **A1**/**A2** for a discussion of the relationship to LoRA-SB. Given the potential for follow-up work, we will include a discussion of LoRA-SB in a revised draft.

---

> > ### Comment · Reviewer_35pC · 2025-08-06
> > **better initialization schemes are argued to be orthogonal**
> >
> > The authors alleviated my _mild_ concern/question as to whether a better initialization scheme for LoRA adapters could be sufficient to maintain a higher stable rank through training. I would've still liked to see some experimental validation of this, but I understand that the rebuttal period might not give you sufficient time to do so. If this paper is accepted, it will be great to see an ablation across better/SOTA initialization algorithms. The collapse of the stable rank is a dynamic property -- that I agree with. It's more a question of the extent to which Polar will continue to offer an improvement on top of that. This should be very easy to show by simply monitoring the stable rank for LoRA-SB.
> >
> > I am keeping my score the same. Thanks for the thoughtful response.

---

### Decision · Program_Chairs · 2025-09-17

**Decision:**

Accept (poster)

**Comment:**

In this paper, the authors propose a novel parameter-efficient fine-tuning method to address the low stable rank issue in LoRA. The presentation is clear, the method is technically sound, and the authors have adequately addressed the reviewers' concerns during rebuttal. I recommend acceptance.